# Effective End-to-end Unsupervised Outlier Detection via Inlier Priority of Discriminative Network

**Siqi Wang**[1]*, **Yijie Zeng**[2]*, **Xinwang Liu**[1], **En Zhu**[1], **Jianping Yin**[3], **Chuanfu Xu**[1], **Marius Kloft**[4]

[1]National University of Defense Technology,      [2]Nanyang Technological University
[3]Dongguan University of Technology,      [4]Technische Universität Kaiserslautern
wangsiqi10c@nudt.edu.cn, yzeng004@e.ntu.edu.sg, {xinwangliu, enzhu}@nudt.edu.cn
jpyin@dgut.edu.cn, xuchuanfu@nudt.edu.cn, kloft@cs.uni-kl.de

## Abstract

Despite the wide success of deep neural networks (DNN), little progress has been made on end-to-end unsupervised outlier detection (UOD) from high dimensional data like raw images. In this paper, we propose a framework named $E^3Outlier$, which can perform UOD in a both **effective** and **end-to-end** manner: First, instead of the commonly-used autoencoders in previous end-to-end UOD methods, $E^3Outlier$ for the first time leverages a discriminative DNN for better representation learning, by using *surrogate supervision* to create multiple pseudo classes from original unlabelled data. Next, unlike classic UOD that utilizes data characteristics like density or proximity, we exploit a novel property named *inlier priority* to enable end-to-end UOD by discriminative DNN. We demonstrate theoretically and empirically that the intrinsic class imbalance of inliers/outliers will make the network prioritize minimizing inliers' loss when inliers/outliers are indiscriminately fed into the network for training, which enables us to differentiate outliers directly from DNN's outputs. Finally, based on inlier priority, we propose the negative entropy based score as a simple and effective outlierness measure. Extensive evaluations show that $E^3Outlier$ significantly advances UOD performance by up to 30% AUROC against state-of-the-art counterparts, especially on relatively difficult benchmarks.

## 1   Introduction

An outlier is defined as "an observation which deviates so much from the other observations as to arouse suspicions that it was generated by a different mechanism" [1]. In some context of the literature, outliers are also referred as anomalies, deviants, novelties or exceptions [2]. Outlier detection (OD) has broad applications such as financial fraud detection [3], intrusion detection [4], fault detection [5], etc. Various solutions have been proposed to tackle OD (see [6] for a comprehensive review). Based on the availability of labels, those solutions can be accordingly divided into three categories below [7]: **1)** *Supervised* OD (SOD) deals with the case where a training set is provided with both labelled inliers/outliers, but it suffers from expensive data labelling and the rarity of outliers in practice [6]. **2)** *Semi-supervised* OD (SSOD) only requires pure single-class training data that are labelled as "inlier" or "normal", and no outlier is involved during training. **3)** *Unsupervised* OD (UOD) handles completely unlabelled data mixed with outliers, and no data label is provided for training at all.

In this paper we will limit our discussion to **UOD**, as most data are unlabelled in practice and UOD is the most widely applicable [7]. In particular, two *clarifications of concepts* must be made: First, in some literature like [8, 9], "unsupervised outlier/anomaly detection" actually refers to SSOD rather than UOD by our definition. Second, a recent topic is *out-of-distribution sample detection*, which

detects samples that are not from the distribution of training samples [10, 11, 12]. It is similar to SSOD, but it requires well-labelled multi-class data for training rather than single-class data in SSOD. Both cases above are different from UOD that does not use any label information in this paper.

Recently, surging image/video data have inspired important UOD applications in computer vision, e.g. refining web image query results [13] and video abnormal event detection [14]. Unfortunately, despite the remarkable success of end-to-end deep neural networks (DNN) in computer vision [15], an *effective* and *end-to-end* UOD strategy is still under exploration: State-of-the-art methods [16, 17, 18] unexceptionally rely on deep autoencoders (AE) or convolutional autoencoders (CAE) to realize easily achievable DNN based UOD, but they all suffer from AE/CAE's ineffective representation learning (detailed in Sec. 3.1). Motivated by this gap, we aim to address UOD in a both effective and end-to-end fashion, with the application to detect outlier images from contaminated datasets.

**Contributions.** This paper proposes an **e**ffective and **e**nd-to-**e**nd UOD framework named $E^3Outlier$. Specifically, our contributions can be summarized below: **1)** To liberate DNN based UOD from AE/CAE's ineffective representation learning, $E^3Outlier$ for the first time enables us to adopt powerful discriminative DNN architectures like ResNet [19] for representation learning in UOD. This is realized by *surrogate supervision*, which creates multiple pseudo classes by imposing various simple operations on original unlabelled data. **2)** $E^3Outlier$ discovers outliers based on a novel property of discriminative network named *inlier priority*, which evidently differs from previous methods that utilize certain data characteristics (e.g. density, proximity, distance) to perform UOD. Through both theory and experiments, we demonstrate that inlier priority will encourage the network to prioritize the reduction of inliers' loss during network training. On the foundation of inlier priority, $E^3Outlier$ is able to achieve end-to-end UOD by directly inspecting the DNN's outputs, which reflect each datum's priority level. In this way, it avoids the possible suboptimal performance yielded by feeding the DNN's learned representations into a decoupled UOD method [20]. **3)** Based on inlier priority, we explore several strategies and propose a simple and effective negative entropy based score to measure outlierness. Extensive experiments report a remarkable improvement by $E^3Outlier$ against state-of-the-art methods, particularly on relatively difficult benchmarks for unsupervised tasks.

## 2   Related Work

**Classic Outlier Detection.**   For classic SOD, labelled data are utilized to build discriminative models by well-studied supervised binary/multi-class classification techniques, such as support vector machine (SVM) [21], random forest [22] and recent XGBoost [23]. In contrast, SSOD that requires only labelled inliers is much more prevalent, and it is also called *one-class classification* [24] or *novelty detection* [25]. Classic SSOD usually involves training a model on pure inliers and detecting those data that evidently deviate from this model as outliers, and representative SSOD methods include SVM based methods [26, 27], replicator network/autoencoders [28, 29], principle component analysis (PCA)/kernel based PCA [30, 31]. Compared with SOD and SSOD, UOD handles the most challenging case where no labelled data is available. Classic UOD methods discover outliers by examining the basic characteristics of data, such as statistical properties [32], cluster membership [33, 34], density [35, 36, 37], proximity [38, 39], etc. Besides, ensemble methods like isolation forest [40] and its variants [41, 42] are popular in UOD. However, most state-of-the-art UOD methods like [40, 37, 13] still require manual feature extraction from high dimensional data like raw images.

**DNN based Outlier Detection.** DNN's recent success naturally inspires DNN based OD [20]. For SOD, discriminative DNN can be directly applied, while the main issue is the class imbalance of inliers/outliers [20], which is explored by [43, 44, 45, 46]. For SSOD, the case is more difficult as only labelled inliers are provided. DNN solutions for SSOD fall into three types: Mainstream DNN based SSOD methods handle high dimensional data by label-free generative models, i.e. AE/CAE [47, 48, 49, 50] and generative adversarial network (GAN) [51, 52, 53]. The second type extends classic SSOD methods into their deep counterparts, such as deep support vector data description [54] and deep one-class SVM [55]. The last type turns SSOD into SOD by certain means like introducing reference datasets [56], intra-class splitting [57], geometric transformations [58] or synthetic outlier generation [59]. As to UOD, the absence of both inlier and outlier label poses great challenges to combining UOD with DNN, which results in much less progress than SOD and SSOD. In addition to the naive solution that feeds DNN's learned representations into a separated UOD method [20], to our best knowledge only the following works have explored DNN based UOD: Zhou et al. [17] propose a decoupled solution that combines a deep AE with Robust PCA, which decomposes the inputs into a

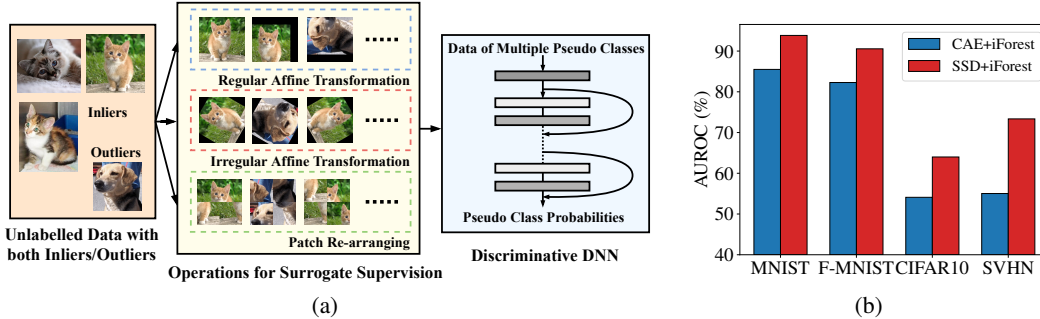

Figure 1: Surrogate supervision workflow (left) and the comparison of learned representations (right).

low-rank part from inliers and a sparse part from outliers; For end-to-end UOD, Xia et al. [16] use deep AE directly and propose a variant that estimates inliers by seeking a threshold that maximizes the inter-class variance of AE's reconstruction loss. A loss function is designed to encourage the separation of estimated inliers/outliers; Zong et al. [18] jointly optimize a deep AE and an estimation network to perform simultaneous representation learning and density estimation for end-to-end UOD.

**Surrogate Supervision.** Recent studies propose surrogate supervision to improve DNN pre-training for downstream high-level tasks like image classification and object detection. It imposes certain operations on unlabelled data to create corresponding pseudo classes and provide supervision signal, such as rotation [60], image patch permutation [61], clustering [62], etc. Surrogate supervision is also called self-supervision (see [63] for a comprehensive survey), but we use surrogate supervision to better distinguish it from AE/CAE, which are also viewed as "self-supervised" in some context. To our best knowledge, our work is the first to connect surrogate supervision with end-to-end UOD.

## 3 The proposed $E^3Outlier$ Framework

**Problem Formulation of UOD.** Considering a data space $\mathcal{X}$ (in this context the space of images), an unlabelled data collection $X \subseteq \mathcal{X}$ consists of an inlier set $X_{in}$ and an outlier set $X_{out}$, which originate from fundamentally different underlying distributions [1]. Our goal is to obtain an end-to-end UOD method $S(\cdot)$ that in the ideal case outputs $S(\mathbf{x}) = 1$ for inlier $\mathbf{x} \in X_{in}$ and $S(\mathbf{x}) = 0$ for outlier $\mathbf{x} \in X_{out}$. In practice, a smaller $S(\mathbf{x})$ indicates a higher likelihood of $\mathbf{x}$ to be an outlier.

### 3.1 Surrogate Supervision Based Effective Representation Learning for UOD

**Why NOT AE/CAE?** We note that existing DNN based UOD methods rely on AE/CAE [16, 17, 18]. However, it is hard for them to handle relatively complex datasets like CIFAR10 and SVHN: As our UOD experiments[2] show in Fig. 1(b), even a sophisticated deep CAE with isolation forest [40] only performs slightly better than random guessing (50% AUROC). Similar results are reported in other AE/CAE based unsupervised tasks like deep clustering [64, 65]. This is because AE/CAE typically adopt mean square error (MSE) as loss function, which forces AE/CAE to focus on reducing low-level pixel-wise error that is not sensitive to human perception, rather than learning high-level semantic features [66, 67]. Therefore, AE/CAE based representation learning is often ineffective.

**Surrogate Supervision.** Discriminative DNNs like ResNet [19] and Wide ResNet (WRN) [68] have proved to be highly effective in learning high-level semantic features, but they have not been explored in UOD due to the lack of supervision. To remedy the absence of data labels and substitute AE/CAE, we propose a *surrogate supervision based discriminative network* (SSD) for more effective representation learning in UOD. Specifically, we first define an operation set with $K$ operations $\mathcal{O} = \{O(\cdot|y)\}_{y=1}^{K}$, where $y$ represents the pseudo label associated with the operation $O(\cdot|y)$. Applying an operation $O(\cdot|y)$ to $\mathbf{x}$ can generate a new datum $\mathbf{x}^{(y)} = O(\mathbf{x}|y)$, and all data generated by the operation $O(\cdot|y)$ belong to the pseudo class with pseudo label $y$. Next, given a datum $\mathbf{x}^{(y')}$, a discriminative DNN with a $K$-node softmax layer is trained to classify the type of applied

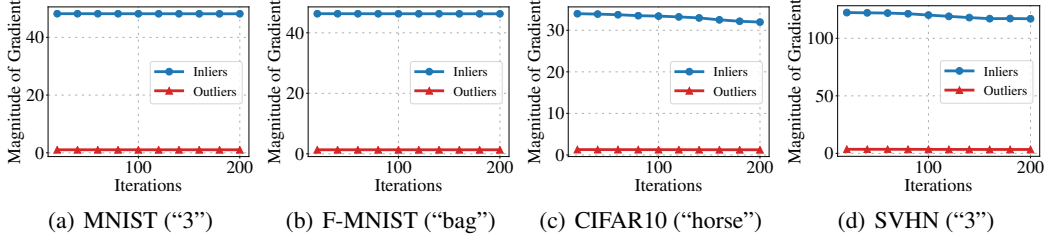

Figure 2: Inliers and outliers' gradient magnitude on example cases of benchmark datasets during SSD training. The class used as inliers is in brackets.

operation, i.e. the DNN is supposed to classify $\mathbf{x}^{(y')}$ into the $y'$-th pseudo class. With $P^{(y)}(\cdot)$ and $\boldsymbol{\theta}$ denoting the probability output by the $y$-th node of softmax layer and DNN's learnable parameters respectively, DNN's output probability vector for $K$ operations is $P(\mathbf{x}^{(y')}|\boldsymbol{\theta}) = [P^{(y)}(\mathbf{x}^{(y')}|\boldsymbol{\theta})]_{y=1}^K$. To train such a DNN with an unlabelled data collection $X = \{\mathbf{x}_i\}_{i=1}^N$, the objective function is:

$$\min_{\boldsymbol{\theta}} \frac{1}{N} \sum_{i=1}^N \mathcal{L}_{SS}(\mathbf{x}_i|\boldsymbol{\theta}) \tag{1}$$

where $\mathcal{L}_{SS}(\mathbf{x}_i|\boldsymbol{\theta})$ is the loss incurred by $\mathbf{x}_i$ under surrogate supervision. When the commonly-used cross entropy loss is used to classify pseudo classes of surrogate supervision, it can be written as:

$$\mathcal{L}_{SS}(\mathbf{x}_i|\boldsymbol{\theta}) = -\frac{1}{K} \sum_{y=1}^K \log(P^{(y)}(\mathbf{x}_i^{(y)}|\boldsymbol{\theta})) = -\frac{1}{K} \sum_{y=1}^K \log(P^{(y)}(O(\mathbf{x}_i|y)|\boldsymbol{\theta})). \tag{2}$$

As to the operation set $\mathcal{O}$, each operation $O(\cdot|y) \in \mathcal{O}$ is defined as a combination of one or more basic transformations from the following transformation sets: **1)** *Rotation*: This set's transformations clock-wisely rotate images by a certain degree. **2)** *Flip*: This set's transformations refer to flipping the image or not. **3)** *Shifting*: This set's transformations shift the image by some pixels along $x$-axis or $y$-axis. **4)** *Patch re-arranging*: This set's transformations partition the image into several equally-sized patches and re-organize them into a new image by a certain permutation. Based on them, we construct three operation subsets, i.e. regular affine transformation set $\mathcal{O}_{RA}$, irregular affine transformation set $\mathcal{O}_{IA}$ and patch re-arranging set $\mathcal{O}_{PR}$ (detailed in Sec.1 in supplementary material). The final operation set is $\mathcal{O} = \mathcal{O}_{RA} \cup \mathcal{O}_{IA} \cup \mathcal{O}_{PR}$, and Fig. 1(a) shows SSD's entire workflow. To verify SSD's effectiveness, we extract the outputs of its penultimate layer as the learned representations, while the outputs of deep CAE's intermediate hidden layer (with the same dimension as SSD) are used for comparison. We feed them into isolation forest [40], which is generally acknowledged to be a good UOD method [69], to perform UOD under the same parameterization. As shown in Fig. 1(b), SSD's learned representations are able to outperform CAE by a large margin ($8\%$-$10\%$ AUROC).

### 3.2 Inlier Priority: The Foundation of End-to-end UOD

**Motivation.** The above simple solution feeds SSD's learned representations into a decoupled UOD method, which may yield suboptimal performance because SSD and the UOD method are trained separately [18, 20]. Our goal is to achieve end-to-end UOD without using a decoupled UOD method. Recall that outliers are essentially rare patterns in a data collection [7], which implies an intrinsic *class imbalance* between inliers/outliers. Class imbalance is unfavorable in machine learning as it leads to the bias towards majority class during training [70, 71]. However, we argue that class imbalance can be favorably exploited in UOD as it gives rise to "*inlier priority*": ***Despite that inliers/outliers are indiscriminately fed into SSD for training, SSD will prioritize the minimization of inliers' loss***. This intuition naturally inspires an end-to-end UOD solution by measuring how well the SSD's output of a datum matches its target pseudo label, which directly indicates its priority level in training and the likelihood to be an inlier. We demonstrate the inlier priority in terms of two aspects below:

**Priority by Gradient Magnitude.** *Our first point is that inliers will produce gradient with stronger magnitude to update the SSD network than outliers*. To demonstrate this point, we consider an SSD

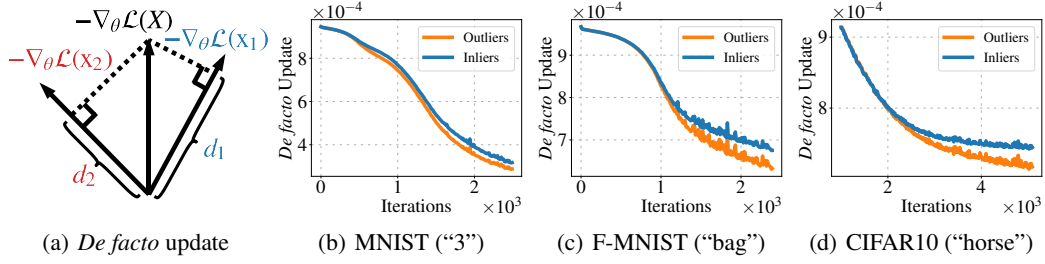

(a) *De facto* update     (b) MNIST ("3")     (c) F-MNIST ("bag")     (d) CIFAR10 ("horse")

Figure 3: An illustration of *de facto* update and some example cases of the average *de facto* update for inliers/outliers during the network training. The class used as inliers is in brackets.

with its network weights randomly initialized by i.i.d. uniform distribution on $[-1, 1]$. Without loss of generality, we consider the gradients w.r.t. the weights associated with the $c$-th class ($1 \leq c \leq K$) between the penultimate layer and softmax layer, $\mathbf{w}_c = [w_{s,c}]_{s=1}^{(L+1)}$ ($w_{L+1,c}$ is bias), because these weights are directly responsible for making predictions. For the commonly-used cross-entropy loss $\mathcal{L}$, only data transformed by the $c$-th operation $X^{(c)} = \{O(\mathbf{x}|c)|\mathbf{x} \in X\}$ are used to update $\mathbf{w}_c$. The gradient vector incurred by $\mathcal{L}$ is denoted by $\nabla_{\mathbf{w}_c}\mathcal{L} = [\nabla_{w_{s,c}}\mathcal{L}]_{s=1}^{(L+1)}$, which will be used to update $\mathbf{w}_c$ in back-propagation based optimizer like Stochastic Gradient Descent (SGD) [72]. Given unlabelled data with $N_{in}$ inliers and $N_{out}$ outliers, it is easy to know that $X^{(c)}$ also contains $N_{in}$ transformed inliers and $N_{out}$ transformed outliers. Here we are interested in the magnitude of transformed inliers and outliers' aggregated gradient to update $\mathbf{w}_c$, i.e. $||\nabla_{\mathbf{w}_c}^{(in)}\mathcal{L}||$ and $||\nabla_{\mathbf{w}_c}^{(out)}\mathcal{L}||$, which directly reflect inliers/outliers' strength to affect the training of SSD. Since SSD is randomly initialized, we need to compute the expectation of gradient magnitude. As shown in Sec. 2 of supplementary material, for a simplified SSD network with a single hidden-layer and sigmoid activation, we can quantitatively derive the following approximation on inliers and outliers' gradient magnitude:

$$\frac{E(||\nabla_{\mathbf{w}_c}^{(in)}\mathcal{L}||^2)}{E(||\nabla_{\mathbf{w}_c}^{(out)}\mathcal{L}||^2)} \approx \frac{N_{in}^2}{N_{out}^2} \tag{3}$$

where $E(\cdot)$ denotes the probability expectation. As the class imbalance between inliers and outliers leads to $N_{in} \gg N_{out}$, we naturally yield $E(||\nabla_{\mathbf{w}_c}^{(in)}\mathcal{L}||) \gg E(||\nabla_{\mathbf{w}_c}^{(out)}\mathcal{L}||)$. Therefore, it serves as a theoretical indication that *the gradient magnitude induced by inliers will be significantly larger than outliers for an untrained SSD network*. Since it is particularly difficult to directly analyze more complex network architectures such as Wide ResNet [68], we empirically examine inliers and outliers' gradient magnitude during training by experiments (see Fig. 2), and the observations on different benchmarks are consistent with the above analysis on the simplified case: The magnitude of inliers' aggregated gradient has constantly been larger than outliers during the process of SSD training.

**Priority by Network Updating Direction.** *Our second point is that the network updating direction of SSD will bias towards the direction that prioritizes reducing inliers' loss during the SSD training.* Since training is dynamic and a theoretical analysis is intractable, we demonstrate this point using an empirical verification by computing inliers/outliers' average "*de facto* update": As illustrated by Fig. 3(a), consider a datum $\mathbf{x}_i$ from a batch of data $X$, and its negative gradient $-\nabla_{\boldsymbol{\theta}}\mathcal{L}(\mathbf{x}_i)$ is the fastest network updating direction to reduce $\mathbf{x}_i$'s loss. However, the network weights $\boldsymbol{\theta}$ are actually updated by the negative gradient of the entire batch $X$, $-\nabla_{\boldsymbol{\theta}}\mathcal{L}(X) = -\frac{1}{N}\sum_i \nabla_{\boldsymbol{\theta}}\mathcal{L}(\mathbf{x}_i)$. It is actually different from the best updating direction for each individual datum. Thus, the *de facto* update $d_i$ for $\mathbf{x}_i$ refers to the actual gradient magnitude that $\mathbf{x}_i$ obtains along its best direction for loss reduction from the network update direction $-\nabla_{\boldsymbol{\theta}}\mathcal{L}(X)$, which can be computed by projecting $-\nabla_{\boldsymbol{\theta}}\mathcal{L}(X)$ onto the direction of $-\nabla_{\boldsymbol{\theta}}\mathcal{L}(\mathbf{x}_i)$: $d_i = -\nabla_{\boldsymbol{\theta}}\mathcal{L}(X) \cdot \frac{-\nabla_{\boldsymbol{\theta}}\mathcal{L}(\mathbf{x}_i)}{||-\nabla_{\boldsymbol{\theta}}\mathcal{L}(\mathbf{x}_i)||}$. In this way, $d_i$ reflects how much effort the network will devote to reduce $\mathbf{x}_i$'s loss, and it is a direct indicator of data's priority during network training. We calculate the average *de facto* update of inliers/outliers w.r.t the weights between SSD's penultimate and softmax layer and visualize some examples in Fig. 3(b)-3(d): Although the average *de facto* update of inliers/outliers is very close at the beginning, the average *de*

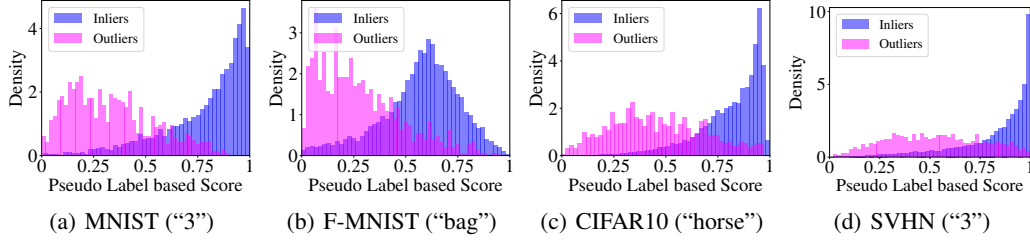

|  (a) MNIST ("3") | (b) F-MNIST ("bag") | (c) CIFAR10 ("horse") | (d) SVHN ("3") |

Figure 4: Normalized histograms of inliers/outliers' $S_{pl}(\mathbf{x})$. The class used as inliers is in brackets.

*facto* update of inliers becomes evidently higher than outliers as the training continues, which implies that *SSD will devote more efforts to reducing inliers' loss by its network updating direction.*

**Remarks on Inlier Priority. 1)** Based on the discussion above, inliers will gain priority in terms of both the gradient magnitude and the updating direction of SSD's network weights. Such priority leads to a lower loss for inliers after training, which enables us to discern outliers by SSD's outputs and serves as a foundation of end-to-end UOD. **2)** Intuitively, inlier priority will also happen when using AE/CAE based end-to-end UOD methods. However, the effect of inlier priority is severely diminished in this case for two reasons: First, AE/CAE typically uses the raw image pixels as learning targets, but the intra-class difference of inlier images can be very large, which means AE/CAE usually does not have a unified learning target like SSD. Second, AE/CAE is ineffective in learning high-level representations (as we discussed in Sec. 3.1), which makes it difficult to capture common high-level semantics of inlier images. Both factors above disable inliers from being a joint force to dominate the training of AE and produce a strong inlier priority effect like SSD, which is also demonstrated by AE/CAE's poor UOD performance in empirical evaluation (see experimental results in Sec. 4.2).

## 3.3 Scoring Strategies for UOD

Based on inlier priority, we need a strategy $S(\cdot)$ to score a datum $\mathbf{x}$. Given $\mathbf{x}^{(y)} = O(\mathbf{x}|y)$ and the probability vector $P(\mathbf{x}^{(y)}|\boldsymbol{\theta})$ from SSD's softmax layer, we explore three strategies below:

**Pseudo Label based Score (PL)**: Inlier priority suggests that SSD will prioritize reducing inliers' loss during training. For the datum $\mathbf{x}^{(y)}$, we note that the calculation of its cross entropy loss only depends on the probability $P^{(y)}(\mathbf{x}^{(y)}|\boldsymbol{\theta})$ that corresponds to its pseudo label $y$ in $P(\mathbf{x}^{(y)}|\boldsymbol{\theta})$. Thus, we propose a direct scoring strategy $S_{pl}(\mathbf{x})$ by averaging $P^{(y)}(\mathbf{x}^{(y)}|\boldsymbol{\theta})$ for all $K$ operations:

$$S_{pl}(\mathbf{x}) = \frac{1}{K}\sum_{y=1}^{K} P^{(y)}(\mathbf{x}^{(y)}|\boldsymbol{\theta}). \tag{4}$$

**Maximum Probability based Score (MP)**: PL seems to be an ideal score. However, we note that operations for surrogate supervision do not always create sufficiently separable classes, e.g. image with a digit "8" is still an "8" when applying a flip operation. Hence, misclassifications will happen and the probability $P^{(y)}(\mathbf{x}^{(y)}|\boldsymbol{\theta})$ that corresponds to pseudo label $y$ may not be the only or the best indicator to reflect how well the loss of a datum is reduced. Therefore, instead of $P^{(y)}(\mathbf{x}^{(y)}|\boldsymbol{\theta})$, we alternatively adopt the maximum probability of $P(\mathbf{x}^{(y)}|\boldsymbol{\theta})$ to calculate the score $S_{mp}(\mathbf{x})$ as follows:

$$S_{mp}(\mathbf{x}) = \frac{1}{K}\sum_{y=1}^{K} \max_{t} P^{(t)}(\mathbf{x}^{(y)}|\boldsymbol{\theta}). \tag{5}$$

**Negative Entropy based Score (NE)**. Both strategies above rely on a single probability retrieved from $P(\mathbf{x}^{(y)}|\boldsymbol{\theta})$, while the information of the rest $(K-1)$ classes' probability is ignored. If we consider the entire probability distribution $P(\mathbf{x}^{(y)}|\boldsymbol{\theta})$, the training actually encourages SSD to output a probability distribution closer to the label's one-hot distribution. With inlier priority, we can expect SSD to output a sharper probability distribution $P(\mathbf{x}^{(y)}|\boldsymbol{\theta})$ for inliers and a more uniform $P(\mathbf{x}^{(y)}|\boldsymbol{\theta})$

for outliers. Thus, we propose to use information entropy $H(\cdot)$ [73] as a simple and effective measure to the sharpness of a distribution, which gives the negative entropy based score $S_{ne}(\mathbf{x})$:

$$S_{ne}(\mathbf{x}) = -\frac{1}{K}\sum_{y=1}^{K}H(P(\mathbf{x}^{(y)}|\boldsymbol{\theta})) = \frac{1}{K}\sum_{y=1}^{K}\sum_{t=1}^{K}P^{(t)}(\mathbf{x}^{(y)}|\boldsymbol{\theta})\log(P^{(t)}(\mathbf{x}^{(y)}|\boldsymbol{\theta})). \quad (6)$$

A comparison of PL/MP/NE is given in Sec. 4.2. In Fig. 4(a)-4(d), we calculate the most intuitive $S_{pl}(\mathbf{x})$ of inliers/outliers on benchmarks and visualize the normalized histograms of $S_{pl}(\mathbf{x})$, which are favorably separable for UOD. Besides, such results also verify the effectiveness of inlier priority.

## 4 Experiments

### 4.1 Experiment Setup

**UOD Performance Evaluation on Image Benchmarks.** We follow the standard procedure from previous image UOD literature [13, 16, 17] to construct an image set with outliers: Given a standard image benchmark, all images from a class with one common semantic concept (e.g. "horse", "bag") are retrieved as inliers, while outliers are randomly sampled from the rest of classes by an outlier ratio $\rho$. We vary $\rho$ from 5% to 25% by a step of 5%. The assigned inlier/outlier labels are strictly unknown to UOD methods and only used for evaluation. Each class of a benchmark is used as inliers in turn and the performance on all classes is averaged as the overall UOD performance. The experiments are repeated for 5 times to report the average results. Five public benchmarks: MNIST [74], Fashion-MNIST (F-MNIST) [75], CIFAR10 [76], SVHN [77], CIFAR100 [76] are used for experiments[3]. Raw pixels are directly used as inputs with their intensity normalized into $[-1, 1]$. As for evaluation, we adopt the commonly-used Area under the Receiver Operating Characteristic curve (AUROC) and Area under the Precision-Recall curve (AUPR) as threshold-independent metrics [78].

**Implementation Details and Compared Methods**. For $E^3Outlier$, we use an $n = 10$ layer wide ResNet (WRN) with a widen factor $k = 4$ as the backbone DNN architecture. $K = 111$ operations are used for surrogate supervision, and NE is used as the scoring strategy. Since surrogate supervision augments original data by $K$ times, we train WRN for $\lceil\frac{250}{K}\rceil$ epochs. The batch size is 128. A learning rate 0.001 and a weight decay 0.0005 are adopted. The SGD optimizer with momentum 0.9 is used for MNIST and F-MNIST, while the Adam optimizer with $\boldsymbol{\beta} = (0.9, 0.999)$ is used for CIFAR10, CIFAR100 and SVHN for better convergence. We compare $E^3Outlier$ with the baselines and existing state-of-the-art DNN based UOD methods (reviewed in Sec. 2) below: **1)** CAE [79]. It directly uses CAE's reconstruction loss to perform UOD. **2)** CAE-IF. It feeds CAE's learned representations into isolation forest (IF) [40] as explained in Sec. 3.1. **3)** Discriminative reconstruction based autoencoder (DRAE) [16]. **4)** Robust deep autoencoder (RDAE) [17]. **5)** Deep autoencoding gaussian mixture model (DAGMM) [18]. **6)** SSD-IF. It shares $E^3Outlier$'s SSD part but feeds SSD's learned representations into IF to perform UOD. For all AE based UOD methods above, we adopt the same CAE architecture from [58] with a 4-layer encoder and 4-layer decoder. We do not use more complex CAE (e.g. CAE using skip connection [80] or more layers) since they usually lower outliers' reconstruction error as well and do not contribute to CAE's UOD performance. The hyperparameters of the compared methods are set to recommended values (if provided) or the values that produce the best performance. More implementation details are given in Sec. 1 of the supplementary material. Our codes and results can be verified at `https://github.com/demonzyj56/E3Outlier`.

### 4.2 UOD Performance Comparison and Discussion

**UOD Performance Comparison**. We report the numerical results on each benchmark under $\rho = 10\%$ and 20% in Table 1, and UOD performance by AUROC under $\rho$ from 5% to 25% is shown in Fig. 5(a)-Fig. 5(e) (full results are given in Sec. 4 of supplementary material). AUPR-in and AUPR-out in Table 1 denote the AUPR calculated when inliers and outliers are used as positive class respectively. We draw the following observations from those results: Above all, $E^3Outlier$ overwhelmingly outperforms existing DNN based UOD methods by a large margin. As Table 1 shows, $E^3Outlier$ usually improves AUROC/AUPR by 5% to 30% when compared with state-of-the-art UOD methods. In particular, $E^3Outlier$ produces a significant performance leap

Table 1: AUROC/AUPR-in/AUPR-out (%) for UOD methods. The best performance is in bold.

| Dataset | $\rho$ | CAE | CAE-IF | DRAE | RDAE | DAGMM | SSD-IF | $E^3Outlier$ |
|---|---|---|---|---|---|---|---|---|
| MNIST | 10% | 68.0/92.0/32.9 | 85.5/97.8/49.0 | 66.9/93.0/30.5 | 71.8/93.1/35.8 | 64.0/92.9/26.6 | 93.8/99.2/**68.7** | **94.1/99.3**/67.5 |
| | 20% | 64.0/82.7/40.7 | 81.5/93.6/57.2 | 67.2/86.6/42.5 | 67.0/84.2/43.2 | 65.9/86.4/41.3 | 90.5/97.3/71.0 | **91.3/97.6/72.3** |
| F-MNIST | 10% | 70.3/94.3/29.3 | 82.3/97.2/40.3 | 67.1/93.9/25.5 | 75.3/95.8/31.7 | 64.0/92.7/30.3 | 90.6/98.5/68.6 | **93.3/99.0/75.9** |
| | 20% | 64.4/85.3/36.8 | 77.8/92.2/49.0 | 65.7/86.9/36.6 | 70.9/89.2/41.4 | 66.0/86.7/43.5 | 87.6/95.6/71.4 | **91.2/97.1/78.9** |
| CIFAR10 | 10% | 55.9/91.0/14.4 | 54.1/90.2/13.7 | 56.0/90.7/14.7 | 55.4/90.7/14.0 | 56.1/91.3/15.6 | 64.0/93.5/18.3 | **83.5/97.5/43.4** |
| | 20% | 54.7/81.6/25.5 | 53.8/80.7/25.3 | 55.6/81.7/26.8 | 54.2/81.0/25.7 | 54.7/81.8/26.3 | 60.2/85.0/28.3 | **79.3/93.1/52.7** |
| SVHN | 10% | 51.2/90.3/10.6 | 55.0/91.4/11.9 | 51.0/90.3/10.5 | 52.1/90.6/10.8 | 50.0/90.0/19.3 | 73.4/95.9/22.0 | **86.0/98.0/36.7** |
| | 20% | 50.7/80.2/20.7 | 54.0/82.0/22.4 | 50.6/80.4/20.5 | 51.8/80.9/21.1 | 50.0/79.9/29.6 | 69.2/89.5/33.7 | **81.0/93.4/47.0** |
| CIFAR100 | 10% | 55.2/91.0/14.5 | 54.5/90.7/13.8 | 55.6/90.9/15.0 | 55.8/90.9/15.0 | 54.9/91.1/14.2 | 55.6/91.5/13.0 | **79.2/96.8/33.3** |
| | 20% | 54.4/81.7/25.6 | 53.5/80.9/25.1 | 55.5/81.8/27.0 | 54.9/81.5/26.5 | 53.8/81.5/24.7 | 54.3/82.1/23.4 | **77.0/92.4/46.5** |

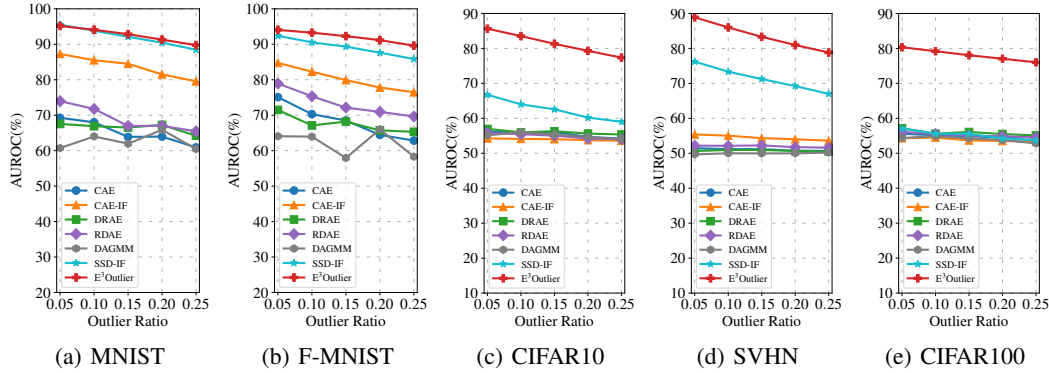

Figure 5: UOD performance (AUROC) comparison with varying $\rho$ from 5% to 25%.

($\geq 20\%$ AUROC gain) on CIFAR10, SVHN and CIFAR100, which have constantly been difficult benchmarks for UOD. Next, end-to-end $E^3Outlier$ almost consistently outperforms its decoupled counterpart SSD-IF. Although SSD-IF performs closely to $E^3Outlier$ in simple cases, $E^3Outlier$ evidently prevails over SSD-IF on CIFAR10/SVHN/CIFAR100 by 11% to 24% AUROC gain. By contrast, the decoupled CAE-IF/RDAE get better UOD performance than their end-to-end counterparts CAE/DRAE/DAGMM on MNIST/F-MNIST, and all of them yield inferior performance on CIFAR10/SVHN/CIFAR100. Hence, observations above have justified $E^3Outlier$ as a highly effective and end-to-end UOD solution. In addition, we would like to make two remarks: **1)** We must point out that the data augmentation effect (surrogate supervision will augment the training data by $K$ times) is not the reason why $E^3Outlier$ outperforms existing methods by a large margin. Experiments show that when we train CAE with the same training data with $E^3Outlier$, the performance typically becomes worse than original CAE (e.g. 55.5%/63.9%/54.2%/50.0%/53.8% AUROC on MNIST/F-MNIST/CIFAR10/SVHN/CIFAR100 when $\rho = 10\%$). By contrast, $E^3Outlier$ can effectively exploit the high-level discriminative label information from data of pseudo classes, which is fundamentally different from generative models like AE/CAE. **2)** To fairly compare the quality of learned representation for CAE and SSD, CAE's hidden layer by default shares SSD's penultimate layer dimension, which is fixed to 256 by Wide-ResNet architecture. A different latent dimension may influence CAE's performance, but it cannot enable CAE to perform comparably to $E^3Outlier$, especially on difficult datasets like CIFAR10. We also test other values for CAE's latent dimensions, and experimental results show that even for a carefully selected latent dimension (e.g. 64) that performs best on most benchmarks, it brings minimal gain to CAE's performance on difficult datasets CIFAR10/CIFAR100 (e.g. 56.3%/56.1% AUROC when $\rho = 10\%$), and on simpler datasets (MNIST/F-MNIST/SVHN) CAE's performance (71.9%/75.6%/53.4%, $\rho = 10\%$) is still far behind $E^3Outlier$ (94.1%/93.3%/86.0%) despite some limited improvement. More importantly, a prior choice of the optimal latent dimension or CAE architecture for UOD is difficult in itself.

**Discussion**. We discuss five factors that are related to our $E^3Outlier$ framework's performance by experiments. Since the trends under different values of $\rho$ are fairly similar, we visualize the results when using $\rho = 10\%$: **1)** Operation set for surrogate supervision (see Fig. 6(a)): We test the UOD

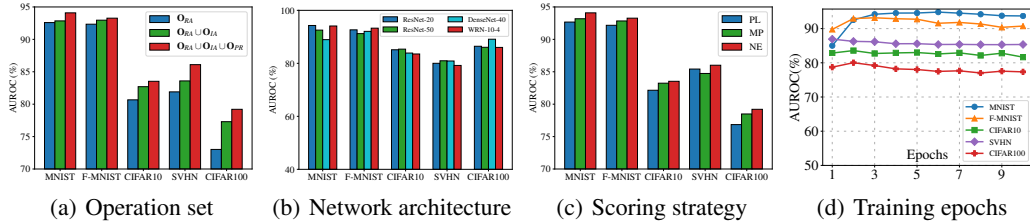

| (a) Operation set | (b) Network architecture | (c) Scoring strategy | (d) Training epochs |

Figure 6: Different factors' influence on $E^3 Outlier$'s performance under $\rho = 10\%$.

performance with different combinations of operation subsets to be $\mathcal{O}$. The results suggest that $\mathcal{O}_{RA}$ alone already works satisfactorily, but a union of $\mathcal{O}_{RA}$, $\mathcal{O}_{IA}$ and $\mathcal{O}_{PR}$ produces the best performance, which reflects the extendibility of operation sets. **2)** Network architecture (see Fig. 6(b)): In addition to WRN, we explore ResNet-20/ResNet-50 [19] and DenseNet-40 [81] for SSD with other settings fixed. The results show that those architectures basically achieve satisfactory UOD performance with minor differences, which verifies the applicability of different network architectures. In particular, we note that a more complex architecture (ResNet-50/DenseNet-40) improves the UOD performance on relatively complex datasets (CIFAR10, SVHN and CIFAR100), but its performance is inferior on simple datasets. **3)** Scoring strategy (see Fig. 6(c)): Among three scoring strategies (PL/MP/NE) proposed in Sec. 3.3, NE constantly yields the best performance by up to 2.3% AUC gain compared with PL/MP, while MP also outperforms the naive PL. Thus, we use the NE by default for $E^3 Outlier$. **4)** Training epochs (see Fig. 6(d)): We measure the UOD performance when the SSD is trained by 1 to 10 epochs respectively. In general, the UOD performance is improved at the initial stage of training (less than 3 training epochs) and then stabilizes as the training epochs continue to increase. **5)** Outlier ratio: First, we note that sometimes the ratio of outliers can be very small (e.g. $\leq 1\%$), so we also test $E^3 Outlier$'s performance in such case. The experiments show that $E^3 Outlier$ still achieves satisfactory performance: For example, when $\rho = 0.5\%$, $E^3 Outlier$ achieves 96.0%/93.6%/87.4%/91.0%/80.7% AUROC for MNIST/F-MNIST/CIFAR10/SVHN/CIFAR100 respectively, which is even better than the case with a higher outlier ratio. We also notice that the performance of $E^3 Outlier$ tends to drop as the outlier ratio $\rho$ increases. This is reasonable in the setting of UOD because the "outlierness" of outliers will decrease as their number increases, i.e. they are less likely to be viewed as "outliers" under the unsupervised setting as they gradually play a more important role in constituting the original unlabelled data.

## 5    Conclusion

In this paper, we propose a framework named $E^3 Outlier$ to achieve effective and end-to-end UOD from raw image data. $E^3 Outlier$ exploits surrogate supervision rather than traditional AE/CAE for representation learning in UOD, while a new property named inlier priority is demonstrated theoretically and empirically as the foundation of end-to-end UOD. By inlier priority and the negative entropy based score, $E^3 Outlier$ achieves significant UOD performance leap when compared with state-of-the-art DNN based UOD methods. For future research, it is interesting to explore a quantitative measure of each operation's effectiveness for surrogate supervision and develop effective late fusion strategies of different operations for scoring. As an open framework, different network architectures, surrogate supervision operations and scoring strategies can also be explored for $E^3 Outlier$.

## Acknowledgement

This work is supported by National Key R&D Program of China 2018YFB1003203 and National Natural Science Foundation of China (NSFC) under Grant No. 61773392, 61672528. This work is also supported by the German Research Foundation (DFG) award KL 2698/2-1 and by the German Federal Ministry of Education and Research (BMBF) awards 031L0023A, 01IS18051A, and 031B0770E. Xinwang Liu, En Zhu and Jianping Yin are corresponding authors of this paper.

## Footnotes

*Authors contribute equally.

[2]All UOD experiments in Sec. 3 follow the setup detailed in Sec. 4.1 and the outlier ratio is fixed to 10%.

[3]As all images are viewed as unlabelled in UOD, we do not split train/test set. CIFAR100 uses 20 superclasses.

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
