[Supplementary Material]

# Supplementary Material for "Effective End-to-end Unsupervised Outlier Detection via Inlier Priority of Discriminative Network"

## 1 Operation Set $\mathcal{O}$ for Surrogate Supervision

For $\mathcal{O}$, each operation $O(\cdot|y) \in \mathcal{O}$ is defined as a combination of one or more basic transformations from transformation sets below: **1)** *Rotation*, which includes simple rotation transformations that clock-wisely rotates images by integer times of $90°$: $\mathcal{T}_{SR} = \{Rot(\cdot, (y-1)\cdot 90°)\}_{y=1}^4$, and irregular rotation transformations by integer times of $30°$ (transformations already in $\mathcal{T}_{SR}$ are excluded): $\mathcal{T}_{IR} = \{Rot(\cdot, (y-1)\cdot 30°)\}_{y=1}^{12} - \mathcal{T}_{SR}$. **2)** *Flip*: $\mathcal{T}_F = \{Flip(\cdot, y)\}_{y=0}^1$, where $y = 1/0$ refers to flipping the image or not. **3)** *Shifting*, which includes x-axis shifting: $\mathcal{T}_{Sx} = \{S_x(\cdot, (y-2)\cdot D)\}_{y=1}^3$ and y-axis shifting: $\mathcal{T}_{Sy} = \{S_y(\cdot, (y-2)\cdot D)\}_{y=1}^3$ ($D$ is the step of shifting). **4)** *Patch re-arranging*, which partitions the image into $M$ equally-sized patches and re-organizes them into a new image by a permutation selected from $M!$ possible permutations: $\mathcal{T}_{PR} = \{PR(\cdot, perm_y)\}_{y=1}^{M!}$. Then we join the transformation sets into three operation subsets, regular affine operation set $\mathcal{O}_{RA}$, irregular affine operation set $\mathcal{O}_{IA}$ and patch re-arranging operation set $\mathcal{O}_{PR}$:

$$\mathcal{O}_{RA} = \mathcal{T}_{SR} \times \mathcal{T}_F \times (\mathcal{T}_{Sx} \times \mathcal{T}_{Sy}), \qquad \mathcal{O}_{IA} = \mathcal{T}_{IR} \times \mathcal{T}_F, \qquad \mathcal{O}_{PR} = \mathcal{T}_{PR} \tag{1}$$

where "$\times$" refers to Cartesian product. Therefore, we obtain $4 \times 2 \times (3 \times 3) = 72$ operations for $\mathcal{O}_{RA}$, $(12-4) \times 2 = 16$ operations for $\mathcal{O}_{IA}$ and $4! = 24$ operations for $\mathcal{O}_{PR}$, so the final $\mathcal{O}$ has $72 + 16 + 24 - 1 = 111$ operations in total (The permutation that corresponds to the original image in $\mathcal{O}_{PR}$ is excluded). In our experiments, we set $D = 8$ pixels and $M = 4$. There are several remarks: **1)** The reason why we construct operation sets as (1) is to augment the number of operations without joining two transformations that both produce image artifact (e.g. shifting and irregular rotation), which will degrade the UOD performance. **2)** It is NOT compulsory to apply all operations in $\mathcal{O}$ to provide surrogate supervision. In fact, our experiments show that $\mathcal{O}_{RA}$ can produce good UOD performance as well, but a union of $\mathcal{O}_{RA}$, $\mathcal{O}_{IA}$ and $\mathcal{O}_{PR}$ will produce the best results. **3)** Other operations like colorization [1] or temporal shuffle [2] can be incorporated into the surrogate supervision for better UOD performance or deal with other data types, such as videos or optical flow.

## 2 Theoretical Derivation on Priority by Gradient Magnitude

To obtain (3) in the original manuscript, we consider an SSD with its network weights randomly initialized by i.i.d. uniform distribution on $[-1, 1]$. Suppose that the network of SSD has an $(L+1)$-node penultimate layer and a final $K$-node softmax layer. We discuss the case of inliers $X_{in}$ first: For cross-entropy loss $\mathcal{L}$, only transformed inliers generated by the $c$-th operation $X_{in}^{(c)} = \{\mathbf{x}^{(c)} | \mathbf{x} \in X_{in}\}$ are used to update $\mathbf{w}_c$. The gradient vector incurred by $X_{in}^{(c)}$ is denoted by $\nabla_{\mathbf{w}_c}^{(in)}\mathcal{L} = [\nabla_{w_{s,c}}\mathcal{L}]_{s=1}^{(L+1)}$ with its element $\nabla_{w_{s,c}}\mathcal{L}$ given by:

$$\nabla_{w_{s,c}}\mathcal{L} = \sum_{i=1}^{N_{in}} \nabla_{w_{s,c}}\mathcal{L}(\mathbf{x}_i) = \sum_{i=1}^{N_{in}} (P^{(c)}(\mathbf{x}_i) - 1)h^{(s)}(\mathbf{x}_i) \tag{2}$$

where $N_{in}$ is the inlier number ($N_{out}$ is the outlier number), $P^{(c)}(\mathbf{x}_i)$ is the $c$-th node's output of the softmax layer and $h^{(s)}(\mathbf{x}_i)$ is the $s$-th node's output of the penultimate layer for $\mathbf{x}_i \in X_{in}^{(c)}$. Since SSD is randomly initialized, we compute the expectation of inliers' gradient magnitude to update $\mathbf{w}_c$, i.e. $E(||\nabla_{\mathbf{w}_c}^{(in)}\mathcal{L}||_2^2)$. As $||\nabla_{\mathbf{w}_c}^{(in)}\mathcal{L}||_2^2 = \sum_{s=1}^{L+1}(\nabla_{w_{s,c}}\mathcal{L})^2$, it needs to compute the term below:

$$E((\nabla_{w_{s,c}}\mathcal{L})^2) = E((\sum_{i=1}^{N_{in}}\nabla_{w_{s,c}}\mathcal{L}(\mathbf{x}_i))^2) = \sum_{i=1}^{N_{in}}\sum_{j=1}^{N_{in}}E(\nabla_{w_{s,c}}\mathcal{L}(\mathbf{x}_i)\nabla_{w_{s,c}}\mathcal{L}(\mathbf{x}_j)). \quad (3)$$

To compute (3), we first define a function $g_{ij}^{(s,c)}$ as follows:

$$g_{ij}^{(s,c)} = \nabla_{w_{s,c}}\mathcal{L}(\mathbf{x}_i)\nabla_{w_{s,c}}\mathcal{L}(\mathbf{x}_j) = (P^{(c)}(\mathbf{x}_i) - 1)(P^{(c)}(\mathbf{x}_j) - 1)h^{(s)}(\mathbf{x}_i)h^{(s)}(\mathbf{x}_j) \quad (4)$$

where $h^{(s)}(\mathbf{x}_i)$ is the penultimate layer's $s$-th node's output and $P^{(c)}(\mathbf{x}_i)$ denotes the softmax layer's $c$-th node's output for $\mathbf{x}_i$. Our goal is to compute $E(g_{ij}^{(s,c)})$ w.r.t the weights between the penultimate layer and the final softmax layer, which is a $(L+1) \times K$ vector $\mathbf{w} = [\mathbf{w}_c]_{c=1}^{K}$, with the weights associated with the $c$-th class ($1 \leq c \leq K$) to be a $(L+1)$ column vector $\mathbf{w}_c = [w_{s,c}]_{s=1}^{L+1}$. To simplify computation, we use the second-order Taylor series expansion of $g_{ij}^{(s,c)}$:

$$g_{ij}^{(s,c)}(\mathbf{w}) \approx g_{ij}^{(s,c)}(\boldsymbol{\mu}) + \nabla_{\mathbf{w}}g_{ij}^{(s,c)}(\boldsymbol{\mu}) \cdot (\mathbf{w} - \boldsymbol{\mu}) + \frac{1}{2}(\mathbf{w} - \boldsymbol{\mu})^T \cdot \nabla_{\mathbf{w}}^2 g_{ij}^{(s,c)}(\boldsymbol{\mu}) \cdot (\mathbf{w} - \boldsymbol{\mu}) \quad (5)$$

where $\boldsymbol{\mu}$ is the expectation of $\mathbf{w}$. Since each weight in $\mathbf{w}$ is drawn from i.i.d uniform distribution on $[-1, 1]$, we have $\mu_{s,c} = E(w_{s,c}) = 0$, $E(w_{s,c}^2) = \frac{1}{3}$ and $E(w_{s,c}w_{t,c}) = 0$ ($s \neq t$). Therefore, the expectation of $g_{ij}^{(s,c)}$ w.r.t. $\mathbf{w}$ is approximated as

$$E(g_{ij}^{(s,c)}(\mathbf{w})) \approx g_{ij}^{(s,c)}(\mathbf{0}) + \frac{1}{2}\sum_{t=1}^{L+1}\sum_{l=1}^{K}\nabla_{w_{t,l}}^2 g_{ij}^{(s,c)}(\mathbf{0})E(w_{s,c}^2) = g_{ij}^{(s,c)}(\mathbf{0}) + \frac{1}{6}\sum_{t=1}^{L+1}\sum_{l=1}^{K}\nabla_{w_{t,l}}^2 g_{ij}^{(s,c)}(\mathbf{0}). \quad (6)$$

Thus, computing $E(g_{ij}^{(s,c)}(\mathbf{w}))$ requires the computation of $\nabla_{w_{t,l}}^2 g_{ij}^{(s,c)}(\mathbf{0})$. Recall the softmax probability is computed by:

$$P^{(c)}(\mathbf{x}_i) = \frac{e^{\mathbf{h}^\top(\mathbf{x}_i)\cdot\mathbf{w}_c}}{\sum_{l=1}^{K}e^{\mathbf{h}^\top(\mathbf{x}_i)\cdot\mathbf{w}_l}} \quad (7)$$

where $\mathbf{h}(\mathbf{x}_i) = [h^{(s)}(\mathbf{x}_i)]_{s=1}^{L+1}$ is penultimate layer's output for $\mathbf{x}_i$. Since $\mathbf{h}(\mathbf{x}_i)$ is independent of $\mathbf{w}$, we have:

$$\nabla_{w_{t,l}}P^{(c)}(\mathbf{x}_i) = -P^{(c)}(\mathbf{x}_i)(\delta_{c,l} - P^{(l)}(\mathbf{x}_i)) \cdot h^{(t)}(\mathbf{x}_i) \quad (8)$$

where $\delta_{c,l} = 1$ if $c = l$ and $\delta_{c,l} = 0$ otherwise. Using (4), (7) and (8), we can calculate $\nabla_{w_{t,l}}^2 g_{ij}^{(s,c)}$ by:

$$\begin{aligned}
\nabla_{w_{t,l}}^2 g_{ij}^{(s,c)} = h^{(s)}(\mathbf{x}_i)h^{(s)}(\mathbf{x}_j)\Big[&-(h^{(t)}(\mathbf{x}_i))^2 \cdot P^{(c)}(\mathbf{x}_i)(\delta_{c,l} - P^{(l)}(\mathbf{x}_i))^2(1 - P^{(c)}(\mathbf{x}_j)) \\
&+(h^{(t)}(\mathbf{x}_i))^2 \cdot P^{(c)}(\mathbf{x}_i)P^{(l)}(\mathbf{x}_i)(1 - P^{(l)}(\mathbf{x}_i))(1 - P^{(c)}(\mathbf{x}_j)) \\
&+2 \cdot h^{(t)}(\mathbf{x}_i)h^{(t)}(\mathbf{x}_j) \cdot P^{(c)}(\mathbf{x}_i)P^{(c)}(\mathbf{x}_j)(\delta_{c,l} - P^{(l)}(\mathbf{x}_i))(\delta_{c,l} - P^{(l)}(\mathbf{x}_j)) \\
&-(h^{(t)}(\mathbf{x}_j))^2 \cdot P^{(c)}(\mathbf{x}_j)(\delta_{c,l} - P^{(l)}(\mathbf{x}_j))^2(1 - P^{(c)}(\mathbf{x}_i)) \\
&+(h^{(t)}(\mathbf{x}_j))^2 \cdot P^{(c)}(\mathbf{x}_j)P^{(l)}(\mathbf{x}_j)(1 - P^{(l)}(\mathbf{x}_j))(1 - P^{(c)}(\mathbf{x}_i))\Big].
\end{aligned} \quad (9)$$

Therefore, in the summation term of (6), we have $(L+1)$ terms that satisfy $c = l$, and in this case $\nabla^2_{w_{t,l}} g^{(s,c)}_{ij}|_{\mathbf{w}=\mathbf{0}}$ is:

$$h^{(s)}(\mathbf{x}_i)h^{(s)}(\mathbf{x}_j)\Big[(h^{(t)}(\mathbf{x}_i))^2\frac{(K-1)^2(2-K)}{K^4} + (h^{(t)}(\mathbf{x}_j))^2\frac{(K-1)^2(2-K)}{K^4} + 2h^{(t)}(\mathbf{x}_i)h^{(t)}(\mathbf{x}_j)\frac{(K-1)^2}{K^4}\Big].$$
(10)

For the rest $(L+1)(K-1)$ terms in the summation term that satisfy $c \neq l$, $\nabla^2_{w_{t,l}} g^{(s,c)}_{ij}|_{\mathbf{w}=\mathbf{0}}$ is:

$$h^{(s)}(\mathbf{x}_i)h^{(s)}(\mathbf{x}_j)\Big[(h^{(t)}(\mathbf{x}_i))^2\frac{(K-1)(K-2)}{K^4} + (h^{(t)}(\mathbf{x}_j))^2\frac{(K-1)(K-2)}{K^4} + 2h^{(t)}(\mathbf{x}_i)h^{(t)}(\mathbf{x}_j)\frac{1}{K^4}\Big].$$
(11)

By substituting (10) and (11) into (6), we can obtain the result of (5) in the original manuscript:

$$E(\nabla_{w_{s,c}}\mathcal{L}(\mathbf{x}_i)\nabla_{w_{s,c}}\mathcal{L}(\mathbf{x}_j)) = E(g^{(s,c)}_{ij}(\mathbf{w})) \approx h^{(s)}(\mathbf{x}_i)h^{(s)}(\mathbf{x}_j)\Big[\frac{(K-1)^2}{K^2} + \frac{(K-1)}{3K^3}\sum_{t=1}^{L+1} h^{(t)}(\mathbf{x}_i)h^{(t)}(\mathbf{x}_j)\Big].$$
(12)

It remains to calculate the expectation of $h^{(t)}(\mathbf{x}_i)h^{(t)}(\mathbf{x}_j)$ in (12). To make its calculation tractable, we consider a simplified case of a network with a single hidden-layer and sigmoid activation. In this case, by [3, Lemma 3.b], the expectation of $h^{(s)}(\mathbf{x}_i)h^{(s)}(\mathbf{x}_j)$ w.r.t. the randomly initialized weights between the input and hidden layer satisfies $E(h^{(s)}(\mathbf{x}_i)h^{(s)}(\mathbf{x}_j)) \approx \frac{1}{4}$ and $E(h^{(s)}(\mathbf{x}_i)^2 h^{(s)}(\mathbf{x}_j)^2) \approx \frac{1}{16}$. Thus, by definition of $||\nabla^{(in)}_{\mathbf{w}_c}\mathcal{L}||_2^2$ and (3), we yield:

$$E(||\nabla^{(in)}_{\mathbf{w}_c}\mathcal{L}||_2^2) \approx N_{in}^2\Big[(L+1)(\frac{(K-1)^2}{4K^2} + \frac{(K-1)(L+1)}{48K^3})\Big] \triangleq N_{in}^2 \cdot A.$$
(13)

Since $L, K, A$ are constant, (13) suggests that the magnitude of inliers' gradient $E(||\nabla^{(in)}_{\mathbf{w}_c}\mathcal{L}||_2^2)$ is proportional to $N_{in}^2$. Similarly, outliers' gradient magnitude $E(||\nabla^{(out)}_{\mathbf{w}_c}\mathcal{L}||_2^2) \approx N_{out}^2 \cdot A$. Thus, it is easy to obtain:

$$\frac{E(||\nabla^{(in)}_{\mathbf{w}_c}\mathcal{L}||^2)}{E(||\nabla^{(out)}_{\mathbf{w}_c}\mathcal{L}||^2)} \approx \frac{N_{in}^2}{N_{out}^2}$$
(14)

## 3  Implementation Details of Compared Methods

For all AE based methods, we adopt a deep CAE with the architecture below: $conv(k = 3, s = 2) - bn - Relu - conv(k = 3, s = 2) - bn - relu - conv(k = 3, s = 2) - bn - relu - reshape - fc(4096, 256) - tanh - fc(256, 4096) - bn - relu - reshape - deconv(k = 3, s = 2) - bn - relu - deconv(k = 3, s = 2) - bn - relu - deconv(k = 3, s = 2) - tanh$, while $k$ and $s$ refer to kernel size and stride. For each individual method, the parameters are set as follows: **1)** CAE [4]. CAE is trained by Mean Square Error Loss (MSE) and its reconstruction loss is directly used to perform UOD. The CAE is trained by default Adam optimizer in PyTorch[1] for 250 epochs with learning rate 0.001 and weight decay 0.0005. The batch size is 128. **2)** CAE-IF. CAE-IF is a decoupled/hybrid method that feeds the learned representations of CAE into isolation forest (IF) [5]. The training of CAE is the illustrated above, and the IF is realized by scikit-learn framework[2]. The contamination parameter of IF is set by $p = \rho$ to yield better UOD performance for comparison with $E^3$*Outlier*, and other parameters are set to default values in scikit-learn. **3)** Discriminative reconstruction based autoencoder (DRAE) [6]. We set DRAE's encouraging term weight $\lambda = 0.1$ as recommended in [6], while other training setting is the same as CAE. **4)** Robust deep autoencoder (RDAE) [7]. We set $\lambda = 0.00065$ for RDAE's regularization, which performs best in the empirical evaluation of [7]. To yield the best performance, we use 20 outer epochs and 1 inner epochs for the

alternating optimization. **5)** Deep autoencoding gaussian mixture model (DAGMM) [8]. As suggested by [8], we adopt $\lambda_1 = 0.1$, $\lambda_2 = 0.005$ for the energy regularization term and the singularity penalty term respectively. An Adam optimizer with the recommended learning rate 0.0001 is used to optimize the CAE and density estimation network for 200 epochs. The batch size is 1024 as set in [8]. **6)** SSD-IF. SSD-IF shares $E^3Outlier$'s SSD part but directly feeds SSD's learned representations into IF to perform UOD, which is a comparison to $E^3Outlier$'s end-to-end UOD. SSD-IF shares exactly the same IF setting with CAE-IF. The experiments are run on a PC with dual NVIDIA Titan Xp GPU, 64 GiB RAM and Intel 7820X CPU, under a programming environment with Python 3.6, PyTorch 0.4.1 and Keras 2.2.0.

## 4 Detailed Results of UOD Performance

We present detailed results of UOD performance comparison on each class of benchmarks in Table 1-15. The AUROC results are given in Table 1-5, while the AUPR results are given in Table 6-15. Note that only NE based performance is shown for $E^3Outlier$ due to limit of space. In each table, the best overall UOD performance ("*average*") on each benchmark is shown in bold.

## Footnotes

[1]https://pytorch.org/

[2]https://scikit-learn.org/

## References

[1] Richard Zhang, Phillip Isola, and Alexei A Efros. Colorful image colorization. In *European conference on computer vision*, pages 649–666. Springer, 2016.

[2] Ishan Misra, C Lawrence Zitnick, and Martial Hebert. Shuffle and learn: unsupervised learning using temporal order verification. In *European Conference on Computer Vision*, pages 527–544. Springer, 2016.

[3] Rangachari Anand, Kishan G Mehrotra, Chilukuri K Mohan, and Sanjay Ranka. An improved algorithm for neural network classification of imbalanced training sets. *IEEE Transactions on Neural Networks*, 4(6):962–969, 1993.

[4] Jonathan Masci, Ueli Meier, Dan Cireşan, and Jürgen Schmidhuber. Stacked convolutional auto-encoders for hierarchical feature extraction. In *International Conference on Artificial Neural Networks*, pages 52–59. Springer, 2011.

[5] Fei Tony Liu, Kai Ming Ting, and Zhi-Hua Zhou. Isolation forest. In *2008 Eighth IEEE International Conference on Data Mining*, pages 413–422. IEEE, 2008.

[6] Yan Xia, Xudong Cao, Fang Wen, Gang Hua, and Jian Sun. Learning discriminative reconstructions for unsupervised outlier removal. In *Proceedings of the IEEE International Conference on Computer Vision (ICCV)*, pages 1511–1519, 2015.

[7] Chong Zhou and Randy C Paffenroth. Anomaly detection with robust deep autoencoders. In *Proceedings of the 23rd ACM SIGKDD International Conference on Knowledge Discovery and Data Mining*, pages 665–674. ACM, 2017.

[8] Bo Zong, Qi Song, Martin Renqiang Min, Wei Cheng, Cristian Lumezanu, Daeki Cho, and Haifeng Chen. Deep autoencoding gaussian mixture model for unsupervised anomaly detection. In *International Conference on Learning Representations (ICLR)*, 2018.

Table 1: AUROC (%) when $\rho = 0.05$

| Dataset | Class name | CAE | CAE-IF | DRAE | RDAE | DAGMM | SSD-IF | $E^3Outlier$ |
|---|---|---|---|---|---|---|---|---|
| MNIST | 0 | $57.43 \pm 3.06$ | $92.78 \pm 0.93$ | $62.39 \pm 3.62$ | $69.63 \pm 2.39$ | $71.30 \pm 21.72$ | $92.54 \pm 0.99$ | $96.25 \pm 0.28$ |
| | 1 | $99.31 \pm 0.09$ | $99.39 \pm 0.06$ | $83.05 \pm 11.22$ | $99.36 \pm 0.08$ | $87.19 \pm 12.48$ | $97.69 \pm 0.21$ | $96.99 \pm 0.55$ |
| | 2 | $59.89 \pm 5.24$ | $77.84 \pm 2.17$ | $69.23 \pm 9.98$ | $62.82 \pm 3.49$ | $50.32 \pm 11.76$ | $94.68 \pm 0.49$ | $93.86 \pm 0.44$ |
| | 3 | $63.72 \pm 1.96$ | $85.18 \pm 1.08$ | $63.11 \pm 7.12$ | $68.33 \pm 2.49$ | $47.45 \pm 4.02$ | $97.59 \pm 0.50$ | $97.72 \pm 0.22$ |
| | 4 | $71.30 \pm 2.10$ | $85.94 \pm 2.85$ | $70.51 \pm 6.28$ | $74.74 \pm 1.52$ | $62.07 \pm 4.94$ | $96.47 \pm 0.59$ | $96.62 \pm 0.35$ |
| | 5 | $61.38 \pm 3.40$ | $81.62 \pm 2.11$ | $57.36 \pm 4.21$ | $67.80 \pm 1.71$ | $53.07 \pm 3.56$ | $96.22 \pm 0.99$ | $93.42 \pm 0.58$ |
| | 6 | $71.89 \pm 4.76$ | $90.45 \pm 1.43$ | $66.68 \pm 9.13$ | $77.43 \pm 1.87$ | $52.15 \pm 12.88$ | $98.50 \pm 0.35$ | $97.36 \pm 0.18$ |
| | 7 | $82.70 \pm 2.55$ | $93.60 \pm 0.58$ | $79.31 \pm 3.45$ | $88.57 \pm 1.44$ | $71.52 \pm 8.51$ | $95.45 \pm 0.54$ | $94.54 \pm 0.45$ |
| | 8 | $47.98 \pm 6.95$ | $74.71 \pm 1.49$ | $49.91 \pm 5.14$ | $50.91 \pm 4.25$ | $52.67 \pm 6.14$ | $89.52 \pm 2.44$ | $89.43 \pm 0.81$ |
| | 9 | $77.08 \pm 2.13$ | $91.21 \pm 0.78$ | $74.00 \pm 7.22$ | $79.99 \pm 2.47$ | $59.45 \pm 7.92$ | $96.44 \pm 0.55$ | $95.42 \pm 0.14$ |
| | average | $69.27 \pm 1.02$ | $87.27 \pm 0.34$ | $67.56 \pm 2.84$ | $73.96 \pm 0.83$ | $60.72 \pm 2.09$ | $\mathbf{95.51 \pm 0.30}$ | $95.16 \pm 0.15$ |
| Fashion-MNIST | t-shirt | $65.88 \pm 5.02$ | $87.28 \pm 0.79$ | $66.83 \pm 6.48$ | $77.48 \pm 1.10$ | $64.53 \pm 15.40$ | $90.77 \pm 1.31$ | $94.00 \pm 0.39$ |
| | trouser | $97.02 \pm 0.21$ | $96.76 \pm 0.52$ | $88.31 \pm 4.18$ | $97.07 \pm 0.27$ | $69.94 \pm 18.29$ | $98.42 \pm 0.10$ | $98.73 \pm 0.18$ |
| | pullover | $65.89 \pm 5.19$ | $84.24 \pm 1.21$ | $66.56 \pm 3.36$ | $73.83 \pm 4.12$ | $50.17 \pm 10.66$ | $90.00 \pm 1.11$ | $91.59 \pm 1.41$ |
| | dress | $79.74 \pm 1.96$ | $87.97 \pm 1.84$ | $76.50 \pm 3.09$ | $80.18 \pm 2.61$ | $77.50 \pm 14.41$ | $92.39 \pm 0.42$ | $92.73 \pm 0.67$ |
| | coat | $72.91 \pm 2.44$ | $84.74 \pm 1.44$ | $64.64 \pm 1.91$ | $74.10 \pm 3.76$ | $55.17 \pm 7.93$ | $90.61 \pm 1.02$ | $91.50 \pm 1.13$ |
| | sandal | $80.91 \pm 4.66$ | $70.19 \pm 2.17$ | $76.13 \pm 6.09$ | $75.56 \pm 12.49$ | $84.09 \pm 6.49$ | $90.34 \pm 1.04$ | $93.42 \pm 0.70$ |
| | shirt | $56.62 \pm 5.11$ | $79.67 \pm 1.25$ | $55.87 \pm 1.99$ | $69.43 \pm 2.39$ | $59.90 \pm 6.33$ | $82.58 \pm 1.46$ | $84.46 \pm 1.15$ |
| | sneaker | $94.26 \pm 1.04$ | $94.60 \pm 1.09$ | $92.75 \pm 1.77$ | $95.00 \pm 0.48$ | $67.44 \pm 26.98$ | $98.66 \pm 0.30$ | $99.17 \pm 0.29$ |
| | bag | $51.22 \pm 4.38$ | $70.40 \pm 4.68$ | $48.00 \pm 2.27$ | $58.78 \pm 4.68$ | $44.84 \pm 8.31$ | $91.37 \pm 0.69$ | $95.47 \pm 0.52$ |
| | ankle-boot | $86.44 \pm 2.16$ | $91.65 \pm 0.93$ | $79.24 \pm 3.36$ | $87.82 \pm 2.43$ | $67.16 \pm 16.22$ | $98.72 \pm 0.17$ | $99.47 \pm 0.09$ |
| | average | $75.09 \pm 0.45$ | $84.75 \pm 0.15$ | $71.48 \pm 0.61$ | $78.93 \pm 1.17$ | $64.07 \pm 5.95$ | $92.39 \pm 0.20$ | $\mathbf{94.05 \pm 0.13}$ |
| CIFAR10 | airplane | $69.24 \pm 2.48$ | $62.68 \pm 0.86$ | $72.09 \pm 0.62$ | $64.37 \pm 3.64$ | $52.24 \pm 9.36$ | $57.98 \pm 3.45$ | $79.44 \pm 1.15$ |
| | automobile | $41.79 \pm 1.73$ | $31.72 \pm 0.86$ | $39.55 \pm 2.12$ | $33.38 \pm 2.07$ | $58.65 \pm 7.06$ | $75.47 \pm 2.96$ | $95.31 \pm 0.38$ |
| | bird | $65.19 \pm 1.49$ | $66.77 \pm 1.25$ | $66.01 \pm 0.78$ | $68.30 \pm 1.80$ | $48.27 \pm 2.65$ | $57.91 \pm 1.95$ | $75.40 \pm 1.94$ |
| | cat | $59.19 \pm 1.74$ | $52.70 \pm 1.69$ | $59.72 \pm 1.09$ | $57.74 \pm 1.17$ | $49.26 \pm 3.83$ | $57.04 \pm 2.65$ | $73.91 \pm 0.55$ |
| | deer | $61.16 \pm 2.54$ | $68.57 \pm 2.20$ | $62.22 \pm 3.42$ | $69.09 \pm 2.04$ | $52.72 \pm 5.71$ | $64.60 \pm 2.48$ | $84.09 \pm 1.39$ |
| | dog | $55.86 \pm 2.06$ | $51.57 \pm 0.68$ | $60.85 \pm 1.01$ | $55.28 \pm 1.74$ | $52.67 \pm 3.55$ | $62.28 \pm 2.57$ | $87.88 \pm 1.32$ |
| | frog | $49.96 \pm 3.02$ | $61.70 \pm 0.69$ | $46.00 \pm 1.30$ | $57.31 \pm 3.26$ | $64.59 \pm 2.15$ | $66.89 \pm 4.23$ | $84.98 \pm 1.02$ |
| | horse | $49.05 \pm 1.78$ | $46.81 \pm 0.84$ | $52.00 \pm 1.25$ | $48.72 \pm 0.81$ | $55.46 \pm 5.33$ | $77.00 \pm 2.23$ | $93.38 \pm 1.00$ |
| | ship | $72.79 \pm 2.31$ | $67.66 \pm 1.23$ | $72.59 \pm 1.21$ | $70.06 \pm 2.51$ | $55.48 \pm 5.76$ | $73.07 \pm 3.10$ | $92.34 \pm 0.89$ |
| | truck | $37.94 \pm 1.37$ | $32.35 \pm 2.63$ | $38.69 \pm 3.16$ | $34.25 \pm 3.42$ | $61.91 \pm 4.75$ | $75.17 \pm 2.24$ | $89.74 \pm 1.14$ |
| | average | $56.22 \pm 0.79$ | $54.25 \pm 0.28$ | $56.97 \pm 0.46$ | $55.85 \pm 0.64$ | $55.12 \pm 1.76$ | $66.74 \pm 0.32$ | $\mathbf{85.65 \pm 0.42}$ |
| CIFAR100 | aquatic mammals | $63.71 \pm 2.30$ | $63.77 \pm 1.73$ | $67.14 \pm 2.66$ | $66.63 \pm 2.18$ | $49.93 \pm 6.68$ | $55.26 \pm 3.29$ | $77.22 \pm 1.56$ |
| | fish | $65.17 \pm 2.02$ | $60.19 \pm 1.34$ | $63.22 \pm 1.13$ | $62.06 \pm 1.51$ | $44.59 \pm 3.47$ | $57.57 \pm 1.83$ | $70.32 \pm 2.41$ |
| | flowers | $38.88 \pm 2.59$ | $36.27 \pm 1.72$ | $37.21 \pm 1.85$ | $37.51 \pm 5.36$ | $66.02 \pm 10.09$ | $54.31 \pm 1.76$ | $79.04 \pm 2.54$ |
| | food containers | $60.73 \pm 4.23$ | $59.80 \pm 2.64$ | $61.84 \pm 0.43$ | $60.11 \pm 1.53$ | $46.68 \pm 3.95$ | $52.21 \pm 3.19$ | $83.32 \pm 0.98$ |
| | fruit and vegetables | $55.21 \pm 4.32$ | $45.63 \pm 0.86$ | $53.41 \pm 2.24$ | $48.58 \pm 3.19$ | $65.91 \pm 2.93$ | $53.69 \pm 3.10$ | $78.31 \pm 1.05$ |
| | household electrical devices | $52.16 \pm 1.47$ | $44.09 \pm 2.21$ | $56.95 \pm 1.88$ | $49.75 \pm 1.88$ | $50.11 \pm 1.26$ | $48.73 \pm 4.67$ | $64.23 \pm 2.02$ |
| | household furniture | $59.73 \pm 4.97$ | $53.85 \pm 2.90$ | $66.22 \pm 2.04$ | $56.15 \pm 3.38$ | $50.62 \pm 7.51$ | $50.67 \pm 2.93$ | $82.26 \pm 0.93$ |
| | inserts | $46.31 \pm 0.92$ | $50.45 \pm 1.82$ | $46.26 \pm 2.11$ | $48.20 \pm 2.68$ | $51.36 \pm 1.29$ | $52.93 \pm 2.99$ | $76.14 \pm 0.92$ |
| | large carnivores | $50.41 \pm 2.65$ | $55.42 \pm 2.76$ | $54.01 \pm 2.75$ | $58.02 \pm 4.39$ | $59.22 \pm 3.13$ | $61.47 \pm 3.48$ | $87.59 \pm 1.34$ |
| | large man-made outdoor things | $65.53 \pm 3.69$ | $63.06 \pm 3.87$ | $70.15 \pm 4.05$ | $66.28 \pm 4.59$ | $57.15 \pm 9.20$ | $67.45 \pm 2.07$ | $88.67 \pm 2.38$ |
| | large natural outdoor scenes | $83.36 \pm 1.74$ | $82.40 \pm 2.64$ | $80.88 \pm 0.96$ | $82.01 \pm 2.28$ | $51.37 \pm 6.78$ | $62.61 \pm 2.48$ | $85.93 \pm 0.65$ |
| | large omnivores and herbivores | $51.13 \pm 3.09$ | $51.06 \pm 3.35$ | $56.66 \pm 1.77$ | $56.04 \pm 3.43$ | $57.51 \pm 1.90$ | $55.61 \pm 0.85$ | $83.65 \pm 0.69$ |
| | medium-sized mammals | $58.80 \pm 2.47$ | $57.48 \pm 1.80$ | $62.64 \pm 2.64$ | $63.09 \pm 3.76$ | $61.02 \pm 2.59$ | $58.72 \pm 2.98$ | $84.72 \pm 1.08$ |
| | non-insert invertebrates | $52.91 \pm 1.39$ | $57.02 \pm 1.97$ | $50.59 \pm 0.92$ | $54.87 \pm 0.77$ | $49.94 \pm 2.39$ | $49.14 \pm 0.57$ | $63.87 \pm 1.65$ |
| | people | $46.69 \pm 1.61$ | $39.44 \pm 2.70$ | $47.94 \pm 1.95$ | $42.20 \pm 2.74$ | $56.84 \pm 2.61$ | $56.42 \pm 3.45$ | $90.61 \pm 0.93$ |
| | reptiles | $54.38 \pm 1.44$ | $55.95 \pm 2.71$ | $53.88 \pm 0.65$ | $56.69 \pm 2.82$ | $48.79 \pm 2.97$ | $52.66 \pm 1.96$ | $69.78 \pm 3.44$ |
| | small mammals | $59.34 \pm 2.09$ | $62.87 \pm 0.70$ | $62.37 \pm 2.23$ | $64.68 \pm 1.14$ | $54.79 \pm 3.84$ | $57.95 \pm 1.80$ | $75.05 \pm 0.88$ |
| | trees | $62.66 \pm 4.30$ | $63.74 \pm 3.50$ | $60.81 \pm 3.37$ | $63.36 \pm 2.66$ | $58.04 \pm 3.05$ | $75.59 \pm 2.72$ | $91.91 \pm 0.57$ |
| | vehicles 1 | $36.04 \pm 2.54$ | $36.45 \pm 1.35$ | $37.85 \pm 4.59$ | $37.72 \pm 3.15$ | $58.24 \pm 3.25$ | $61.66 \pm 3.38$ | $89.94 \pm 0.81$ |
| | vehicles 2 | $48.37 \pm 2.26$ | $47.00 \pm 0.79$ | $53.43 \pm 2.34$ | $48.34 \pm 1.41$ | $49.17 \pm 5.60$ | $57.30 \pm 5.00$ | $84.18 \pm 1.60$ |
| | average | $55.57 \pm 0.60$ | $54.30 \pm 0.26$ | $57.17 \pm 0.58$ | $56.11 \pm 0.65$ | $54.37 \pm 0.51$ | $57.10 \pm 0.67$ | $\mathbf{80.34 \pm 0.20}$ |
| SVHN | 0 | $52.16 \pm 1.67$ | $60.13 \pm 0.98$ | $53.52 \pm 1.50$ | $51.90 \pm 1.10$ | $49.68 \pm 1.87$ | $78.25 \pm 1.08$ | $80.95 \pm 1.23$ |
| | 1 | $54.74 \pm 1.59$ | $61.06 \pm 0.59$ | $55.20 \pm 1.72$ | $58.03 \pm 0.92$ | $48.28 \pm 1.32$ | $69.28 \pm 2.99$ | $78.02 \pm 0.85$ |
| | 2 | $51.42 \pm 1.24$ | $53.94 \pm 0.94$ | $49.88 \pm 0.65$ | $51.87 \pm 0.81$ | $49.57 \pm 1.11$ | $78.31 \pm 2.61$ | $93.12 \pm 0.38$ |
| | 3 | $50.09 \pm 0.85$ | $52.51 \pm 0.65$ | $49.37 \pm 0.61$ | $50.10 \pm 0.94$ | $50.41 \pm 0.55$ | $71.82 \pm 2.49$ | $89.39 \pm 0.61$ |
| | 4 | $52.34 \pm 1.45$ | $56.98 \pm 1.29$ | $52.45 \pm 0.96$ | $53.25 \pm 0.97$ | $49.47 \pm 1.55$ | $76.67 \pm 2.47$ | $94.01 \pm 0.19$ |
| | 5 | $49.42 \pm 1.24$ | $52.68 \pm 0.77$ | $48.55 \pm 1.34$ | $50.18 \pm 0.87$ | $51.16 \pm 0.67$ | $75.18 \pm 2.50$ | $93.25 \pm 0.35$ |
| | 6 | $49.71 \pm 0.98$ | $52.21 \pm 0.66$ | $47.65 \pm 0.73$ | $50.00 \pm 0.69$ | $48.64 \pm 1.31$ | $79.73 \pm 1.72$ | $91.70 \pm 0.18$ |
| | 7 | $54.28 \pm 1.94$ | $57.62 \pm 1.32$ | $52.22 \pm 1.29$ | $54.47 \pm 1.82$ | $49.59 \pm 1.57$ | $84.72 \pm 0.62$ | $95.03 \pm 0.19$ |
| | 8 | $50.13 \pm 1.34$ | $53.90 \pm 1.09$ | $48.94 \pm 1.77$ | $50.78 \pm 1.53$ | $49.54 \pm 1.57$ | $68.04 \pm 2.50$ | $81.39 \pm 1.24$ |
| | 9 | $50.17 \pm 0.77$ | $52.86 \pm 1.72$ | $48.58 \pm 0.36$ | $51.21 \pm 1.06$ | $50.46 \pm 1.28$ | $80.24 \pm 1.37$ | $92.17 \pm 0.73$ |
| | average | $51.45 \pm 0.27$ | $55.39 \pm 0.46$ | $50.64 \pm 0.41$ | $52.18 \pm 0.15$ | $49.68 \pm 0.46$ | $76.22 \pm 0.31$ | $\mathbf{88.90 \pm 0.24}$ |

Table 2: AUROC (%) when $\rho = 0.1$

| Dataset | Class name | CAE | CAE-IF | DRAE | RDAE | DAGMM | SSD-IF | $E^3$Outlier |
|---|---|---|---|---|---|---|---|---|
| MNIST | 0 | 57.50 ± 5.33 | 90.19 ± 2.19 | 61.86 ± 13.38 | 62.51 ± 5.83 | 74.52 ± 26.99 | 90.90 ± 1.01 | 94.38 ± 0.28 |
| | 1 | 99.09 ± 0.16 | 99.27 ± 0.14 | 81.80 ± 7.54 | 99.19 ± 0.08 | 89.23 ± 15.65 | 96.59 ± 0.42 | 95.22 ± 0.33 |
| | 2 | 53.78 ± 5.92 | 74.54 ± 1.23 | 56.94 ± 5.53 | 61.12 ± 1.81 | 48.74 ± 10.55 | 91.70 ± 1.08 | 93.07 ± 0.35 |
| | 3 | 61.16 ± 6.48 | 81.80 ± 2.60 | 67.41 ± 5.25 | 65.91 ± 1.84 | 50.51 ± 3.41 | 95.61 ± 0.87 | 97.20 ± 0.29 |
| | 4 | 69.94 ± 3.91 | 83.30 ± 2.22 | 66.22 ± 4.98 | 76.05 ± 1.20 | 62.22 ± 8.78 | 95.81 ± 0.82 | 95.64 ± 0.30 |
| | 5 | 60.38 ± 0.49 | 79.06 ± 1.11 | 64.34 ± 3.64 | 65.11 ± 2.43 | 52.77 ± 5.37 | 92.77 ± 1.28 | 91.72 ± 0.54 |
| | 6 | 73.47 ± 4.88 | 91.10 ± 0.91 | 74.66 ± 5.53 | 75.03 ± 1.94 | 66.57 ± 11.11 | 97.53 ± 0.08 | 96.79 ± 0.12 |
| | 7 | 81.79 ± 2.46 | 91.90 ± 0.93 | 77.47 ± 3.57 | 85.64 ± 1.11 | 75.49 ± 10.55 | 93.89 ± 0.83 | 93.30 ± 0.21 |
| | 8 | 48.20 ± 1.78 | 73.93 ± 2.57 | 48.75 ± 5.28 | 51.38 ± 2.30 | 56.98 ± 8.17 | 87.80 ± 1.49 | 88.51 ± 0.99 |
| | 9 | 74.61 ± 1.90 | 89.92 ± 1.16 | 69.50 ± 16.65 | 75.71 ± 2.60 | 63.08 ± 4.61 | 95.76 ± 0.54 | 95.12 ± 0.34 |
| | *average* | 67.99 ± 0.70 | 85.50 ± 0.53 | 66.90 ± 2.55 | 71.76 ± 0.99 | 64.01 ± 3.70 | 93.84 ± 0.32 | **94.09 ± 0.13** |
| Fashion-MNIST | t-shirt | 57.17 ± 6.39 | 85.55 ± 0.94 | 61.27 ± 6.82 | 70.56 ± 4.18 | 58.67 ± 13.52 | 88.09 ± 0.76 | 92.55 ± 0.43 |
| | trouser | 96.05 ± 0.30 | 96.34 ± 0.55 | 76.73 ± 13.97 | 96.94 ± 0.36 | 74.20 ± 12.80 | 97.82 ± 0.42 | 98.12 ± 0.16 |
| | pullover | 60.73 ± 2.57 | 82.82 ± 1.33 | 61.55 ± 3.28 | 68.79 ± 4.73 | 52.09 ± 16.12 | 89.41 ± 0.97 | 91.86 ± 0.90 |
| | dress | 77.34 ± 1.80 | 85.22 ± 0.70 | 75.43 ± 3.23 | 79.12 ± 3.22 | 73.04 ± 23.18 | 91.00 ± 0.71 | 89.99 ± 0.84 |
| | coat | 67.42 ± 2.27 | 82.68 ± 1.05 | 60.42 ± 3.50 | 73.23 ± 1.59 | 49.92 ± 14.90 | 89.95 ± 0.37 | 91.41 ± 0.44 |
| | sandal | 73.94 ± 4.21 | 67.28 ± 2.28 | 72.57 ± 4.08 | 73.64 ± 1.33 | 84.47 ± 6.00 | 88.09 ± 1.08 | 93.64 ± 0.29 |
| | shirt | 52.09 ± 3.99 | 77.53 ± 0.78 | 54.25 ± 3.28 | 66.34 ± 3.00 | 57.02 ± 7.42 | 82.06 ± 0.79 | 83.50 ± 1.15 |
| | sneaker | 91.00 ± 1.69 | 91.56 ± 0.75 | 91.69 ± 1.48 | 91.15 ± 1.14 | 75.66 ± 28.60 | 98.18 ± 0.33 | 99.11 ± 0.14 |
| | bag | 45.82 ± 3.31 | 67.18 ± 1.16 | 46.09 ± 1.75 | 51.25 ± 2.93 | 52.68 ± 7.54 | 82.36 ± 1.83 | 93.16 ± 0.38 |
| | ankle-boot | 81.37 ± 2.64 | 86.68 ± 1.47 | 71.20 ± 5.33 | 82.35 ± 5.15 | 61.76 ± 24.67 | 98.58 ± 0.27 | 99.40 ± 0.09 |
| | *average* | 70.29 ± 0.92 | 82.28 ± 0.19 | 67.12 ± 1.76 | 75.34 ± 1.19 | 63.95 ± 9.90 | 90.55 ± 0.18 | **93.27 ± 0.14** |
| CIFAR10 | airplane | 68.34 ± 4.04 | 61.93 ± 1.11 | 71.21 ± 0.83 | 62.65 ± 2.21 | 49.29 ± 7.83 | 52.22 ± 2.43 | 78.22 ± 1.19 |
| | automobile | 37.81 ± 3.07 | 32.69 ± 1.78 | 38.41 ± 0.90 | 33.35 ± 2.22 | 61.62 ± 3.92 | 71.70 ± 2.12 | 93.78 ± 0.49 |
| | bird | 61.87 ± 4.59 | 66.43 ± 1.20 | 65.56 ± 1.06 | 66.73 ± 1.33 | 48.97 ± 4.87 | 54.23 ± 2.75 | 71.28 ± 1.36 |
| | cat | 59.15 ± 2.38 | 51.05 ± 1.14 | 59.72 ± 1.17 | 55.14 ± 2.34 | 50.61 ± 0.82 | 55.55 ± 1.78 | 70.40 ± 1.29 |
| | deer | 64.54 ± 2.29 | 70.42 ± 1.34 | 61.60 ± 1.55 | 70.94 ± 2.37 | 57.03 ± 4.41 | 64.80 ± 1.19 | 82.18 ± 0.22 |
| | dog | 57.40 ± 3.15 | 50.59 ± 1.03 | 61.05 ± 0.91 | 55.62 ± 1.20 | 54.53 ± 4.55 | 62.05 ± 2.88 | 84.80 ± 0.92 |
| | frog | 53.99 ± 4.20 | 61.93 ± 1.10 | 41.03 ± 1.32 | 56.82 ± 2.22 | 62.58 ± 5.92 | 64.62 ± 3.74 | 81.88 ± 1.55 |
| | horse | 46.49 ± 1.29 | 46.01 ± 1.32 | 50.75 ± 1.06 | 49.40 ± 1.86 | 57.27 ± 3.10 | 73.76 ± 0.94 | 91.60 ± 0.48 |
| | ship | 72.33 ± 2.63 | 66.72 ± 0.84 | 72.58 ± 1.78 | 67.77 ± 2.02 | 54.55 ± 14.22 | 68.77 ± 2.78 | 92.00 ± 0.16 |
| | truck | 36.39 ± 2.79 | 33.26 ± 1.49 | 38.24 ± 1.54 | 35.98 ± 1.67 | 64.36 ± 7.05 | 72.27 ± 1.77 | 89.19 ± 0.78 |
| | *average* | 55.83 ± 1.37 | 54.10 ± 0.11 | 56.01 ± 0.10 | 55.44 ± 0.58 | 56.08 ± 2.00 | 64.00 ± 0.30 | **83.53 ± 0.29** |
| CIFAR100 | aquatic mammals | 64.21 ± 3.67 | 61.34 ± 0.70 | 65.61 ± 2.19 | 63.76 ± 1.16 | 49.45 ± 2.51 | 54.87 ± 1.73 | 75.45 ± 2.51 |
| | fish | 63.53 ± 3.38 | 57.71 ± 1.17 | 64.65 ± 1.38 | 60.61 ± 2.92 | 43.67 ± 3.19 | 56.08 ± 1.51 | 69.72 ± 1.69 |
| | flowers | 37.26 ± 9.90 | 38.46 ± 1.95 | 35.15 ± 1.80 | 36.16 ± 4.81 | 64.37 ± 6.98 | 55.77 ± 1.87 | 77.44 ± 0.72 |
| | food containers | 61.06 ± 5.31 | 57.61 ± 1.67 | 63.25 ± 0.96 | 59.11 ± 2.01 | 50.12 ± 2.28 | 48.60 ± 2.97 | 80.03 ± 1.44 |
| | fruit and vegetables | 50.21 ± 4.25 | 42.54 ± 1.95 | 53.08 ± 1.97 | 41.96 ± 5.07 | 62.67 ± 7.49 | 53.85 ± 2.59 | 76.17 ± 1.38 |
| | household electrical devices | 52.50 ± 3.31 | 43.65 ± 0.47 | 52.18 ± 5.58 | 46.37 ± 2.02 | 48.37 ± 2.48 | 49.04 ± 0.97 | 63.25 ± 2.31 |
| | household furniture | 59.89 ± 4.51 | 52.25 ± 1.04 | 62.48 ± 1.68 | 53.67 ± 2.66 | 52.23 ± 4.69 | 46.60 ± 1.03 | 80.08 ± 0.98 |
| | inserts | 48.57 ± 2.52 | 52.56 ± 2.03 | 46.68 ± 1.27 | 49.33 ± 4.21 | 49.26 ± 2.34 | 52.70 ± 0.83 | 72.60 ± 0.94 |
| | large carnivores | 54.25 ± 3.41 | 57.01 ± 2.21 | 50.74 ± 2.87 | 59.44 ± 3.12 | 61.05 ± 0.75 | 61.26 ± 3.29 | 86.23 ± 1.09 |
| | large man-made outdoor things | 65.25 ± 3.51 | 63.03 ± 1.13 | 66.00 ± 2.66 | 67.77 ± 1.29 | 65.50 ± 6.67 | 62.01 ± 1.11 | 88.44 ± 1.03 |
| | large natural outdoor scenes | 76.84 ± 8.52 | 81.77 ± 2.56 | 82.53 ± 0.93 | 79.70 ± 4.07 | 56.33 ± 5.02 | 61.26 ± 2.72 | 85.05 ± 1.21 |
| | large omnivores and herbivores | 53.57 ± 1.32 | 53.13 ± 1.11 | 55.69 ± 1.14 | 59.60 ± 2.37 | 59.10 ± 2.18 | 57.03 ± 3.32 | 83.02 ± 0.87 |
| | medium-sized mammals | 59.30 ± 3.97 | 57.63 ± 1.16 | 56.93 ± 1.21 | 61.60 ± 2.32 | 61.95 ± 2.64 | 58.56 ± 2.35 | 82.99 ± 0.50 |
| | non-insert invertebrates | 52.15 ± 1.83 | 58.10 ± 0.90 | 51.55 ± 0.74 | 55.07 ± 1.33 | 47.09 ± 2.04 | 50.52 ± 1.38 | 62.62 ± 1.52 |
| | people | 47.10 ± 2.76 | 38.68 ± 0.70 | 47.47 ± 1.57 | 45.05 ± 1.35 | 54.60 ± 2.37 | 53.95 ± 2.20 | 89.23 ± 0.51 |
| | reptiles | 55.48 ± 0.85 | 57.29 ± 1.18 | 54.41 ± 1.35 | 56.80 ± 0.62 | 49.98 ± 1.71 | 52.28 ± 2.20 | 70.09 ± 1.81 |
| | small mammals | 60.12 ± 4.90 | 63.57 ± 1.20 | 61.92 ± 1.75 | 65.07 ± 2.18 | 56.21 ± 3.30 | 55.94 ± 1.44 | 75.50 ± 1.62 |
| | trees | 57.28 ± 5.18 | 65.36 ± 1.33 | 57.70 ± 2.59 | 63.95 ± 2.67 | 59.59 ± 6.26 | 69.69 ± 4.03 | 91.66 ± 0.54 |
| | vehicles 1 | 36.98 ± 1.86 | 40.56 ± 1.01 | 34.45 ± 2.06 | 39.18 ± 1.42 | 56.28 ± 1.44 | 58.32 ± 2.77 | 89.62 ± 0.62 |
| | vehicles 2 | 47.82 ± 4.64 | 48.10 ± 2.26 | 49.44 ± 2.34 | 48.83 ± 1.63 | 50.36 ± 2.52 | 54.38 ± 0.38 | 84.96 ± 1.15 |
| | *average* | 55.17 ± 1.51 | 54.52 ± 0.40 | 55.59 ± 0.68 | 55.55 ± 0.81 | 54.91 ± 0.74 | 55.63 ± 0.21 | **79.21 ± 0.29** |
| SVHN | 0 | 50.61 ± 1.80 | 58.86 ± 1.51 | 51.59 ± 1.06 | 52.16 ± 1.70 | 49.83 ± 0.99 | 78.89 ± 1.65 | 73.51 ± 1.19 |
| | 1 | 56.45 ± 1.86 | 60.91 ± 1.43 | 56.54 ± 2.10 | 57.97 ± 1.86 | 49.75 ± 0.33 | 66.66 ± 1.28 | 72.50 ± 1.26 |
| | 2 | 51.17 ± 0.91 | 54.70 ± 1.43 | 51.44 ± 1.34 | 52.15 ± 0.89 | 49.68 ± 1.08 | 73.45 ± 1.54 | 91.63 ± 0.45 |
| | 3 | 50.51 ± 0.49 | 52.18 ± 0.58 | 49.96 ± 0.77 | 50.88 ± 0.86 | 50.95 ± 0.56 | 68.46 ± 2.44 | 87.44 ± 0.23 |
| | 4 | 52.18 ± 1.50 | 56.26 ± 1.50 | 52.47 ± 0.67 | 54.94 ± 0.69 | 49.53 ± 1.33 | 70.15 ± 3.21 | 92.54 ± 0.39 |
| | 5 | 49.06 ± 0.79 | 52.15 ± 1.04 | 48.49 ± 1.05 | 49.59 ± 0.87 | 50.44 ± 1.03 | 74.63 ± 3.12 | 92.05 ± 0.38 |
| | 6 | 49.86 ± 1.09 | 51.21 ± 1.38 | 48.11 ± 1.01 | 50.11 ± 0.57 | 50.55 ± 1.92 | 75.12 ± 2.22 | 89.94 ± 0.37 |
| | 7 | 51.91 ± 0.47 | 57.12 ± 0.92 | 53.15 ± 1.16 | 52.41 ± 1.61 | 48.85 ± 1.97 | 78.91 ± 3.34 | 93.26 ± 0.30 |
| | 8 | 49.42 ± 1.40 | 53.38 ± 1.17 | 48.03 ± 1.37 | 49.79 ± 1.56 | 49.51 ± 0.68 | 69.96 ± 2.11 | 77.24 ± 2.22 |
| | 9 | 50.77 ± 0.67 | 53.57 ± 0.79 | 49.83 ± 0.62 | 51.39 ± 0.51 | 50.98 ± 1.33 | 77.23 ± 1.17 | 90.00 ± 0.35 |
| | *average* | 51.20 ± 0.37 | 55.03 ± 0.67 | 50.96 ± 0.59 | 52.14 ± 0.56 | 50.00 ± 0.45 | 73.35 ± 0.62 | **86.01 ± 0.18** |

Table 3: AUROC (%) when $\rho = 0.15$

| Dataset | Class name | CAE | CAE-IF | DRAE | RDAE | DAGMM | SSD-IF | $E^3$Outlier |
|---|---|---|---|---|---|---|---|---|
| MNIST | 0 | 48.20 ± 5.76 | 88.03 ± 1.81 | 61.63 ± 17.32 | 54.03 ± 5.75 | 74.75 ± 24.67 | 89.48 ± 2.75 | 91.42 ± 0.29 |
| | 1 | 98.75 ± 0.33 | 99.22 ± 0.14 | 87.47 ± 9.82 | 98.93 ± 0.17 | 80.34 ± 29.29 | 96.14 ± 0.61 | 91.94 ± 0.87 |
| | 2 | 51.38 ± 4.00 | 71.94 ± 1.99 | 55.99 ± 6.25 | 53.31 ± 3.83 | 49.72 ± 4.39 | 89.60 ± 0.78 | 92.33 ± 0.24 |
| | 3 | 55.72 ± 2.85 | 79.23 ± 2.30 | 66.26 ± 8.38 | 64.73 ± 1.44 | 56.79 ± 9.84 | 93.83 ± 0.49 | 96.89 ± 0.13 |
| | 4 | 68.00 ± 2.40 | 80.53 ± 1.20 | 69.45 ± 9.19 | 72.22 ± 1.06 | 65.42 ± 11.04 | 94.38 ± 0.75 | 94.50 ± 0.32 |
| | 5 | 55.61 ± 3.69 | 85.37 ± 0.98 | 58.19 ± 2.84 | 51.99 ± 6.43 | 54.14 ± 4.10 | 89.39 ± 1.09 | 90.41 ± 0.55 |
| | 6 | 66.14 ± 3.38 | 88.75 ± 1.94 | 60.31 ± 11.78 | 69.38 ± 1.35 | 57.53 ± 12.61 | 96.21 ± 0.24 | 96.38 ± 0.14 |
| | 7 | 75.57 ± 5.34 | 91.47 ± 0.59 | 77.40 ± 2.35 | 80.66 ± 2.86 | 62.94 ± 16.16 | 92.70 ± 1.00 | 92.49 ± 0.08 |
| | 8 | 46.34 ± 4.51 | 71.36 ± 1.37 | 58.80 ± 9.70 | 49.18 ± 5.09 | 54.19 ± 5.41 | 84.48 ± 1.26 | 87.49 ± 0.94 |
| | 9 | 73.01 ± 2.44 | 89.10 ± 0.87 | 69.74 ± 5.35 | 75.23 ± 3.18 | 63.72 ± 8.46 | 94.98 ± 0.53 | 94.60 ± 0.39 |
| | *average* | 63.87 ± 0.96 | 84.50 ± 0.32 | 66.52 ± 2.72 | 66.97 ± 1.77 | 61.95 ± 2.81 | 92.12 ± 0.45 | **92.85 ± 0.15** |
| Fashion-MNIST | t-shirt | 60.61 ± 3.73 | 84.49 ± 0.56 | 64.85 ± 2.07 | 70.40 ± 3.24 | 54.69 ± 14.71 | 86.86 ± 0.40 | 92.19 ± 0.32 |
| | trouser | 95.42 ± 0.57 | 95.67 ± 0.86 | 93.12 ± 1.51 | 96.41 ± 0.56 | 51.66 ± 15.28 | 97.36 ± 0.65 | 97.48 ± 0.14 |
| | pullover | 59.83 ± 6.78 | 81.61 ± 0.66 | 60.95 ± 5.63 | 65.04 ± 6.54 | 51.20 ± 12.41 | 89.00 ± 0.63 | 91.73 ± 0.85 |
| | dress | 75.63 ± 1.67 | 82.62 ± 1.51 | 73.70 ± 1.81 | 75.31 ± 3.97 | 59.73 ± 10.67 | 89.44 ± 0.99 | 87.66 ± 1.04 |
| | coat | 64.16 ± 1.92 | 81.28 ± 1.25 | 60.31 ± 2.28 | 65.96 ± 8.46 | 53.04 ± 17.50 | 89.34 ± 0.66 | 90.74 ± 0.40 |
| | sandal | 69.33 ± 2.03 | 63.26 ± 1.67 | 72.39 ± 1.27 | 71.89 ± 4.61 | 82.14 ± 12.76 | 86.01 ± 1.63 | 92.63 ± 0.45 |
| | shirt | 51.55 ± 2.41 | 75.81 ± 0.87 | 52.37 ± 1.37 | 62.19 ± 5.14 | 49.20 ± 4.68 | 81.49 ± 0.85 | 82.85 ± 0.55 |
| | sneaker | 87.78 ± 3.06 | 90.94 ± 0.86 | 88.48 ± 2.18 | 89.97 ± 1.96 | 60.34 ± 18.65 | 98.04 ± 0.36 | 98.94 ± 0.20 |
| | bag | 43.85 ± 8.22 | 61.98 ± 1.95 | 44.98 ± 2.90 | 46.16 ± 8.05 | 47.70 ± 11.69 | 77.60 ± 2.37 | 89.44 ± 0.86 |
| | ankle-boot | 76.88 ± 4.26 | 81.43 ± 2.67 | 70.67 ± 2.30 | 78.04 ± 2.70 | 69.76 ± 12.80 | 98.12 ± 0.28 | 99.34 ± 0.08 |
| | *average* | 68.50 ± 1.66 | 79.91 ± 0.56 | 68.18 ± 0.81 | 72.14 ± 0.89 | 57.95 ± 6.79 | 89.33 ± 0.20 | **92.30 ± 0.10** |
| CIFAR10 | airplane | 72.24 ± 3.19 | 62.41 ± 1.45 | 72.02 ± 0.77 | 64.64 ± 2.13 | 52.56 ± 6.24 | 52.05 ± 1.46 | 76.93 ± 0.48 |
| | automobile | 37.28 ± 2.39 | 32.54 ± 0.73 | 37.84 ± 1.11 | 33.26 ± 2.21 | 61.53 ± 2.87 | 70.47 ± 2.94 | 93.42 ± 0.66 |
| | bird | 61.74 ± 1.33 | 65.88 ± 0.38 | 64.73 ± 0.73 | 66.80 ± 0.89 | 46.26 ± 4.70 | 52.68 ± 1.72 | 69.16 ± 0.92 |
| | cat | 58.60 ± 1.14 | 52.07 ± 0.89 | 59.94 ± 0.46 | 55.47 ± 1.10 | 52.13 ± 2.25 | 54.22 ± 2.06 | 66.28 ± 1.22 |
| | deer | 62.84 ± 2.50 | 70.29 ± 0.73 | 62.08 ± 0.20 | 70.04 ± 1.89 | 53.80 ± 2.75 | 63.81 ± 1.51 | 80.24 ± 0.48 |
| | dog | 57.39 ± 1.72 | 50.61 ± 0.75 | 60.37 ± 1.14 | 55.09 ± 1.87 | 53.00 ± 1.99 | 62.91 ± 1.67 | 80.15 ± 1.34 |
| | frog | 52.43 ± 3.20 | 61.39 ± 0.69 | 43.62 ± 1.86 | 55.69 ± 1.91 | 60.94 ± 4.26 | 63.55 ± 1.85 | 78.37 ± 1.21 |
| | horse | 47.64 ± 0.85 | 46.02 ± 0.80 | 50.09 ± 0.81 | 48.50 ± 0.61 | 57.69 ± 4.33 | 68.98 ± 2.44 | 89.37 ± 0.86 |
| | ship | 70.92 ± 3.31 | 66.50 ± 1.01 | 72.76 ± 1.13 | 67.00 ± 5.70 | 54.72 ± 13.11 | 67.87 ± 3.79 | 90.85 ± 0.51 |
| | truck | 40.12 ± 4.11 | 32.59 ± 0.92 | 39.61 ± 0.65 | 34.06 ± 2.38 | 60.97 ± 4.48 | 69.26 ± 1.54 | 88.49 ± 0.67 |
| | *average* | 56.12 ± 0.47 | 54.03 ± 0.08 | 56.31 ± 0.22 | 55.06 ± 0.63 | 55.36 ± 0.54 | 62.58 ± 0.76 | **81.33 ± 0.27** |
| CIFAR100 | aquatic mammals | 64.76 ± 1.48 | 61.36 ± 1.32 | 65.65 ± 1.03 | 64.54 ± 0.90 | 49.52 ± 1.87 | 57.00 ± 2.34 | 74.54 ± 1.32 |
| | fish | 65.21 ± 0.92 | 58.83 ± 0.69 | 63.43 ± 0.75 | 60.55 ± 1.93 | 43.35 ± 5.10 | 56.08 ± 0.65 | 70.39 ± 0.89 |
| | flowers | 35.86 ± 3.60 | 34.66 ± 1.16 | 33.76 ± 2.31 | 35.15 ± 4.08 | 59.45 ± 9.27 | 55.77 ± 1.15 | 76.84 ± 2.26 |
| | food containers | 61.67 ± 1.81 | 59.74 ± 0.59 | 60.94 ± 0.79 | 57.54 ± 2.76 | 47.03 ± 1.06 | 52.25 ± 1.91 | 77.87 ± 0.79 |
| | fruit and vegetables | 52.51 ± 4.58 | 44.93 ± 1.14 | 52.60 ± 1.57 | 48.55 ± 4.36 | 58.92 ± 8.45 | 54.07 ± 2.50 | 74.51 ± 1.61 |
| | household electrical devices | 52.64 ± 2.22 | 43.85 ± 0.87 | 56.27 ± 1.07 | 45.34 ± 2.26 | 51.85 ± 4.27 | 50.16 ± 1.12 | 62.57 ± 2.45 |
| | household furniture | 59.55 ± 2.31 | 54.00 ± 1.83 | 64.79 ± 1.63 | 55.87 ± 2.88 | 51.45 ± 5.34 | 47.99 ± 1.41 | 78.98 ± 1.07 |
| | inserts | 49.74 ± 3.41 | 50.17 ± 1.39 | 45.83 ± 1.37 | 48.83 ± 0.98 | 49.61 ± 5.28 | 52.53 ± 1.56 | 71.86 ± 0.88 |
| | large carnivores | 49.29 ± 1.79 | 54.29 ± 2.52 | 52.95 ± 1.30 | 54.58 ± 0.47 | 59.94 ± 1.74 | 59.70 ± 1.83 | 84.97 ± 2.15 |
| | large man-made outdoor things | 63.75 ± 2.18 | 62.99 ± 1.15 | 69.23 ± 1.25 | 66.91 ± 2.11 | 63.31 ± 4.17 | 63.42 ± 3.07 | 88.10 ± 0.87 |
| | large natural outdoor scenes | 82.94 ± 1.20 | 81.92 ± 1.35 | 82.24 ± 0.52 | 82.24 ± 1.56 | 51.00 ± 3.40 | 56.81 ± 6.28 | 83.81 ± 0.75 |
| | large omnivores and herbivores | 53.23 ± 1.23 | 51.65 ± 0.47 | 58.33 ± 0.78 | 56.39 ± 2.05 | 58.92 ± 2.88 | 54.93 ± 1.52 | 81.51 ± 1.31 |
| | medium-sized mammals | 53.45 ± 4.20 | 55.41 ± 1.11 | 56.76 ± 1.17 | 60.14 ± 1.99 | 60.68 ± 1.91 | 57.72 ± 2.14 | 81.26 ± 1.44 |
| | non-insect invertebrates | 51.75 ± 1.41 | 56.54 ± 0.83 | 49.94 ± 1.08 | 53.14 ± 0.69 | 47.63 ± 1.87 | 49.60 ± 0.85 | 59.93 ± 1.01 |
| | people | 47.20 ± 1.90 | 39.39 ± 0.75 | 48.55 ± 1.37 | 44.22 ± 1.31 | 56.19 ± 3.39 | 51.79 ± 2.71 | 88.26 ± 1.28 |
| | reptiles | 51.82 ± 1.87 | 55.31 ± 1.02 | 53.02 ± 1.52 | 55.61 ± 0.64 | 49.80 ± 2.43 | 53.97 ± 2.13 | 68.07 ± 2.21 |
| | small mammals | 59.50 ± 2.27 | 62.28 ± 1.39 | 61.31 ± 1.25 | 65.04 ± 1.78 | 55.69 ± 3.04 | 57.72 ± 2.59 | 73.37 ± 1.51 |
| | trees | 59.88 ± 2.13 | 63.23 ± 1.62 | 58.18 ± 2.71 | 62.42 ± 1.25 | 60.98 ± 5.25 | 69.51 ± 2.78 | 91.13 ± 0.80 |
| | vehicles 1 | 36.12 ± 1.18 | 36.96 ± 1.09 | 37.99 ± 1.50 | 36.98 ± 1.82 | 58.17 ± 2.02 | 56.77 ± 2.47 | 89.05 ± 0.79 |
| | vehicles 2 | 47.38 ± 2.69 | 45.91 ± 1.65 | 51.48 ± 1.50 | 46.81 ± 0.91 | 51.43 ± 3.29 | 52.79 ± 1.67 | 84.14 ± 0.97 |
| | *average* | 54.91 ± 0.23 | 53.67 ± 0.45 | 56.09 ± 0.45 | 55.04 ± 0.44 | 54.25 ± 0.68 | 55.53 ± 0.48 | **78.06 ± 0.42** |
| SVHN | 0 | 51.13 ± 1.69 | 57.53 ± 0.73 | 51.67 ± 0.87 | 53.24 ± 1.27 | 49.77 ± 1.08 | 77.38 ± 2.97 | 68.03 ± 2.17 |
| | 1 | 54.35 ± 1.78 | 59.87 ± 0.60 | 56.24 ± 1.61 | 56.86 ± 0.53 | 49.25 ± 0.99 | 65.39 ± 2.59 | 69.19 ± 1.81 |
| | 2 | 51.19 ± 0.90 | 54.09 ± 0.90 | 51.25 ± 0.52 | 52.58 ± 1.07 | 49.50 ± 0.76 | 72.52 ± 3.46 | 90.04 ± 0.78 |
| | 3 | 50.00 ± 0.67 | 51.43 ± 0.27 | 49.20 ± 0.62 | 49.85 ± 0.55 | 50.42 ± 0.46 | 67.13 ± 2.24 | 85.95 ± 0.28 |
| | 4 | 52.42 ± 0.83 | 55.14 ± 1.47 | 52.84 ± 0.46 | 54.28 ± 1.09 | 49.50 ± 1.11 | 69.07 ± 1.75 | 90.45 ± 0.36 |
| | 5 | 49.81 ± 0.56 | 51.99 ± 0.33 | 48.84 ± 0.56 | 50.42 ± 0.39 | 49.94 ± 0.46 | 69.14 ± 2.21 | 90.59 ± 0.19 |
| | 6 | 49.28 ± 0.98 | 51.12 ± 0.52 | 47.83 ± 0.87 | 49.74 ± 0.57 | 50.01 ± 0.43 | 71.52 ± 1.43 | 87.92 ± 0.29 |
| | 7 | 53.95 ± 0.97 | 56.44 ± 0.59 | 53.54 ± 0.78 | 54.45 ± 0.53 | 49.87 ± 0.78 | 77.60 ± 1.34 | 91.38 ± 0.49 |
| | 8 | 49.22 ± 1.07 | 52.60 ± 1.06 | 48.46 ± 0.80 | 49.94 ± 1.10 | 50.52 ± 1.06 | 69.52 ± 2.47 | 71.38 ± 1.43 |
| | 9 | 50.41 ± 0.85 | 53.13 ± 0.72 | 49.79 ± 0.59 | 51.10 ± 0.78 | 50.75 ± 1.37 | 73.13 ± 1.64 | 88.30 ± 0.56 |
| | *average* | 51.18 ± 0.38 | 54.33 ± 0.33 | 50.97 ± 0.23 | 52.24 ± 0.26 | 49.95 ± 0.25 | 71.24 ± 0.38 | **83.32 ± 0.46** |

Table 4: AUROC (%) when $\rho = 0.2$

| Dataset | Class name | CAE | CAE-IF | DRAE | RDAE | DAGMM | SSD-IF | $E^3$Outlier |
|---|---|---|---|---|---|---|---|---|
| MNIST | 0 | $47.40 \pm 5.97$ | $87.03 \pm 3.48$ | $57.02 \pm 9.40$ | $52.81 \pm 6.82$ | $70.39 \pm 26.55$ | $86.53 \pm 2.71$ | $89.26 \pm 0.64$ |
| | 1 | $98.77 \pm 0.21$ | $99.16 \pm 0.15$ | $90.07 \pm 12.60$ | $98.78 \pm 0.08$ | $89.85 \pm 12.77$ | $94.51 \pm 0.61$ | $87.63 \pm 1.13$ |
| | 2 | $49.47 \pm 4.52$ | $69.93 \pm 2.67$ | $51.93 \pm 7.26$ | $53.36 \pm 4.15$ | $50.75 \pm 1.82$ | $86.21 \pm 1.10$ | $90.71 \pm 0.45$ |
| | 3 | $58.64 \pm 4.61$ | $79.21 \pm 1.22$ | $71.82 \pm 10.77$ | $62.89 \pm 2.87$ | $60.84 \pm 10.12$ | $91.24 \pm 1.15$ | $96.25 \pm 0.09$ |
| | 4 | $69.64 \pm 3.14$ | $78.71 \pm 1.41$ | $70.44 \pm 5.44$ | $71.95 \pm 2.05$ | $66.32 \pm 13.11$ | $93.71 \pm 0.35$ | $92.90 \pm 0.36$ |
| | 5 | $58.58 \pm 2.31$ | $72.34 \pm 1.30$ | $57.21 \pm 4.44$ | $58.67 \pm 2.29$ | $54.22 \pm 2.29$ | $88.67 \pm 0.66$ | $88.73 \pm 0.71$ |
| | 6 | $62.92 \pm 1.65$ | $87.52 \pm 2.09$ | $63.89 \pm 8.90$ | $65.40 \pm 3.01$ | $72.08 \pm 15.48$ | $95.03 \pm 0.81$ | $95.75 \pm 0.12$ |
| | 7 | $78.79 \pm 3.42$ | $88.67 \pm 0.88$ | $74.75 \pm 7.94$ | $80.94 \pm 1.13$ | $76.95 \pm 8.80$ | $91.57 \pm 0.84$ | $91.76 \pm 0.29$ |
| | 8 | $44.17 \pm 2.43$ | $66.18 \pm 1.89$ | $65.30 \pm 6.30$ | $51.03 \pm 2.28$ | $55.42 \pm 11.44$ | $83.32 \pm 0.73$ | $86.00 \pm 0.80$ |
| | 9 | $71.28 \pm 1.91$ | $86.47 \pm 1.87$ | $70.10 \pm 5.64$ | $74.20 \pm 1.39$ | $61.99 \pm 9.79$ | $93.86 \pm 0.66$ | $94.14 \pm 0.20$ |
| | *average* | $63.97 \pm 1.00$ | $81.52 \pm 0.86$ | $67.25 \pm 2.07$ | $67.00 \pm 0.69$ | $65.88 \pm 2.89$ | $90.46 \pm 0.28$ | $\mathbf{91.31 \pm 0.16}$ |
| Fashion-MNIST | t-shirt | $52.15 \pm 7.19$ | $82.04 \pm 1.37$ | $60.33 \pm 4.96$ | $70.86 \pm 2.05$ | $62.36 \pm 19.64$ | $86.52 \pm 0.65$ | $90.76 \pm 0.48$ |
| | trouser | $94.83 \pm 0.63$ | $94.72 \pm 0.27$ | $90.36 \pm 4.71$ | $95.39 \pm 0.93$ | $77.39 \pm 14.51$ | $96.88 \pm 0.45$ | $96.68 \pm 0.18$ |
| | pullover | $57.76 \pm 2.09$ | $79.12 \pm 1.32$ | $59.91 \pm 2.98$ | $68.59 \pm 3.36$ | $41.03 \pm 9.34$ | $87.98 \pm 0.42$ | $91.37 \pm 0.48$ |
| | dress | $69.04 \pm 4.36$ | $80.17 \pm 0.87$ | $75.28 \pm 1.84$ | $74.12 \pm 2.56$ | $82.69 \pm 3.28$ | $86.21 \pm 1.13$ | $84.58 \pm 0.47$ |
| | coat | $65.55 \pm 3.46$ | $80.29 \pm 1.56$ | $58.62 \pm 2.87$ | $67.29 \pm 5.13$ | $46.52 \pm 18.72$ | $89.05 \pm 0.45$ | $90.61 \pm 0.51$ |
| | sandal | $62.53 \pm 6.39$ | $59.14 \pm 1.43$ | $69.98 \pm 3.09$ | $62.00 \pm 6.63$ | $83.05 \pm 4.82$ | $83.26 \pm 2.43$ | $91.09 \pm 0.85$ |
| | shirt | $50.44 \pm 2.45$ | $74.77 \pm 0.56$ | $52.79 \pm 2.50$ | $60.30 \pm 2.39$ | $53.22 \pm 9.40$ | $79.89 \pm 0.88$ | $82.48 \pm 0.37$ |
| | sneaker | $82.19 \pm 3.95$ | $88.60 \pm 0.41$ | $78.41 \pm 15.06$ | $88.90 \pm 1.83$ | $84.25 \pm 8.10$ | $97.79 \pm 0.16$ | $98.84 \pm 0.14$ |
| | bag | $42.24 \pm 2.63$ | $57.68 \pm 1.73$ | $42.94 \pm 1.61$ | $44.93 \pm 3.49$ | $56.85 \pm 15.87$ | $71.45 \pm 0.97$ | $86.17 \pm 0.77$ |
| | ankle-boot | $67.66 \pm 4.45$ | $81.51 \pm 1.88$ | $68.61 \pm 6.88$ | $77.06 \pm 3.55$ | $72.69 \pm 16.26$ | $97.33 \pm 0.26$ | $99.19 \pm 0.03$ |
| | *average* | $64.44 \pm 1.81$ | $77.81 \pm 0.61$ | $65.72 \pm 1.93$ | $70.94 \pm 1.13$ | $66.00 \pm 4.96$ | $87.64 \pm 0.28$ | $\mathbf{91.18 \pm 0.15}$ |
| CIFAR10 | airplane | $65.69 \pm 6.20$ | $61.96 \pm 0.66$ | $71.06 \pm 0.78$ | $63.37 \pm 4.40$ | $49.73 \pm 6.13$ | $51.10 \pm 2.74$ | $75.91 \pm 0.40$ |
| | automobile | $39.08 \pm 4.45$ | $32.72 \pm 0.94$ | $37.67 \pm 1.04$ | $31.94 \pm 1.36$ | $61.93 \pm 2.17$ | $66.55 \pm 3.87$ | $92.74 \pm 0.59$ |
| | bird | $62.61 \pm 2.37$ | $65.86 \pm 0.69$ | $65.49 \pm 0.57$ | $67.51 \pm 0.82$ | $47.19 \pm 5.58$ | $53.55 \pm 2.95$ | $66.29 \pm 1.05$ |
| | cat | $58.02 \pm 2.31$ | $51.92 \pm 0.86$ | $58.95 \pm 0.44$ | $54.70 \pm 2.24$ | $53.12 \pm 1.50$ | $53.53 \pm 3.16$ | $61.47 \pm 2.24$ |
| | deer | $62.10 \pm 2.45$ | $69.89 \pm 0.64$ | $62.16 \pm 0.93$ | $70.11 \pm 0.72$ | $52.27 \pm 5.44$ | $61.49 \pm 2.07$ | $78.33 \pm 1.00$ |
| | dog | $55.38 \pm 2.20$ | $50.49 \pm 0.67$ | $59.97 \pm 0.36$ | $54.66 \pm 0.72$ | $54.45 \pm 4.28$ | $60.45 \pm 1.55$ | $77.15 \pm 1.95$ |
| | frog | $49.61 \pm 4.28$ | $61.16 \pm 0.35$ | $42.92 \pm 1.63$ | $53.48 \pm 2.24$ | $57.74 \pm 6.99$ | $61.52 \pm 1.93$ | $75.79 \pm 1.52$ |
| | horse | $45.86 \pm 1.46$ | $45.20 \pm 0.77$ | $48.87 \pm 0.70$ | $46.02 \pm 0.34$ | $53.26 \pm 8.04$ | $64.63 \pm 2.51$ | $88.67 \pm 0.63$ |
| | ship | $70.17 \pm 4.04$ | $66.56 \pm 0.48$ | $71.49 \pm 0.92$ | $67.67 \pm 4.57$ | $51.41 \pm 8.13$ | $64.93 \pm 2.95$ | $89.85 \pm 0.58$ |
| | truck | $38.23 \pm 2.60$ | $31.77 \pm 0.69$ | $37.85 \pm 0.65$ | $32.86 \pm 1.75$ | $65.72 \pm 6.64$ | $64.38 \pm 1.94$ | $86.95 \pm 0.46$ |
| | *average* | $54.67 \pm 1.18$ | $53.75 \pm 0.10$ | $55.64 \pm 0.32$ | $54.23 \pm 0.89$ | $54.68 \pm 0.70$ | $60.21 \pm 0.82$ | $\mathbf{79.32 \pm 0.16}$ |
| CIFAR100 | aquatic mammals | $65.03 \pm 2.19$ | $61.19 \pm 1.11$ | $66.23 \pm 1.83$ | $64.99 \pm 0.88$ | $51.91 \pm 3.85$ | $54.54 \pm 2.02$ | $74.42 \pm 1.20$ |
| | fish | $63.52 \pm 2.50$ | $58.95 \pm 0.79$ | $63.23 \pm 0.99$ | $60.76 \pm 2.44$ | $45.19 \pm 2.85$ | $55.70 \pm 1.81$ | $68.56 \pm 0.91$ |
| | flowers | $35.24 \pm 2.94$ | $34.65 \pm 0.81$ | $33.00 \pm 1.40$ | $36.57 \pm 4.25$ | $62.87 \pm 5.93$ | $53.80 \pm 1.25$ | $77.16 \pm 1.34$ |
| | food containers | $60.68 \pm 2.02$ | $60.27 \pm 0.65$ | $60.56 \pm 0.73$ | $58.81 \pm 1.93$ | $42.55 \pm 3.33$ | $48.15 \pm 2.13$ | $76.41 \pm 1.16$ |
| | fruit and vegetables | $51.40 \pm 4.99$ | $43.92 \pm 1.18$ | $52.71 \pm 0.69$ | $45.91 \pm 2.84$ | $60.61 \pm 3.85$ | $51.89 \pm 1.39$ | $73.18 \pm 1.06$ |
| | household electrical devices | $50.91 \pm 1.41$ | $43.34 \pm 0.49$ | $55.87 \pm 0.42$ | $45.86 \pm 2.80$ | $49.73 \pm 2.81$ | $49.49 \pm 2.28$ | $60.99 \pm 1.60$ |
| | household furniture | $58.24 \pm 2.97$ | $54.19 \pm 0.35$ | $63.86 \pm 0.59$ | $57.46 \pm 2.15$ | $47.54 \pm 3.38$ | $47.38 \pm 1.13$ | $78.01 \pm 1.65$ |
| | inserts | $48.78 \pm 2.33$ | $49.31 \pm 1.12$ | $45.02 \pm 0.37$ | $47.89 \pm 2.27$ | $49.55 \pm 2.97$ | $52.52 \pm 1.48$ | $68.89 \pm 0.86$ |
| | large carnivores | $49.30 \pm 3.77$ | $53.34 \pm 1.47$ | $52.04 \pm 1.69$ | $55.09 \pm 2.51$ | $55.76 \pm 2.22$ | $56.10 \pm 1.13$ | $83.77 \pm 1.83$ |
| | large man-made outdoor things | $65.23 \pm 1.79$ | $62.65 \pm 1.85$ | $68.63 \pm 1.52$ | $65.86 \pm 1.12$ | $59.61 \pm 7.38$ | $61.25 \pm 4.10$ | $87.65 \pm 0.55$ |
| | large natural outdoor scenes | $79.64 \pm 2.47$ | $81.86 \pm 1.04$ | $79.54 \pm 0.92$ | $80.81 \pm 2.54$ | $49.87 \pm 3.88$ | $57.45 \pm 1.92$ | $83.32 \pm 0.75$ |
| | large omnivores and herbivores | $53.34 \pm 2.10$ | $51.87 \pm 1.01$ | $58.39 \pm 1.23$ | $57.22 \pm 1.00$ | $58.69 \pm 2.23$ | $55.98 \pm 2.14$ | $80.57 \pm 0.71$ |
| | medium-sized mammals | $53.87 \pm 2.03$ | $55.62 \pm 1.74$ | $56.76 \pm 1.73$ | $61.91 \pm 2.92$ | $61.36 \pm 2.37$ | $58.52 \pm 1.34$ | $80.11 \pm 1.51$ |
| | non-insert invertebrates | $51.48 \pm 1.79$ | $56.30 \pm 1.18$ | $49.33 \pm 0.75$ | $53.20 \pm 1.36$ | $46.49 \pm 2.26$ | $50.06 \pm 1.73$ | $59.03 \pm 0.80$ |
| | people | $49.22 \pm 2.67$ | $39.42 \pm 1.51$ | $48.44 \pm 1.78$ | $43.86 \pm 1.55$ | $57.82 \pm 1.65$ | $53.03 \pm 2.78$ | $88.27 \pm 0.68$ |
| | reptiles | $52.92 \pm 0.94$ | $55.67 \pm 0.47$ | $52.08 \pm 0.92$ | $55.54 \pm 0.91$ | $50.13 \pm 1.36$ | $52.40 \pm 1.30$ | $67.33 \pm 1.33$ |
| | small mammals | $59.11 \pm 1.61$ | $62.78 \pm 0.39$ | $60.75 \pm 0.14$ | $63.06 \pm 1.19$ | $53.96 \pm 1.00$ | $56.14 \pm 1.37$ | $71.88 \pm 0.98$ |
| | trees | $58.00 \pm 3.25$ | $62.55 \pm 1.80$ | $56.64 \pm 2.11$ | $60.07 \pm 1.82$ | $59.22 \pm 3.03$ | $66.21 \pm 2.24$ | $90.16 \pm 0.33$ |
| | vehicles 1 | $34.41 \pm 3.14$ | $36.77 \pm 1.07$ | $36.92 \pm 1.25$ | $36.03 \pm 1.06$ | $59.11 \pm 1.98$ | $54.72 \pm 2.86$ | $88.46 \pm 0.30$ |
| | vehicles 2 | $48.39 \pm 2.44$ | $46.09 \pm 1.58$ | $50.29 \pm 1.48$ | $46.99 \pm 2.47$ | $51.97 \pm 3.31$ | $50.26 \pm 2.11$ | $82.85 \pm 1.05$ |
| | *average* | $54.44 \pm 0.92$ | $53.54 \pm 0.42$ | $55.51 \pm 0.34$ | $54.89 \pm 0.56$ | $53.80 \pm 0.61$ | $54.28 \pm 0.34$ | $\mathbf{77.05 \pm 0.17}$ |
| SVHN | 0 | $50.47 \pm 1.80$ | $56.47 \pm 0.90$ | $50.97 \pm 0.50$ | $52.15 \pm 0.86$ | $50.33 \pm 1.17$ | $77.10 \pm 2.23$ | $62.41 \pm 1.00$ |
| | 1 | $54.42 \pm 1.08$ | $59.45 \pm 0.63$ | $55.67 \pm 0.65$ | $55.82 \pm 0.81$ | $49.15 \pm 0.57$ | $63.49 \pm 1.81$ | $65.99 \pm 0.86$ |
| | 2 | $50.27 \pm 0.72$ | $53.34 \pm 0.85$ | $51.04 \pm 0.84$ | $51.63 \pm 0.74$ | $49.42 \pm 0.78$ | $69.23 \pm 2.88$ | $87.90 \pm 0.90$ |
| | 3 | $49.65 \pm 0.32$ | $51.32 \pm 0.76$ | $48.95 \pm 0.26$ | $50.14 \pm 0.57$ | $50.33 \pm 0.48$ | $62.81 \pm 0.89$ | $84.68 \pm 0.54$ |
| | 4 | $52.21 \pm 1.52$ | $54.99 \pm 0.55$ | $52.66 \pm 1.07$ | $53.35 \pm 0.61$ | $50.53 \pm 0.91$ | $64.27 \pm 2.93$ | $88.74 \pm 0.29$ |
| | 5 | $49.06 \pm 1.05$ | $51.97 \pm 0.53$ | $48.93 \pm 0.28$ | $50.10 \pm 0.86$ | $50.00 \pm 0.73$ | $67.25 \pm 3.35$ | $89.37 \pm 0.34$ |
| | 6 | $49.40 \pm 1.31$ | $50.95 \pm 0.78$ | $47.72 \pm 0.96$ | $49.72 \pm 0.40$ | $49.51 \pm 0.56$ | $71.80 \pm 1.24$ | $85.78 \pm 0.51$ |
| | 7 | $53.06 \pm 0.93$ | $55.60 \pm 0.86$ | $52.89 \pm 0.48$ | $53.46 \pm 1.84$ | $49.50 \pm 1.03$ | $73.55 \pm 3.72$ | $90.20 \pm 0.36$ |
| | 8 | $49.22 \pm 1.22$ | $51.86 \pm 0.46$ | $48.06 \pm 0.73$ | $49.95 \pm 0.71$ | $50.04 \pm 0.70$ | $70.70 \pm 3.10$ | $68.50 \pm 1.06$ |
| | 9 | $49.56 \pm 1.09$ | $54.22 \pm 0.32$ | $49.57 \pm 0.44$ | $51.18 \pm 0.77$ | $51.07 \pm 0.56$ | $72.19 \pm 1.58$ | $86.18 \pm 0.52$ |
| | *average* | $50.73 \pm 0.38$ | $54.02 \pm 0.20$ | $50.64 \pm 0.34$ | $51.75 \pm 0.30$ | $49.99 \pm 0.24$ | $69.24 \pm 0.83$ | $\mathbf{80.97 \pm 0.25}$ |

Table 5: AUROC (%) when $\rho = 0.25$

| Dataset | Class name | CAE | CAE-IF | DRAE | RDAE | DAGMM | SSD-IF | $E^3$Outlier |
|---|---|---|---|---|---|---|---|---|
| MNIST | 0 | $44.50 \pm 5.86$ | $83.08 \pm 2.45$ | $61.27 \pm 9.24$ | $49.53 \pm 3.85$ | $64.22 \pm 24.96$ | $84.20 \pm 1.42$ | $85.52 \pm 1.74$ |
| | 1 | $98.18 \pm 0.96$ | $98.54 \pm 0.29$ | $97.07 \pm 0.66$ | $97.75 \pm 0.95$ | $81.83 \pm 19.71$ | $93.62 \pm 1.10$ | $82.77 \pm 0.51$ |
| | 2 | $46.92 \pm 2.16$ | $64.45 \pm 2.93$ | $57.49 \pm 7.90$ | $52.42 \pm 3.46$ | $45.93 \pm 6.86$ | $82.79 \pm 1.32$ | $89.74 \pm 0.57$ |
| | 3 | $54.43 \pm 3.24$ | $75.92 \pm 1.84$ | $58.29 \pm 12.36$ | $58.66 \pm 2.83$ | $55.16 \pm 7.52$ | $89.35 \pm 0.86$ | $95.46 \pm 0.49$ |
| | 4 | $63.21 \pm 3.60$ | $77.60 \pm 1.80$ | $62.12 \pm 2.29$ | $68.82 \pm 2.90$ | $65.82 \pm 14.37$ | $91.76 \pm 0.37$ | $92.42 \pm 0.35$ |
| | 5 | $55.63 \pm 2.30$ | $71.44 \pm 1.85$ | $62.00 \pm 12.01$ | $62.65 \pm 3.27$ | $51.54 \pm 3.39$ | $84.60 \pm 1.48$ | $87.46 \pm 0.62$ |
| | 6 | $60.05 \pm 3.85$ | $84.12 \pm 1.14$ | $57.92 \pm 10.02$ | $68.12 \pm 3.35$ | $59.33 \pm 17.67$ | $93.04 \pm 0.97$ | $95.13 \pm 0.27$ |
| | 7 | $77.74 \pm 2.94$ | $87.58 \pm 1.31$ | $75.43 \pm 3.23$ | $80.65 \pm 2.67$ | $67.69 \pm 10.77$ | $90.24 \pm 1.02$ | $90.97 \pm 0.15$ |
| | 8 | $38.03 \pm 5.44$ | $67.08 \pm 2.50$ | $44.06 \pm 4.47$ | $41.77 \pm 1.98$ | $53.44 \pm 4.44$ | $81.74 \pm 1.37$ | $84.24 \pm 0.62$ |
| | 9 | $70.43 \pm 1.83$ | $85.71 \pm 1.25$ | $66.54 \pm 8.09$ | $74.04 \pm 2.06$ | $59.80 \pm 8.80$ | $93.18 \pm 0.75$ | $93.94 \pm 0.23$ |
| | *average* | $60.91 \pm 0.90$ | $79.55 \pm 0.35$ | $64.22 \pm 3.40$ | $65.44 \pm 0.47$ | $60.48 \pm 2.17$ | $88.45 \pm 0.30$ | $\mathbf{89.77 \pm 0.25}$ |
| Fashion-MNIST | t-shirt | $51.35 \pm 5.03$ | $82.35 \pm 0.65$ | $61.18 \pm 7.42$ | $65.58 \pm 4.53$ | $62.07 \pm 20.08$ | $85.13 \pm 1.59$ | $89.58 \pm 0.64$ |
| | trouser | $94.45 \pm 0.58$ | $92.85 \pm 0.85$ | $87.03 \pm 8.43$ | $95.02 \pm 0.95$ | $66.81 \pm 11.67$ | $96.06 \pm 0.33$ | $95.79 \pm 0.30$ |
| | pullover | $51.09 \pm 2.75$ | $79.38 \pm 0.85$ | $59.24 \pm 2.50$ | $67.71 \pm 2.89$ | $41.70 \pm 13.02$ | $87.12 \pm 0.60$ | $91.06 \pm 0.61$ |
| | dress | $70.15 \pm 2.27$ | $79.68 \pm 0.96$ | $73.83 \pm 1.22$ | $73.57 \pm 2.90$ | $53.85 \pm 16.78$ | $84.23 \pm 1.75$ | $83.23 \pm 0.50$ |
| | coat | $61.87 \pm 6.75$ | $78.71 \pm 0.37$ | $57.67 \pm 1.63$ | $67.58 \pm 2.44$ | $48.46 \pm 9.78$ | $89.15 \pm 0.77$ | $89.94 \pm 0.22$ |
| | sandal | $62.20 \pm 7.20$ | $57.61 \pm 2.12$ | $66.46 \pm 2.24$ | $63.48 \pm 6.44$ | $76.87 \pm 10.95$ | $79.99 \pm 2.21$ | $85.62 \pm 6.90$ |
| | shirt | $48.71 \pm 2.60$ | $73.97 \pm 0.76$ | $52.32 \pm 2.02$ | $62.51 \pm 3.12$ | $55.12 \pm 5.27$ | $78.60 \pm 0.59$ | $81.35 \pm 0.20$ |
| | sneaker | $82.02 \pm 2.40$ | $87.01 \pm 0.92$ | $85.73 \pm 3.35$ | $85.86 \pm 1.50$ | $73.12 \pm 27.97$ | $97.13 \pm 0.35$ | $98.77 \pm 0.12$ |
| | bag | $38.24 \pm 6.38$ | $55.94 \pm 1.52$ | $42.14 \pm 2.58$ | $40.97 \pm 5.54$ | $49.15 \pm 8.40$ | $64.85 \pm 3.36$ | $81.84 \pm 0.79$ |
| | ankle-boot | $68.29 \pm 5.62$ | $77.23 \pm 3.00$ | $67.48 \pm 2.54$ | $74.06 \pm 3.69$ | $56.02 \pm 26.03$ | $96.20 \pm 0.50$ | $98.94 \pm 0.04$ |
| | *average* | $62.84 \pm 1.59$ | $76.47 \pm 0.36$ | $65.31 \pm 0.73$ | $69.63 \pm 1.72$ | $58.32 \pm 9.69$ | $85.85 \pm 0.54$ | $\mathbf{89.61 \pm 0.55}$ |
| CIFAR10 | airplane | $68.18 \pm 3.42$ | $61.77 \pm 1.12$ | $70.68 \pm 1.08$ | $62.92 \pm 3.98$ | $46.75 \pm 8.26$ | $49.26 \pm 1.13$ | $73.72 \pm 1.03$ |
| | automobile | $37.69 \pm 2.33$ | $32.52 \pm 0.46$ | $37.49 \pm 0.86$ | $30.95 \pm 0.42$ | $62.80 \pm 3.80$ | $64.81 \pm 2.31$ | $91.29 \pm 0.88$ |
| | bird | $61.70 \pm 1.26$ | $65.92 \pm 0.45$ | $64.84 \pm 0.24$ | $66.72 \pm 0.58$ | $45.55 \pm 2.42$ | $52.59 \pm 1.69$ | $64.81 \pm 1.59$ |
| | cat | $58.12 \pm 0.61$ | $51.82 \pm 0.70$ | $54.44 \pm 0.48$ | $54.50 \pm 1.86$ | $52.10 \pm 3.73$ | $53.62 \pm 1.06$ | $57.06 \pm 0.93$ |
| | deer | $61.78 \pm 2.55$ | $69.98 \pm 0.44$ | $61.77 \pm 1.18$ | $68.40 \pm 0.77$ | $52.24 \pm 3.50$ | $60.73 \pm 1.08$ | $76.38 \pm 1.47$ |
| | dog | $55.10 \pm 1.15$ | $49.89 \pm 0.63$ | $59.59 \pm 0.57$ | $54.49 \pm 1.02$ | $56.66 \pm 3.16$ | $59.29 \pm 2.14$ | $75.80 \pm 1.41$ |
| | frog | $48.09 \pm 2.73$ | $61.07 \pm 0.68$ | $42.07 \pm 0.69$ | $55.68 \pm 2.32$ | $55.95 \pm 5.20$ | $60.45 \pm 2.72$ | $72.56 \pm 1.72$ |
| | horse | $45.19 \pm 1.86$ | $45.12 \pm 1.04$ | $49.00 \pm 0.87$ | $45.84 \pm 1.14$ | $56.45 \pm 4.57$ | $65.23 \pm 2.53$ | $86.74 \pm 1.46$ |
| | ship | $68.15 \pm 4.72$ | $66.19 \pm 0.44$ | $71.01 \pm 0.90$ | $68.99 \pm 2.73$ | $50.48 \pm 7.22$ | $63.22 \pm 4.36$ | $88.99 \pm 0.79$ |
| | truck | $38.15 \pm 1.39$ | $31.61 \pm 0.75$ | $37.87 \pm 0.69$ | $31.44 \pm 2.13$ | $61.64 \pm 6.80$ | $61.04 \pm 1.98$ | $86.41 \pm 0.32$ |
| | *average* | $54.21 \pm 0.72$ | $53.59 \pm 0.12$ | $55.38 \pm 0.27$ | $53.99 \pm 0.33$ | $54.06 \pm 2.00$ | $59.02 \pm 0.77$ | $\mathbf{77.37 \pm 0.20}$ |
| CIFAR100 | aquatic mammals | $63.65 \pm 1.99$ | $61.24 \pm 0.72$ | $65.99 \pm 0.81$ | $63.20 \pm 2.28$ | $48.76 \pm 3.15$ | $54.68 \pm 1.15$ | $73.16 \pm 0.87$ |
| | fish | $65.34 \pm 2.22$ | $58.84 \pm 1.31$ | $63.35 \pm 1.02$ | $61.16 \pm 2.23$ | $42.34 \pm 4.43$ | $56.13 \pm 3.24$ | $68.24 \pm 1.41$ |
| | flowers | $33.31 \pm 2.29$ | $34.94 \pm 0.93$ | $32.12 \pm 0.85$ | $36.28 \pm 3.20$ | $53.53 \pm 10.34$ | $56.32 \pm 2.77$ | $74.20 \pm 1.50$ |
| | food containers | $61.64 \pm 1.71$ | $59.15 \pm 0.81$ | $60.58 \pm 0.90$ | $54.74 \pm 3.53$ | $45.46 \pm 3.91$ | $49.07 \pm 1.96$ | $74.25 \pm 1.36$ |
| | fruit and vegetables | $53.06 \pm 4.49$ | $44.35 \pm 0.95$ | $52.14 \pm 0.78$ | $47.43 \pm 2.36$ | $59.34 \pm 3.85$ | $51.83 \pm 1.73$ | $72.70 \pm 1.01$ |
| | household electrical devices | $53.86 \pm 1.71$ | $42.74 \pm 0.84$ | $55.48 \pm 0.81$ | $45.72 \pm 2.75$ | $49.85 \pm 4.72$ | $48.92 \pm 1.57$ | $58.62 \pm 1.37$ |
| | household furniture | $61.31 \pm 0.40$ | $52.85 \pm 0.75$ | $62.89 \pm 0.92$ | $54.10 \pm 5.09$ | $50.90 \pm 6.71$ | $45.07 \pm 1.06$ | $76.86 \pm 0.87$ |
| | inserts | $46.16 \pm 2.05$ | $49.31 \pm 0.78$ | $45.08 \pm 0.77$ | $46.89 \pm 1.90$ | $46.74 \pm 3.78$ | $53.48 \pm 1.43$ | $67.86 \pm 0.70$ |
| | large carnivores | $49.06 \pm 2.19$ | $53.09 \pm 1.39$ | $51.08 \pm 0.91$ | $53.89 \pm 1.63$ | $58.98 \pm 1.78$ | $56.95 \pm 2.40$ | $81.59 \pm 1.35$ |
| | large man-made outdoor things | $63.98 \pm 5.13$ | $62.29 \pm 1.26$ | $68.30 \pm 1.36$ | $65.69 \pm 2.70$ | $62.44 \pm 2.67$ | $57.60 \pm 1.61$ | $88.01 \pm 0.51$ |
| | large natural outdoor scenes | $81.47 \pm 0.92$ | $81.20 \pm 0.25$ | $79.50 \pm 0.52$ | $80.08 \pm 1.68$ | $49.68 \pm 5.38$ | $59.53 \pm 1.74$ | $83.34 \pm 0.80$ |
| | large omnivores and herbivores | $52.35 \pm 2.08$ | $51.90 \pm 0.73$ | $56.70 \pm 0.85$ | $56.45 \pm 1.33$ | $57.75 \pm 3.62$ | $54.23 \pm 1.16$ | $80.64 \pm 1.09$ |
| | medium-sized mammals | $55.50 \pm 1.74$ | $54.86 \pm 0.82$ | $55.15 \pm 1.34$ | $58.00 \pm 1.76$ | $58.99 \pm 1.67$ | $56.15 \pm 1.40$ | $77.84 \pm 0.78$ |
| | non-insert invertebrates | $51.59 \pm 0.66$ | $56.49 \pm 0.70$ | $49.62 \pm 0.82$ | $53.54 \pm 1.77$ | $45.97 \pm 2.21$ | $49.07 \pm 1.61$ | $57.37 \pm 1.25$ |
| | people | $47.30 \pm 1.64$ | $38.96 \pm 1.39$ | $48.49 \pm 1.48$ | $43.11 \pm 2.73$ | $57.04 \pm 2.56$ | $51.35 \pm 2.03$ | $86.00 \pm 0.87$ |
| | reptiles | $52.25 \pm 1.05$ | $55.74 \pm 0.41$ | $52.07 \pm 0.71$ | $56.29 \pm 1.33$ | $49.14 \pm 3.40$ | $53.49 \pm 1.24$ | $67.28 \pm 0.72$ |
| | small mammals | $60.15 \pm 1.55$ | $62.96 \pm 0.51$ | $59.96 \pm 0.99$ | $64.62 \pm 0.67$ | $53.91 \pm 2.25$ | $54.96 \pm 1.17$ | $71.26 \pm 0.52$ |
| | trees | $58.89 \pm 3.12$ | $62.31 \pm 1.14$ | $56.51 \pm 1.33$ | $63.35 \pm 1.93$ | $59.46 \pm 1.49$ | $63.09 \pm 1.23$ | $90.00 \pm 0.48$ |
| | vehicles 1 | $33.99 \pm 0.89$ | $36.68 \pm 1.03$ | $36.75 \pm 1.52$ | $38.95 \pm 2.38$ | $56.58 \pm 2.49$ | $52.23 \pm 2.79$ | $88.20 \pm 0.56$ |
| | vehicles 2 | $47.13 \pm 2.39$ | $45.81 \pm 0.77$ | $50.12 \pm 1.11$ | $46.82 \pm 2.45$ | $49.54 \pm 2.44$ | $49.46 \pm 2.38$ | $83.05 \pm 0.95$ |
| | *average* | $54.60 \pm 0.41$ | $53.29 \pm 0.37$ | $55.09 \pm 0.31$ | $54.52 \pm 0.32$ | $52.82 \pm 1.22$ | $53.68 \pm 0.76$ | $\mathbf{76.02 \pm 0.17}$ |
| SVHN | 0 | $50.62 \pm 0.10$ | $56.24 \pm 0.72$ | $50.53 \pm 0.80$ | $52.28 \pm 0.45$ | $50.13 \pm 0.66$ | $74.71 \pm 0.83$ | $56.63 \pm 1.28$ |
| | 1 | $54.99 \pm 1.51$ | $58.86 \pm 0.51$ | $55.26 \pm 0.53$ | $55.65 \pm 1.33$ | $50.11 \pm 0.49$ | $64.13 \pm 1.88$ | $63.59 \pm 0.38$ |
| | 2 | $51.19 \pm 0.82$ | $52.71 \pm 0.85$ | $50.87 \pm 0.67$ | $51.81 \pm 0.93$ | $50.11 \pm 0.58$ | $65.21 \pm 1.07$ | $86.58 \pm 0.73$ |
| | 3 | $49.52 \pm 0.54$ | $50.71 \pm 0.56$ | $49.11 \pm 0.88$ | $49.88 \pm 0.44$ | $50.08 \pm 0.74$ | $58.98 \pm 3.55$ | $82.94 \pm 0.43$ |
| | 4 | $51.83 \pm 0.91$ | $54.57 \pm 0.50$ | $52.52 \pm 0.63$ | $53.38 \pm 0.18$ | $50.59 \pm 0.68$ | $62.90 \pm 2.15$ | $87.40 \pm 0.91$ |
| | 5 | $48.61 \pm 0.41$ | $51.68 \pm 0.63$ | $48.81 \pm 0.63$ | $49.69 \pm 0.77$ | $50.67 \pm 0.40$ | $65.47 \pm 2.63$ | $88.47 \pm 0.26$ |
| | 6 | $48.83 \pm 0.74$ | $50.65 \pm 0.80$ | $47.90 \pm 1.04$ | $49.14 \pm 0.67$ | $49.79 \pm 0.47$ | $68.29 \pm 2.31$ | $84.30 \pm 0.42$ |
| | 7 | $53.22 \pm 1.23$ | $56.07 \pm 0.67$ | $52.14 \pm 1.29$ | $53.93 \pm 0.56$ | $49.04 \pm 1.41$ | $73.25 \pm 2.03$ | $88.79 \pm 0.25$ |
| | 8 | $48.55 \pm 0.77$ | $52.08 \pm 0.41$ | $48.56 \pm 0.96$ | $50.06 \pm 0.58$ | $50.56 \pm 0.47$ | $67.52 \pm 1.78$ | $64.92 \pm 1.14$ |
| | 9 | $49.48 \pm 1.14$ | $52.53 \pm 0.94$ | $49.23 \pm 0.34$ | $50.31 \pm 0.55$ | $50.94 \pm 0.74$ | $69.56 \pm 2.29$ | $84.74 \pm 0.81$ |
| | *average* | $50.68 \pm 0.20$ | $53.61 \pm 0.17$ | $50.49 \pm 0.35$ | $51.61 \pm 0.17$ | $50.20 \pm 0.25$ | $67.00 \pm 0.71$ | $\mathbf{78.84 \pm 0.26}$ |

Table 6: AUPR-in (%) (inliers as positive class) when $\rho = 0.05$

| Dataset | Class name | CAE | CAE-IF | DRAE | RDAE | DAGMM | SSD-IF | $E^3$*Outlier* |
|---|---|---|---|---|---|---|---|---|
| MNIST | 0 | 94.08 ± 0.52 | 99.58 ± 0.06 | 95.02 ± 0.68 | 96.22 ± 0.61 | 97.23 ± 2.65 | 99.55 ± 0.07 | 99.79 ± 0.02 |
| | 1 | 99.96 ± 0.00 | 99.97 ± 0.00 | 98.53 ± 1.43 | 99.97 ± 0.00 | 99.03 ± 1.13 | 99.88 ± 0.01 | 99.82 ± 0.04 |
| | 2 | 94.89 ± 0.90 | 98.21 ± 0.26 | 96.65 ± 1.86 | 95.34 ± 0.66 | 94.85 ± 1.97 | 99.68 ± 0.04 | 99.66 ± 0.03 |
| | 3 | 95.26 ± 0.32 | 99.04 ± 0.09 | 96.05 ± 1.14 | 96.33 ± 0.54 | 94.31 ± 0.82 | 99.86 ± 0.03 | 99.87 ± 0.01 |
| | 4 | 96.26 ± 0.62 | 98.94 ± 0.30 | 96.68 ± 0.94 | 96.83 ± 0.22 | 96.49 ± 0.64 | 99.78 ± 0.04 | 99.81 ± 0.02 |
| | 5 | 95.02 ± 0.50 | 98.76 ± 0.20 | 95.35 ± 0.81 | 95.93 ± 0.37 | 95.35 ± 0.65 | 99.77 ± 0.07 | 99.62 ± 0.04 |
| | 6 | 96.53 ± 0.71 | 99.41 ± 0.09 | 96.28 ± 1.45 | 97.15 ± 0.54 | 95.10 ± 1.91 | 99.92 ± 0.02 | 99.86 ± 0.01 |
| | 7 | 98.06 ± 0.41 | 99.61 ± 0.04 | 98.21 ± 0.37 | 99.00 ± 0.14 | 97.37 ± 0.89 | 99.68 ± 0.05 | 99.67 ± 0.04 |
| | 8 | 92.59 ± 1.01 | 97.99 ± 0.19 | 93.56 ± 1.22 | 93.29 ± 0.84 | 95.46 ± 0.86 | 99.31 ± 0.21 | 99.29 ± 0.06 |
| | 9 | 97.12 ± 0.57 | 99.45 ± 0.05 | 97.57 ± 0.97 | 97.70 ± 0.53 | 96.06 ± 1.24 | 99.78 ± 0.05 | 99.74 ± 0.01 |
| | *average* | 95.98 ± 0.16 | 99.10 ± 0.04 | 96.39 ± 0.46 | 96.78 ± 0.15 | 96.13 ± 0.28 | **99.72 ± 0.03** | 99.71 ± 0.01 |
| Fashion-MNIST | t-shirt | 97.05 ± 0.59 | 99.19 ± 0.07 | 97.21 ± 0.95 | 98.43 ± 0.12 | 96.53 ± 1.91 | 99.41 ± 0.10 | 99.64 ± 0.04 |
| | trouser | 99.83 ± 0.02 | 99.82 ± 0.04 | 99.09 ± 0.44 | 99.83 ± 0.02 | 96.74 ± 2.77 | 99.92 ± 0.01 | 99.93 ± 0.01 |
| | pullover | 96.61 ± 0.99 | 98.88 ± 0.12 | 96.97 ± 0.59 | 97.89 ± 0.45 | 95.05 ± 1.85 | 99.08 ± 0.17 | 99.24 ± 0.18 |
| | dress | 98.50 ± 0.20 | 99.17 ± 0.15 | 98.24 ± 0.28 | 98.50 ± 0.27 | 97.95 ± 1.42 | 99.43 ± 0.05 | 99.54 ± 0.05 |
| | coat | 97.58 ± 0.39 | 98.70 ± 0.18 | 96.19 ± 0.45 | 97.67 ± 0.52 | 95.82 ± 1.10 | 99.19 ± 0.09 | 99.23 ± 0.15 |
| | sandal | 98.54 ± 0.45 | 97.32 ± 0.29 | 98.16 ± 0.58 | 97.56 ± 1.85 | 98.58 ± 0.52 | 99.19 ± 0.12 | 99.54 ± 0.06 |
| | shirt | 96.09 ± 0.70 | 98.53 ± 0.12 | 96.05 ± 0.32 | 97.75 ± 0.19 | 96.42 ± 0.92 | 98.47 ± 0.15 | 98.85 ± 0.10 |
| | sneaker | 99.62 ± 0.09 | 99.66 ± 0.08 | 99.42 ± 0.14 | 99.66 ± 0.05 | 86.98 ± 18.99 | 99.92 ± 0.02 | 99.95 ± 0.02 |
| | bag | 94.86 ± 0.94 | 97.40 ± 0.21 | 94.72 ± 0.40 | 96.41 ± 0.71 | 94.56 ± 1.07 | 99.45 ± 0.07 | 99.75 ± 0.03 |
| | ankle-boot | 98.98 ± 0.21 | 99.47 ± 0.07 | 98.19 ± 0.40 | 99.14 ± 0.20 | 96.83 ± 2.15 | 99.92 ± 0.01 | 99.97 ± 0.01 |
| | *average* | 97.77 ± 0.09 | 98.81 ± 0.02 | 97.42 ± 0.11 | 98.28 ± 0.15 | 95.54 ± 1.95 | 99.40 ± 0.02 | **99.56 ± 0.02** |
| CIFAR10 | airplane | 97.45 ± 0.32 | 96.74 ± 0.12 | 97.82 ± 0.09 | 96.96 ± 0.40 | 95.30 ± 1.25 | 96.04 ± 0.40 | 98.44 ± 0.16 |
| | automobile | 93.63 ± 0.43 | 90.57 ± 0.21 | 92.70 ± 0.29 | 91.03 ± 0.41 | 95.94 ± 1.23 | 97.98 ± 0.36 | 99.70 ± 0.03 |
| | bird | 96.87 ± 0.25 | 97.16 ± 0.21 | 97.05 ± 0.23 | 97.36 ± 0.32 | 94.53 ± 0.45 | 96.09 ± 0.32 | 98.25 ± 0.17 |
| | cat | 96.20 ± 0.18 | 95.04 ± 0.22 | 96.13 ± 0.23 | 95.79 ± 0.12 | 94.87 ± 0.59 | 95.89 ± 0.37 | 98.12 ± 0.06 |
| | deer | 96.06 ± 0.47 | 97.25 ± 0.30 | 96.30 ± 0.34 | 97.32 ± 0.26 | 95.26 ± 0.76 | 96.94 ± 0.41 | 98.80 ± 0.17 |
| | dog | 95.44 ± 0.34 | 94.81 ± 0.20 | 96.09 ± 0.15 | 95.30 ± 0.36 | 95.35 ± 0.42 | 96.66 ± 0.38 | 99.19 ± 0.09 |
| | frog | 94.89 ± 0.42 | 96.16 ± 0.23 | 93.84 ± 0.30 | 95.69 ± 0.41 | 96.66 ± 0.32 | 97.09 ± 0.45 | 99.00 ± 0.09 |
| | horse | 94.27 ± 0.20 | 93.69 ± 0.26 | 94.41 ± 0.25 | 94.08 ± 0.32 | 95.54 ± 0.76 | 98.21 ± 0.21 | 99.60 ± 0.07 |
| | ship | 97.69 ± 0.27 | 97.22 ± 0.14 | 97.83 ± 0.09 | 97.49 ± 0.28 | 95.70 ± 0.45 | 97.81 ± 0.28 | 99.46 ± 0.08 |
| | truck | 92.67 ± 0.21 | 90.86 ± 0.43 | 92.39 ± 0.61 | 91.39 ± 0.61 | 96.29 ± 0.54 | 97.93 ± 0.24 | 99.16 ± 0.15 |
| | *average* | 95.52 ± 0.14 | 94.95 ± 0.07 | 95.46 ± 0.09 | 95.24 ± 0.09 | 95.54 ± 0.22 | 97.06 ± 0.06 | **98.97 ± 0.04** |
| CIFAR100 | aquatic mammals | 96.50 ± 0.31 | 96.65 ± 0.27 | 96.96 ± 0.24 | 96.96 ± 0.24 | 95.03 ± 0.79 | 95.70 ± 0.42 | 98.28 ± 0.13 |
| | fish | 96.96 ± 0.26 | 96.45 ± 0.20 | 96.69 ± 0.18 | 96.65 ± 0.31 | 94.14 ± 0.65 | 95.92 ± 0.26 | 97.69 ± 0.31 |
| | flowers | 93.46 ± 0.45 | 92.20 ± 0.27 | 92.31 ± 1.12 | 92.22 ± 1.12 | 96.90 ± 1.25 | 95.67 ± 0.21 | 98.43 ± 0.24 |
| | food containers | 96.69 ± 0.54 | 96.52 ± 0.39 | 96.92 ± 0.11 | 96.66 ± 0.25 | 94.60 ± 0.53 | 95.59 ± 0.40 | 98.92 ± 0.08 |
| | fruit and vegetables | 95.91 ± 0.72 | 94.14 ± 0.30 | 95.73 ± 0.31 | 94.37 ± 0.59 | 97.14 ± 0.24 | 95.43 ± 0.57 | 98.47 ± 0.08 |
| | household electrical devices | 95.54 ± 0.20 | 94.08 ± 0.50 | 95.96 ± 0.30 | 94.78 ± 0.38 | 94.94 ± 0.31 | 94.89 ± 0.63 | 97.02 ± 0.24 |
| | household furniture | 96.41 ± 0.54 | 95.43 ± 0.35 | 97.07 ± 0.31 | 95.69 ± 0.49 | 94.99 ± 1.06 | 95.12 ± 0.40 | 98.87 ± 0.07 |
| | inserts | 94.87 ± 0.38 | 94.77 ± 0.35 | 94.42 ± 0.34 | 94.73 ± 0.28 | 95.27 ± 0.20 | 95.38 ± 0.58 | 98.22 ± 0.12 |
| | large carnivores | 94.46 ± 0.37 | 95.14 ± 0.53 | 94.78 ± 0.31 | 95.37 ± 0.63 | 96.08 ± 0.46 | 96.47 ± 0.31 | 99.18 ± 0.15 |
| | large man-made outdoor things | 96.64 ± 0.40 | 96.47 ± 0.58 | 97.25 ± 0.57 | 96.72 ± 0.68 | 95.95 ± 1.06 | 97.27 ± 0.16 | 99.17 ± 0.24 |
| | large natural outdoor scenes | 98.86 ± 0.11 | 98.74 ± 0.23 | 98.68 ± 0.09 | 98.70 ± 0.15 | 95.23 ± 1.16 | 96.71 ± 0.30 | 99.10 ± 0.06 |
| | large omnivores and herbivores | 94.42 ± 0.40 | 94.81 ± 0.57 | 95.17 ± 0.34 | 95.39 ± 0.59 | 95.80 ± 0.23 | 95.88 ± 0.14 | 98.91 ± 0.03 |
| | medium-sized mammals | 95.93 ± 0.31 | 95.76 ± 0.28 | 96.20 ± 0.43 | 96.36 ± 0.46 | 96.23 ± 0.41 | 96.03 ± 0.16 | 98.92 ± 0.11 |
| | non-insect invertebrates | 95.76 ± 0.21 | 96.36 ± 0.23 | 95.51 ± 0.15 | 96.11 ± 0.18 | 94.95 ± 0.41 | 95.02 ± 0.20 | 96.96 ± 0.21 |
| | people | 94.11 ± 0.31 | 92.71 ± 0.56 | 94.25 ± 0.30 | 93.15 ± 0.51 | 95.96 ± 0.38 | 95.79 ± 0.56 | 99.45 ± 0.06 |
| | reptiles | 95.28 ± 0.22 | 95.88 ± 0.28 | 95.27 ± 0.14 | 95.92 ± 0.31 | 94.72 ± 0.57 | 95.35 ± 0.42 | 97.41 ± 0.52 |
| | small mammals | 95.74 ± 0.50 | 96.40 ± 0.20 | 96.13 ± 0.30 | 96.60 ± 0.23 | 95.39 ± 0.68 | 96.17 ± 0.14 | 98.14 ± 0.08 |
| | trees | 96.16 ± 0.81 | 96.17 ± 0.47 | 95.46 ± 0.57 | 96.09 ± 0.45 | 95.74 ± 0.66 | 98.11 ± 0.25 | 99.52 ± 0.04 |
| | vehicles 1 | 92.11 ± 0.53 | 92.10 ± 0.39 | 92.13 ± 0.77 | 92.35 ± 0.41 | 96.00 ± 0.48 | 96.54 ± 0.37 | 99.28 ± 0.15 |
| | vehicles 2 | 94.62 ± 0.23 | 94.36 ± 0.15 | 95.38 ± 0.27 | 94.52 ± 0.19 | 95.04 ± 0.90 | 96.12 ± 0.62 | 98.89 ± 0.15 |
| | *average* | 95.52 ± 0.11 | 95.26 ± 0.08 | 95.61 ± 0.13 | 95.47 ± 0.13 | 95.51 ± 0.09 | 95.96 ± 0.09 | **98.54 ± 0.03** |
| SVHN | 0 | 95.40 ± 0.32 | 96.49 ± 0.21 | 95.51 ± 0.23 | 95.25 ± 0.12 | 94.87 ± 0.27 | 98.39 ± 0.11 | 98.44 ± 0.19 |
| | 1 | 95.64 ± 0.16 | 96.49 ± 0.09 | 95.80 ± 0.32 | 96.14 ± 0.21 | 94.71 ± 0.38 | 97.42 ± 0.35 | 98.49 ± 0.09 |
| | 2 | 95.13 ± 0.29 | 95.56 ± 0.16 | 94.94 ± 0.12 | 95.32 ± 0.16 | 94.92 ± 0.32 | 98.46 ± 0.25 | 99.62 ± 0.03 |
| | 3 | 94.95 ± 0.17 | 95.32 ± 0.11 | 94.94 ± 0.23 | 95.01 ± 0.21 | 95.04 ± 0.17 | 97.84 ± 0.23 | 99.38 ± 0.05 |
| | 4 | 95.31 ± 0.17 | 96.02 ± 0.16 | 95.47 ± 0.10 | 95.50 ± 0.15 | 94.97 ± 0.31 | 98.29 ± 0.23 | 99.68 ± 0.01 |
| | 5 | 94.90 ± 0.17 | 95.48 ± 0.13 | 94.75 ± 0.30 | 95.06 ± 0.17 | 95.22 ± 0.17 | 98.23 ± 0.24 | 99.61 ± 0.03 |
| | 6 | 95.03 ± 0.20 | 95.44 ± 0.21 | 94.73 ± 0.10 | 95.03 ± 0.08 | 94.87 ± 0.21 | 98.59 ± 0.14 | 99.50 ± 0.03 |
| | 7 | 95.61 ± 0.24 | 96.03 ± 0.16 | 95.36 ± 0.22 | 95.57 ± 0.35 | 94.90 ± 0.25 | 99.03 ± 0.07 | 99.73 ± 0.01 |
| | 8 | 94.90 ± 0.25 | 95.67 ± 0.21 | 95.00 ± 0.26 | 95.17 ± 0.21 | 94.94 ± 0.29 | 97.39 ± 0.35 | 98.68 ± 0.13 |
| | 9 | 95.00 ± 0.10 | 95.41 ± 0.26 | 94.83 ± 0.14 | 95.23 ± 0.18 | 95.03 ± 0.13 | 98.66 ± 0.11 | 99.55 ± 0.05 |
| | *average* | 95.19 ± 0.06 | 95.79 ± 0.08 | 95.13 ± 0.04 | 95.33 ± 0.05 | 94.95 ± 0.09 | 98.23 ± 0.03 | **99.27 ± 0.03** |

Table 7: AUPR-in (%) (inliers as positive class) when $\rho = 0.1$

| Dataset | Class name | CAE | CAE-IF | DRAE | RDAE | DAGMM | SSD-IF | $E^3$Outlier |
|---|---|---|---|---|---|---|---|---|
| MNIST | 0 | $88.93 \pm 1.38$ | $98.76 \pm 0.31$ | $90.84 \pm 4.12$ | $90.61 \pm 1.84$ | $94.66 \pm 6.03$ | $98.84 \pm 0.14$ | $99.33 \pm 0.03$ |
| | 1 | $99.90 \pm 0.02$ | $99.92 \pm 0.01$ | $97.46 \pm 1.17$ | $99.91 \pm 0.01$ | $98.26 \pm 2.79$ | $99.60 \pm 0.05$ | $99.43 \pm 0.05$ |
| | 2 | $88.81 \pm 1.63$ | $95.66 \pm 0.40$ | $89.84 \pm 1.85$ | $90.49 \pm 0.55$ | $89.24 \pm 3.08$ | $98.92 \pm 0.16$ | $99.20 \pm 0.05$ |
| | 3 | $90.84 \pm 1.87$ | $97.43 \pm 0.47$ | $93.14 \pm 1.49$ | $91.90 \pm 0.46$ | $89.78 \pm 0.80$ | $99.47 \pm 0.12$ | $99.68 \pm 0.04$ |
| | 4 | $92.36 \pm 1.39$ | $97.28 \pm 0.54$ | $92.94 \pm 1.53$ | $93.73 \pm 0.73$ | $92.69 \pm 2.24$ | $99.46 \pm 0.12$ | $99.48 \pm 0.04$ |
| | 5 | $90.06 \pm 0.28$ | $96.87 \pm 0.27$ | $92.49 \pm 1.59$ | $91.54 \pm 0.72$ | $90.73 \pm 1.54$ | $99.02 \pm 0.22$ | $98.98 \pm 0.08$ |
| | 6 | $93.29 \pm 1.09$ | $98.88 \pm 0.13$ | $94.81 \pm 1.94$ | $93.99 \pm 0.70$ | $93.53 \pm 2.48$ | $99.72 \pm 0.01$ | $99.64 \pm 0.01$ |
| | 7 | $95.82 \pm 1.31$ | $98.95 \pm 0.14$ | $95.71 \pm 0.75$ | $97.05 \pm 0.52$ | $95.43 \pm 2.17$ | $99.14 \pm 0.19$ | $99.12 \pm 0.03$ |
| | 8 | $86.54 \pm 0.35$ | $95.59 \pm 0.76$ | $88.06 \pm 1.78$ | $87.54 \pm 0.59$ | $91.47 \pm 2.25$ | $98.29 \pm 0.24$ | $98.39 \pm 0.18$ |
| | 9 | $93.22 \pm 0.71$ | $98.67 \pm 0.20$ | $94.23 \pm 4.20$ | $93.94 \pm 0.91$ | $93.10 \pm 0.88$ | $99.46 \pm 0.09$ | $99.42 \pm 0.05$ |
| | *average* | $91.98 \pm 0.27$ | $97.80 \pm 0.12$ | $92.95 \pm 0.87$ | $93.07 \pm 0.32$ | $92.89 \pm 0.77$ | $99.19 \pm 0.05$ | $\mathbf{99.27 \pm 0.02}$ |
| Fashion-MNIST | t-shirt | $91.59 \pm 1.73$ | $98.10 \pm 0.14$ | $93.29 \pm 1.49$ | $95.38 \pm 0.92$ | $91.58 \pm 3.80$ | $98.33 \pm 0.13$ | $99.08 \pm 0.06$ |
| | trouser | $99.54 \pm 0.03$ | $99.58 \pm 0.07$ | $95.95 \pm 3.30$ | $99.66 \pm 0.05$ | $95.64 \pm 2.46$ | $99.76 \pm 0.05$ | $99.80 \pm 0.02$ |
| | pullover | $92.07 \pm 0.62$ | $97.41 \pm 0.31$ | $92.35 \pm 1.00$ | $94.38 \pm 1.33$ | $90.77 \pm 4.41$ | $97.99 \pm 0.25$ | $98.46 \pm 0.25$ |
| | dress | $96.62 \pm 0.33$ | $97.82 \pm 0.14$ | $96.22 \pm 0.71$ | $96.86 \pm 0.57$ | $93.07 \pm 8.49$ | $98.63 \pm 0.14$ | $98.65 \pm 0.13$ |
| | coat | $93.59 \pm 0.77$ | $96.97 \pm 0.24$ | $91.18 \pm 1.17$ | $94.78 \pm 0.36$ | $89.56 \pm 3.81$ | $98.29 \pm 0.07$ | $98.34 \pm 0.21$ |
| | sandal | $95.71 \pm 1.05$ | $93.97 \pm 0.68$ | $95.69 \pm 0.85$ | $95.90 \pm 0.30$ | $97.32 \pm 0.82$ | $97.95 \pm 0.20$ | $99.11 \pm 0.04$ |
| | shirt | $90.25 \pm 1.35$ | $96.69 \pm 0.20$ | $91.78 \pm 0.81$ | $94.79 \pm 0.68$ | $91.63 \pm 1.79$ | $96.79 \pm 0.18$ | $97.45 \pm 0.23$ |
| | sneaker | $98.65 \pm 0.27$ | $98.85 \pm 0.12$ | $98.72 \pm 0.21$ | $98.79 \pm 0.18$ | $94.14 \pm 8.05$ | $99.77 \pm 0.05$ | $99.88 \pm 0.02$ |
| | bag | $88.46 \pm 1.04$ | $94.00 \pm 0.27$ | $89.02 \pm 0.66$ | $90.56 \pm 1.07$ | $91.05 \pm 2.11$ | $97.25 \pm 0.36$ | $99.19 \pm 0.05$ |
| | ankle-boot | $96.91 \pm 0.55$ | $98.08 \pm 0.26$ | $94.28 \pm 1.54$ | $96.80 \pm 1.43$ | $92.28 \pm 7.00$ | $99.81 \pm 0.04$ | $99.93 \pm 0.01$ |
| | *average* | $94.34 \pm 0.25$ | $97.15 \pm 0.06$ | $93.85 \pm 0.44$ | $95.79 \pm 0.34$ | $92.70 \pm 2.92$ | $98.46 \pm 0.04$ | $\mathbf{98.99 \pm 0.04}$ |
| CIFAR10 | airplane | $94.61 \pm 1.02$ | $93.06 \pm 0.33$ | $95.35 \pm 0.26$ | $93.48 \pm 0.47$ | $90.03 \pm 1.78$ | $90.70 \pm 0.77$ | $96.56 \pm 0.20$ |
| | automobile | $86.35 \pm 0.99$ | $82.78 \pm 0.49$ | $85.35 \pm 0.28$ | $83.33 \pm 0.61$ | $92.68 \pm 0.58$ | $95.20 \pm 0.35$ | $99.12 \pm 0.04$ |
| | bird | $92.71 \pm 1.47$ | $94.24 \pm 0.41$ | $93.98 \pm 0.34$ | $94.44 \pm 0.34$ | $89.48 \pm 1.61$ | $91.17 \pm 0.72$ | $95.62 \pm 0.33$ |
| | cat | $92.23 \pm 0.61$ | $89.71 \pm 0.31$ | $92.08 \pm 0.56$ | $90.85 \pm 0.82$ | $89.96 \pm 0.26$ | $91.50 \pm 0.52$ | $95.50 \pm 0.28$ |
| | deer | $93.18 \pm 0.95$ | $94.76 \pm 0.43$ | $92.43 \pm 0.46$ | $94.99 \pm 0.46$ | $91.34 \pm 1.03$ | $93.85 \pm 0.30$ | $97.16 \pm 0.11$ |
| | dog | $91.31 \pm 0.80$ | $89.39 \pm 0.33$ | $92.10 \pm 0.23$ | $90.91 \pm 0.34$ | $91.11 \pm 0.96$ | $93.24 \pm 0.66$ | $97.86 \pm 0.16$ |
| | frog | $91.14 \pm 1.07$ | $92.43 \pm 0.44$ | $86.72 \pm 0.41$ | $91.21 \pm 0.62$ | $92.90 \pm 1.29$ | $93.75 \pm 0.73$ | $97.42 \pm 0.26$ |
| | horse | $88.14 \pm 0.18$ | $87.56 \pm 0.44$ | $88.66 \pm 0.36$ | $88.72 \pm 0.59$ | $91.63 \pm 0.89$ | $95.67 \pm 0.24$ | $98.94 \pm 0.07$ |
| | ship | $95.26 \pm 0.55$ | $94.21 \pm 0.21$ | $95.53 \pm 0.38$ | $94.37 \pm 0.46$ | $90.67 \pm 2.99$ | $94.67 \pm 0.52$ | $98.87 \pm 0.03$ |
| | truck | $85.32 \pm 1.00$ | $83.31 \pm 0.50$ | $85.15 \pm 0.57$ | $84.20 \pm 0.50$ | $93.03 \pm 1.93$ | $95.22 \pm 0.22$ | $98.29 \pm 0.19$ |
| | *average* | $91.03 \pm 0.35$ | $90.15 \pm 0.05$ | $90.73 \pm 0.06$ | $90.65 \pm 0.17$ | $91.28 \pm 0.57$ | $93.50 \pm 0.10$ | $\mathbf{97.53 \pm 0.06}$ |
| CIFAR100 | aquatic mammals | $92.97 \pm 1.02$ | $92.72 \pm 0.19$ | $93.38 \pm 0.45$ | $93.11 \pm 0.17$ | $89.71 \pm 0.89$ | $91.21 \pm 0.51$ | $96.08 \pm 0.54$ |
| | fish | $93.53 \pm 0.87$ | $92.30 \pm 0.27$ | $93.65 \pm 0.43$ | $93.00 \pm 0.74$ | $88.16 \pm 0.57$ | $91.43 \pm 0.41$ | $95.24 \pm 0.31$ |
| | flowers | $86.32 \pm 3.56$ | $85.38 \pm 0.28$ | $84.67 \pm 0.45$ | $84.70 \pm 1.65$ | $93.43 \pm 1.65$ | $91.71 \pm 0.38$ | $96.47 \pm 0.22$ |
| | food containers | $93.29 \pm 1.21$ | $92.30 \pm 0.58$ | $93.98 \pm 0.23$ | $92.81 \pm 0.56$ | $90.23 \pm 0.56$ | $90.16 \pm 0.81$ | $97.20 \pm 0.26$ |
| | fruit and vegetables | $90.71 \pm 1.36$ | $87.19 \pm 0.67$ | $91.44 \pm 0.80$ | $86.94 \pm 1.71$ | $93.07 \pm 1.75$ | $90.92 \pm 0.76$ | $96.60 \pm 0.22$ |
| | household electrical devices | $91.31 \pm 0.95$ | $88.25 \pm 0.13$ | $90.72 \pm 1.49$ | $88.76 \pm 0.62$ | $89.38 \pm 0.48$ | $89.75 \pm 0.20$ | $93.65 \pm 0.06$ |
| | household furniture | $92.62 \pm 1.13$ | $90.25 \pm 0.48$ | $93.02 \pm 0.59$ | $90.82 \pm 0.81$ | $90.65 \pm 0.91$ | $89.12 \pm 0.53$ | $97.28 \pm 0.14$ |
| | inserts | $90.07 \pm 1.26$ | $90.35 \pm 0.65$ | $88.93 \pm 0.49$ | $89.86 \pm 0.97$ | $90.29 \pm 0.56$ | $90.62 \pm 0.18$ | $95.65 \pm 0.23$ |
| | large carnivores | $90.37 \pm 1.20$ | $91.05 \pm 0.69$ | $88.66 \pm 1.04$ | $91.36 \pm 0.74$ | $92.30 \pm 0.18$ | $92.48 \pm 0.75$ | $98.08 \pm 0.22$ |
| | large man-made outdoor things | $93.27 \pm 0.96$ | $92.83 \pm 0.35$ | $93.68 \pm 0.76$ | $93.73 \pm 0.45$ | $93.59 \pm 1.71$ | $93.31 \pm 0.41$ | $98.26 \pm 0.23$ |
| | large natural outdoor scenes | $95.97 \pm 1.89$ | $97.26 \pm 0.47$ | $97.53 \pm 0.20$ | $96.92 \pm 0.71$ | $91.35 \pm 1.39$ | $93.13 \pm 0.73$ | $97.97 \pm 0.21$ |
| | large omnivores and herbivores | $89.62 \pm 0.63$ | $90.38 \pm 0.35$ | $90.05 \pm 0.23$ | $90.81 \pm 0.69$ | $91.98 \pm 0.59$ | $91.61 \pm 1.06$ | $97.62 \pm 0.17$ |
| | medium-sized mammals | $92.04 \pm 1.03$ | $91.81 \pm 0.20$ | $90.92 \pm 0.65$ | $92.34 \pm 0.67$ | $92.29 \pm 0.72$ | $92.05 \pm 0.56$ | $97.55 \pm 0.12$ |
| | non-insert invertebrates | $91.21 \pm 0.52$ | $93.00 \pm 0.29$ | $91.44 \pm 0.11$ | $91.86 \pm 0.32$ | $89.19 \pm 0.56$ | $90.00 \pm 0.40$ | $93.56 \pm 0.44$ |
| | people | $88.44 \pm 0.94$ | $85.70 \pm 0.29$ | $88.11 \pm 0.64$ | $87.44 \pm 0.50$ | $91.29 \pm 0.64$ | $91.15 \pm 0.50$ | $98.69 \pm 0.07$ |
| | reptiles | $90.88 \pm 0.50$ | $91.98 \pm 0.32$ | $90.79 \pm 0.45$ | $91.69 \pm 0.15$ | $90.04 \pm 0.42$ | $90.50 \pm 0.66$ | $94.87 \pm 0.46$ |
| | small mammals | $91.84 \pm 1.30$ | $93.06 \pm 0.36$ | $92.21 \pm 0.46$ | $93.33 \pm 0.58$ | $91.15 \pm 1.17$ | $91.56 \pm 0.20$ | $96.20 \pm 0.34$ |
| | trees | $90.85 \pm 1.79$ | $92.49 \pm 0.40$ | $90.51 \pm 0.60$ | $92.16 \pm 0.61$ | $92.17 \pm 1.63$ | $95.10 \pm 0.80$ | $98.95 \pm 0.10$ |
| | vehicles 1 | $85.29 \pm 0.39$ | $86.51 \pm 0.36$ | $84.10 \pm 0.57$ | $85.92 \pm 0.47$ | $91.61 \pm 0.44$ | $92.32 \pm 0.80$ | $98.53 \pm 0.11$ |
| | vehicles 2 | $89.13 \pm 1.39$ | $88.97 \pm 0.54$ | $89.85 \pm 0.58$ | $89.22 \pm 0.63$ | $90.53 \pm 0.69$ | $91.22 \pm 0.41$ | $97.79 \pm 0.23$ |
| | *average* | $90.99 \pm 0.39$ | $90.69 \pm 0.08$ | $90.88 \pm 0.21$ | $90.89 \pm 0.26$ | $91.12 \pm 0.23$ | $91.47 \pm 0.06$ | $\mathbf{96.81 \pm 0.06}$ |
| SVHN | 0 | $90.02 \pm 0.66$ | $92.42 \pm 0.46$ | $90.48 \pm 0.27$ | $90.69 \pm 0.34$ | $90.08 \pm 0.16$ | $96.76 \pm 0.32$ | $95.20 \pm 0.24$ |
| | 1 | $91.65 \pm 0.55$ | $92.93 \pm 0.38$ | $91.77 \pm 0.74$ | $92.18 \pm 0.52$ | $89.95 \pm 0.50$ | $94.23 \pm 0.43$ | $95.99 \pm 0.28$ |
| | 2 | $90.12 \pm 0.30$ | $91.27 \pm 0.49$ | $90.43 \pm 0.48$ | $90.43 \pm 0.33$ | $89.75 \pm 0.25$ | $96.01 \pm 0.32$ | $99.05 \pm 0.07$ |
| | 3 | $90.18 \pm 0.14$ | $90.60 \pm 0.21$ | $90.09 \pm 0.29$ | $90.30 \pm 0.28$ | $90.20 \pm 0.19$ | $94.94 \pm 0.58$ | $98.48 \pm 0.05$ |
| | 4 | $90.49 \pm 0.39$ | $91.71 \pm 0.36$ | $90.92 \pm 0.22$ | $91.46 \pm 0.30$ | $89.83 \pm 0.50$ | $95.30 \pm 0.65$ | $99.16 \pm 0.05$ |
| | 5 | $89.59 \pm 0.29$ | $90.75 \pm 0.32$ | $89.66 \pm 0.18$ | $90.02 \pm 0.19$ | $89.96 \pm 0.30$ | $96.27 \pm 0.56$ | $99.07 \pm 0.04$ |
| | 6 | $90.08 \pm 0.32$ | $90.52 \pm 0.34$ | $89.60 \pm 0.40$ | $90.19 \pm 0.18$ | $90.19 \pm 0.56$ | $96.26 \pm 0.45$ | $98.73 \pm 0.04$ |
| | 7 | $90.58 \pm 0.05$ | $91.82 \pm 0.35$ | $90.82 \pm 0.40$ | $90.67 \pm 0.36$ | $89.94 \pm 0.63$ | $96.96 \pm 0.53$ | $99.23 \pm 0.04$ |
| | 8 | $89.78 \pm 0.30$ | $91.04 \pm 0.32$ | $89.60 \pm 0.43$ | $90.03 \pm 0.48$ | $89.84 \pm 0.28$ | $95.18 \pm 0.38$ | $96.33 \pm 0.49$ |
| | 9 | $90.17 \pm 0.19$ | $91.02 \pm 0.21$ | $90.02 \pm 0.18$ | $90.43 \pm 0.24$ | $90.22 \pm 0.53$ | $96.66 \pm 0.21$ | $98.79 \pm 0.05$ |
| | *average* | $90.27 \pm 0.08$ | $91.41 \pm 0.19$ | $90.34 \pm 0.15$ | $90.64 \pm 0.14$ | $90.00 \pm 0.14$ | $95.86 \pm 0.12$ | $\mathbf{98.00 \pm 0.05}$ |

Table 8: AUPR-in (%) (inliers as positive class) when $\rho = 0.15$

| Dataset | Class name | CAE | CAE-IF | DRAE | RDAE | DAGMM | SSD-IF | $E^3$Outlier |
|---|---|---|---|---|---|---|---|---|
| MNIST | 0 | 80.78 ± 1.94 | 97.60 ± 0.46 | 86.77 ± 7.19 | 83.40 ± 2.27 | 92.69 ± 7.89 | 97.88 ± 0.63 | 98.36 ± 0.06 |
| | 1 | 99.78 ± 0.06 | 99.87 ± 0.02 | 97.44 ± 2.24 | 99.81 ± 0.03 | 93.55 ± 10.88 | 99.30 ± 0.12 | 98.44 ± 0.19 |
| | 2 | 82.33 ± 1.19 | 91.92 ± 0.74 | 85.58 ± 3.06 | 83.33 ± 1.32 | 84.61 ± 1.79 | 97.86 ± 0.15 | 98.62 ± 0.05 |
| | 3 | 84.14 ± 1.00 | 95.18 ± 0.74 | 90.32 ± 3.20 | 87.02 ± 0.93 | 86.94 ± 2.74 | 98.83 ± 0.10 | 99.44 ± 0.02 |
| | 4 | 87.48 ± 0.49 | 94.79 ± 0.34 | 91.07 ± 3.35 | 89.27 ± 0.19 | 89.82 ± 3.76 | 98.83 ± 0.17 | 98.97 ± 0.05 |
| | 5 | 84.04 ± 1.73 | 96.90 ± 0.35 | 84.95 ± 0.98 | 83.23 ± 2.22 | 86.32 ± 2.20 | 97.67 ± 0.24 | 98.15 ± 0.13 |
| | 6 | 87.22 ± 1.15 | 97.70 ± 0.48 | 87.58 ± 5.50 | 89.16 ± 0.64 | 87.78 ± 4.73 | 99.33 ± 0.04 | 99.38 ± 0.03 |
| | 7 | 91.05 ± 1.98 | 98.32 ± 0.14 | 92.80 ± 0.74 | 93.85 ± 1.04 | 88.21 ± 5.35 | 98.41 ± 0.28 | 98.45 ± 0.03 |
| | 8 | 80.19 ± 1.50 | 92.35 ± 0.67 | 87.25 ± 4.61 | 81.16 ± 1.88 | 85.54 ± 1.30 | 96.51 ± 0.33 | 97.27 ± 0.23 |
| | 9 | 89.49 ± 1.09 | 97.73 ± 0.22 | 91.10 ± 1.80 | 90.86 ± 0.91 | 89.18 ± 3.23 | 98.99 ± 0.13 | 99.01 ± 0.08 |
| | *average* | 86.65 ± 0.25 | 96.23 ± 0.12 | 89.49 ± 1.20 | 88.11 ± 0.56 | 88.47 ± 0.84 | 98.36 ± 0.11 | **98.61 ± 0.03** |
| Fashion-MNIST | t-shirt | 88.16 ± 1.92 | 96.82 ± 0.19 | 90.84 ± 0.99 | 92.77 ± 1.37 | 76.51 ± 14.59 | 97.04 ± 0.04 | 98.50 ± 0.07 |
| | trouser | 99.16 ± 0.11 | 99.21 ± 0.17 | 98.61 ± 0.34 | 99.34 ± 0.12 | 78.62 ± 18.27 | 99.55 ± 0.11 | 99.57 ± 0.02 |
| | pullover | 87.15 ± 2.83 | 95.66 ± 0.22 | 88.13 ± 2.21 | 90.43 ± 2.02 | 86.20 ± 4.64 | 96.88 ± 0.16 | 97.54 ± 0.29 |
| | dress | 94.01 ± 0.49 | 95.80 ± 0.36 | 93.83 ± 0.51 | 93.92 ± 1.41 | 86.91 ± 3.51 | 97.53 ± 0.27 | 97.35 ± 0.24 |
| | coat | 88.70 ± 0.89 | 94.94 ± 0.28 | 87.31 ± 1.28 | 89.25 ± 2.80 | 82.60 ± 10.02 | 97.16 ± 0.21 | 97.15 ± 0.12 |
| | sandal | 92.01 ± 0.61 | 89.45 ± 0.48 | 93.37 ± 0.41 | 93.19 ± 1.04 | 94.87 ± 3.58 | 96.26 ± 0.53 | 98.38 ± 0.16 |
| | shirt | 85.54 ± 1.48 | 94.50 ± 0.22 | 86.92 ± 0.78 | 90.65 ± 1.67 | 85.18 ± 2.77 | 94.80 ± 0.30 | 95.83 ± 0.14 |
| | sneaker | 97.04 ± 0.95 | 98.08 ± 0.24 | 96.99 ± 0.67 | 97.82 ± 0.46 | 72.83 ± 19.02 | 99.61 ± 0.08 | 99.79 ± 0.05 |
| | bag | 82.00 ± 3.48 | 88.51 ± 0.88 | 83.19 ± 1.29 | 82.94 ± 3.90 | 82.85 ± 3.71 | 94.20 ± 1.09 | 97.95 ± 0.22 |
| | ankle-boot | 92.70 ± 2.14 | 95.47 ± 0.98 | 91.70 ± 0.95 | 94.13 ± 1.02 | 90.41 ± 4.48 | 99.62 ± 0.06 | 99.87 ± 0.02 |
| | *average* | 90.65 ± 0.61 | 94.84 ± 0.19 | 91.09 ± 0.34 | 92.44 ± 0.28 | 83.70 ± 3.52 | 97.27 ± 0.08 | **98.19 ± 0.03** |
| CIFAR10 | airplane | 93.13 ± 1.25 | 89.65 ± 0.54 | 93.14 ± 0.23 | 90.66 ± 0.65 | 85.56 ± 2.30 | 85.80 ± 0.42 | 94.36 ± 0.09 |
| | automobile | 79.77 ± 1.23 | 75.76 ± 0.26 | 78.90 ± 0.43 | 76.39 ± 0.88 | 89.17 ± 0.56 | 92.33 ± 0.64 | 98.60 ± 0.18 |
| | bird | 88.91 ± 0.57 | 90.79 ± 0.26 | 90.37 ± 0.32 | 91.41 ± 0.27 | 83.24 ± 1.60 | 86.14 ± 0.68 | 92.48 ± 0.42 |
| | cat | 88.13 ± 0.36 | 84.87 ± 0.29 | 88.08 ± 0.51 | 86.26 ± 0.24 | 85.36 ± 0.67 | 86.77 ± 0.69 | 91.71 ± 0.48 |
| | deer | 89.00 ± 1.23 | 91.86 ± 0.31 | 88.68 ± 0.33 | 91.95 ± 0.66 | 85.72 ± 0.91 | 90.25 ± 0.64 | 95.13 ± 0.22 |
| | dog | 86.79 ± 0.68 | 84.14 ± 0.33 | 87.85 ± 0.42 | 85.96 ± 0.71 | 85.77 ± 0.76 | 89.85 ± 0.64 | 95.53 ± 0.36 |
| | frog | 85.98 ± 1.25 | 88.24 ± 0.33 | 81.41 ± 0.70 | 86.31 ± 0.77 | 88.46 ± 1.31 | 89.95 ± 0.55 | 94.97 ± 0.33 |
| | horse | 83.21 ± 0.35 | 81.75 ± 0.32 | 83.11 ± 0.51 | 82.81 ± 0.38 | 87.56 ± 1.37 | 91.65 ± 0.79 | 97.88 ± 0.21 |
| | ship | 91.91 ± 1.41 | 91.06 ± 0.39 | 93.23 ± 0.40 | 91.08 ± 2.03 | 85.83 ± 4.03 | 91.51 ± 1.23 | 97.94 ± 0.13 |
| | truck | 80.33 ± 1.79 | 76.06 ± 0.39 | 79.39 ± 0.41 | 76.87 ± 0.92 | 88.83 ± 1.55 | 91.80 ± 0.40 | 97.13 ± 0.26 |
| | *average* | 86.71 ± 0.18 | 85.42 ± 0.07 | 86.42 ± 0.12 | 85.97 ± 0.22 | 86.55 ± 0.26 | 89.61 ± 0.24 | **95.57 ± 0.09** |
| CIFAR100 | aquatic mammals | 89.58 ± 0.78 | 88.64 ± 0.63 | 89.92 ± 0.55 | 89.82 ± 0.50 | 84.42 ± 0.67 | 87.55 ± 0.69 | 93.69 ± 0.43 |
| | fish | 90.56 ± 0.49 | 88.70 ± 0.10 | 89.81 ± 0.24 | 89.09 ± 0.74 | 82.78 ± 1.68 | 87.29 ± 0.22 | 92.90 ± 0.43 |
| | flowers | 79.81 ± 1.91 | 77.46 ± 0.39 | 77.04 ± 0.90 | 77.58 ± 1.47 | 87.93 ± 3.18 | 87.22 ± 0.56 | 94.23 ± 0.76 |
| | food containers | 89.98 ± 0.75 | 88.82 ± 0.21 | 89.99 ± 0.40 | 88.40 ± 1.29 | 84.05 ± 0.51 | 86.71 ± 0.53 | 95.20 ± 0.15 |
| | fruit and vegetables | 86.66 ± 2.13 | 82.42 ± 0.46 | 86.70 ± 0.85 | 83.84 ± 2.54 | 87.39 ± 3.05 | 86.38 ± 1.08 | 94.20 ± 0.49 |
| | household electrical devices | 86.84 ± 0.82 | 82.53 ± 0.44 | 87.56 ± 0.78 | 83.05 ± 0.78 | 85.59 ± 1.39 | 85.04 ± 0.69 | 90.26 ± 0.89 |
| | household furniture | 88.31 ± 1.06 | 85.98 ± 0.95 | 90.13 ± 0.81 | 86.61 ± 1.12 | 85.47 ± 1.94 | 83.98 ± 0.55 | 95.50 ± 0.26 |
| | inserts | 86.07 ± 1.42 | 84.69 ± 0.66 | 83.11 ± 0.68 | 84.65 ± 0.55 | 84.54 ± 1.69 | 85.69 ± 0.54 | 92.90 ± 0.38 |
| | large carnivores | 83.60 ± 1.30 | 85.20 ± 1.17 | 84.29 ± 0.52 | 85.24 ± 0.34 | 87.99 ± 0.64 | 88.16 ± 0.67 | 96.71 ± 0.57 |
| | large man-made outdoor things | 89.03 ± 1.01 | 88.88 ± 0.62 | 90.99 ± 0.64 | 90.03 ± 1.02 | 89.23 ± 1.84 | 90.08 ± 1.09 | 97.07 ± 0.39 |
| | large natural outdoor scenes | 96.06 ± 0.44 | 95.83 ± 0.36 | 95.80 ± 0.09 | 95.86 ± 0.46 | 84.94 ± 1.29 | 87.66 ± 2.09 | 96.62 ± 0.17 |
| | large omnivores and herbivores | 84.52 ± 0.58 | 84.66 ± 0.42 | 86.41 ± 0.24 | 86.33 ± 0.83 | 87.90 ± 1.20 | 86.69 ± 0.33 | 95.92 ± 0.35 |
| | medium-sized mammals | 85.32 ± 2.51 | 86.27 ± 0.53 | 86.16 ± 0.53 | 87.70 ± 0.96 | 88.19 ± 0.81 | 87.71 ± 0.67 | 95.60 ± 0.45 |
| | non-insert invertebrates | 86.59 ± 0.56 | 88.75 ± 0.33 | 86.38 ± 0.34 | 87.60 ± 0.27 | 84.10 ± 0.83 | 84.73 ± 0.22 | 89.21 ± 0.44 |
| | people | 83.21 ± 1.09 | 79.23 ± 0.45 | 83.02 ± 0.58 | 81.10 ± 0.88 | 87.02 ± 1.19 | 85.68 ± 0.85 | 97.77 ± 0.28 |
| | reptiles | 85.11 ± 1.00 | 86.96 ± 0.52 | 85.47 ± 0.67 | 87.11 ± 0.29 | 85.24 ± 0.80 | 86.28 ± 0.71 | 91.50 ± 0.98 |
| | small mammals | 87.19 ± 1.29 | 88.75 ± 0.57 | 87.90 ± 0.31 | 89.74 ± 0.89 | 86.54 ± 1.07 | 88.02 ± 1.04 | 93.44 ± 0.66 |
| | trees | 87.11 ± 1.28 | 87.93 ± 0.70 | 85.30 ± 1.30 | 87.57 ± 0.57 | 88.86 ± 1.87 | 92.10 ± 0.93 | 98.29 ± 0.20 |
| | vehicles 1 | 78.41 ± 0.43 | 78.54 ± 0.50 | 78.49 ± 0.44 | 78.54 ± 0.61 | 87.76 ± 1.00 | 87.64 ± 0.92 | 97.54 ± 0.20 |
| | vehicles 2 | 84.10 ± 1.03 | 82.86 ± 0.49 | 85.59 ± 0.51 | 83.32 ± 0.29 | 85.62 ± 1.15 | 86.52 ± 0.47 | 96.28 ± 0.31 |
| | *average* | 86.40 ± 0.12 | 85.66 ± 0.16 | 86.50 ± 0.17 | 86.16 ± 0.19 | 86.28 ± 0.28 | 87.06 ± 0.16 | **94.74 ± 0.12** |
| SVHN | 0 | 85.08 ± 0.68 | 87.99 ± 0.27 | 85.83 ± 0.21 | 86.33 ± 0.49 | 85.04 ± 0.53 | 94.59 ± 0.99 | 90.34 ± 0.92 |
| | 1 | 86.55 ± 0.73 | 88.87 ± 0.27 | 87.53 ± 0.66 | 87.64 ± 0.28 | 84.60 ± 0.44 | 90.83 ± 0.77 | 92.93 ± 0.44 |
| | 2 | 85.35 ± 0.52 | 86.55 ± 0.54 | 85.59 ± 0.32 | 86.07 ± 0.47 | 84.78 ± 0.14 | 93.57 ± 1.01 | 98.25 ± 0.14 |
| | 3 | 85.01 ± 0.32 | 85.55 ± 0.22 | 84.80 ± 0.22 | 84.89 ± 0.20 | 85.05 ± 0.20 | 91.88 ± 0.83 | 97.36 ± 0.07 |
| | 4 | 85.84 ± 0.43 | 86.98 ± 0.57 | 86.55 ± 0.25 | 86.90 ± 0.51 | 84.65 ± 0.66 | 92.35 ± 0.65 | 98.31 ± 0.07 |
| | 5 | 84.74 ± 0.28 | 85.90 ± 0.27 | 84.73 ± 0.38 | 85.18 ± 0.32 | 84.79 ± 0.22 | 92.55 ± 0.69 | 98.28 ± 0.05 |
| | 6 | 84.80 ± 0.44 | 85.71 ± 0.32 | 84.26 ± 0.40 | 85.15 ± 0.35 | 76.69 ± 17.10 | 93.28 ± 0.40 | 97.63 ± 0.06 |
| | 7 | 86.73 ± 0.26 | 87.33 ± 0.22 | 86.26 ± 0.37 | 86.66 ± 0.24 | 85.15 ± 0.67 | 94.86 ± 0.33 | 98.44 ± 0.10 |
| | 8 | 84.51 ± 0.44 | 86.13 ± 0.61 | 84.38 ± 0.39 | 84.97 ± 0.50 | 85.39 ± 0.42 | 92.38 ± 0.76 | 92.25 ± 0.59 |
| | 9 | 85.01 ± 0.32 | 86.40 ± 0.40 | 84.95 ± 0.25 | 85.47 ± 0.33 | 85.22 ± 0.46 | 93.74 ± 0.56 | 97.79 ± 0.10 |
| | *average* | 85.36 ± 0.13 | 86.74 ± 0.13 | 85.49 ± 0.06 | 85.93 ± 0.14 | 84.14 ± 1.61 | 93.00 ± 0.14 | **96.16 ± 0.14** |

Table 9: AUPR-in (%) (inliers as positive class) when $\rho = 0.2$

| Dataset | Class name | CAE | CAE-IF | DRAE | RDAE | DAGMM | SSD-IF | $E^3$Outlier |
|---|---|---|---|---|---|---|---|---|
| MNIST | 0 | $74.97 \pm 2.44$ | $96.38 \pm 1.11$ | $80.59 \pm 5.14$ | $77.65 \pm 2.76$ | $88.47 \pm 11.29$ | $96.26 \pm 0.83$ | $97.04 \pm 0.23$ |
| | 1 | $99.70 \pm 0.06$ | $99.80 \pm 0.03$ | $95.74 \pm 6.62$ | $99.71 \pm 0.02$ | $96.79 \pm 4.20$ | $98.54 \pm 0.19$ | $96.48 \pm 0.38$ |
| | 2 | $76.10 \pm 1.94$ | $88.03 \pm 1.40$ | $79.41 \pm 4.76$ | $78.15 \pm 1.84$ | $80.55 \pm 0.75$ | $95.92 \pm 0.45$ | $97.62 \pm 0.13$ |
| | 3 | $80.36 \pm 2.15$ | $93.59 \pm 0.48$ | $89.29 \pm 5.38$ | $81.91 \pm 1.41$ | $84.43 \pm 4.65$ | $97.59 \pm 0.30$ | $99.06 \pm 0.02$ |
| | 4 | $84.94 \pm 2.11$ | $92.40 \pm 0.51$ | $88.24 \pm 3.26$ | $85.72 \pm 0.93$ | $86.39 \pm 5.14$ | $98.22 \pm 0.13$ | $98.12 \pm 0.11$ |
| | 5 | $79.99 \pm 1.05$ | $90.37 \pm 0.75$ | $82.44 \pm 2.56$ | $80.37 \pm 0.96$ | $81.66 \pm 1.46$ | $96.60 \pm 0.31$ | $96.93 \pm 0.26$ |
| | 6 | $82.18 \pm 0.84$ | $96.41 \pm 0.65$ | $84.39 \pm 5.21$ | $83.41 \pm 1.49$ | $88.85 \pm 6.41$ | $98.75 \pm 0.23$ | $99.00 \pm 0.02$ |
| | 7 | $89.59 \pm 1.97$ | $96.60 \pm 0.33$ | $90.03 \pm 4.31$ | $91.41 \pm 0.77$ | $90.43 \pm 3.06$ | $97.49 \pm 0.31$ | $97.62 \pm 0.14$ |
| | 8 | $73.67 \pm 1.11$ | $86.72 \pm 1.16$ | $86.62 \pm 2.82$ | $76.77 \pm 1.10$ | $82.41 \pm 5.55$ | $94.87 \pm 0.22$ | $95.68 \pm 0.33$ |
| | 9 | $85.85 \pm 0.96$ | $95.86 \pm 0.73$ | $89.33 \pm 2.23$ | $86.86 \pm 0.83$ | $84.17 \pm 4.65$ | $98.25 \pm 0.22$ | $98.52 \pm 0.08$ |
| | *average* | $82.73 \pm 0.46$ | $93.62 \pm 0.29$ | $86.61 \pm 1.16$ | $84.20 \pm 0.30$ | $86.42 \pm 1.04$ | $97.25 \pm 0.09$ | $\mathbf{97.61 \pm 0.05}$ |
| Fashion-MNIST | t-shirt | $79.47 \pm 3.87$ | $94.88 \pm 0.54$ | $85.85 \pm 2.34$ | $90.60 \pm 1.11$ | $85.04 \pm 8.64$ | $95.88 \pm 0.27$ | $97.49 \pm 0.14$ |
| | trouser | $98.61 \pm 0.28$ | $98.63 \pm 0.07$ | $97.26 \pm 1.47$ | $98.78 \pm 0.37$ | $92.16 \pm 5.52$ | $99.27 \pm 0.11$ | $99.20 \pm 0.04$ |
| | pullover | $81.64 \pm 1.83$ | $93.13 \pm 0.52$ | $83.72 \pm 1.73$ | $88.20 \pm 1.94$ | $76.63 \pm 5.58$ | $95.32 \pm 0.17$ | $96.38 \pm 0.23$ |
| | dress | $89.07 \pm 2.67$ | $93.42 \pm 0.38$ | $92.28 \pm 0.66$ | $91.01 \pm 1.22$ | $92.88 \pm 0.73$ | $95.31 \pm 0.62$ | $95.13 \pm 0.24$ |
| | coat | $85.34 \pm 2.51$ | $92.65 \pm 0.66$ | $82.21 \pm 1.53$ | $85.47 \pm 3.00$ | $77.65 \pm 8.80$ | $96.03 \pm 0.23$ | $95.90 \pm 0.36$ |
| | sandal | $85.59 \pm 3.24$ | $83.80 \pm 0.99$ | $89.40 \pm 1.67$ | $85.76 \pm 3.66$ | $93.39 \pm 2.26$ | $93.78 \pm 1.06$ | $97.27 \pm 0.34$ |
| | shirt | $79.39 \pm 1.21$ | $92.07 \pm 0.25$ | $82.74 \pm 0.78$ | $86.33 \pm 0.64$ | $82.00 \pm 4.07$ | $92.11 \pm 0.37$ | $94.17 \pm 0.18$ |
| | sneaker | $93.77 \pm 1.74$ | $96.58 \pm 0.17$ | $91.06 \pm 7.45$ | $96.70 \pm 0.68$ | $94.36 \pm 3.09$ | $99.39 \pm 0.07$ | $99.69 \pm 0.04$ |
| | bag | $75.16 \pm 0.97$ | $82.75 \pm 1.13$ | $76.55 \pm 1.18$ | $77.49 \pm 2.42$ | $84.12 \pm 6.97$ | $89.34 \pm 1.28$ | $96.16 \pm 0.20$ |
| | ankle-boot | $85.28 \pm 2.62$ | $93.90 \pm 0.74$ | $87.45 \pm 3.25$ | $91.47 \pm 1.61$ | $88.35 \pm 7.66$ | $99.27 \pm 0.07$ | $99.78 \pm 0.01$ |
| | *average* | $85.33 \pm 0.82$ | $92.18 \pm 0.30$ | $86.85 \pm 0.92$ | $89.18 \pm 0.67$ | $86.66 \pm 2.70$ | $95.57 \pm 0.13$ | $\mathbf{97.12 \pm 0.07}$ |
| CIFAR10 | airplane | $87.49 \pm 2.97$ | $85.68 \pm 0.29$ | $90.18 \pm 0.40$ | $86.53 \pm 2.07$ | $80.45 \pm 2.69$ | $80.71 \pm 1.41$ | $91.84 \pm 0.12$ |
| | automobile | $74.55 \pm 2.35$ | $69.43 \pm 0.41$ | $72.64 \pm 0.54$ | $69.39 \pm 0.59$ | $85.21 \pm 0.90$ | $87.61 \pm 1.62$ | $97.81 \pm 0.26$ |
| | bird | $85.55 \pm 1.21$ | $87.48 \pm 0.39$ | $87.19 \pm 0.26$ | $88.36 \pm 0.70$ | $78.72 \pm 3.08$ | $81.85 \pm 1.21$ | $88.46 \pm 0.60$ |
| | cat | $83.47 \pm 1.39$ | $79.75 \pm 0.43$ | $83.39 \pm 0.23$ | $80.96 \pm 0.99$ | $81.04 \pm 0.57$ | $81.64 \pm 1.59$ | $86.40 \pm 0.90$ |
| | deer | $84.68 \pm 1.46$ | $88.80 \pm 0.48$ | $84.68 \pm 0.50$ | $89.23 \pm 0.32$ | $80.45 \pm 2.05$ | $85.85 \pm 0.99$ | $92.67 \pm 0.38$ |
| | dog | $81.40 \pm 1.10$ | $78.84 \pm 0.38$ | $83.47 \pm 0.27$ | $80.85 \pm 0.42$ | $81.92 \pm 1.87$ | $84.96 \pm 0.72$ | $92.69 \pm 0.78$ |
| | frog | $79.95 \pm 1.91$ | $83.91 \pm 0.40$ | $75.20 \pm 0.91$ | $80.68 \pm 1.27$ | $82.23 \pm 3.45$ | $85.67 \pm 0.95$ | $91.92 \pm 0.69$ |
| | horse | $76.93 \pm 0.91$ | $75.62 \pm 0.38$ | $77.20 \pm 0.42$ | $76.05 \pm 0.31$ | $80.93 \pm 3.71$ | $87.15 \pm 1.26$ | $96.84 \pm 0.15$ |
| | ship | $88.52 \pm 1.89$ | $87.82 \pm 0.31$ | $90.10 \pm 0.42$ | $88.30 \pm 2.30$ | $80.45 \pm 3.22$ | $87.20 \pm 1.11$ | $96.80 \pm 0.23$ |
| | truck | $73.74 \pm 1.77$ | $69.34 \pm 0.28$ | $72.46 \pm 0.40$ | $70.08 \pm 0.85$ | $86.16 \pm 2.99$ | $86.91 \pm 0.99$ | $95.47 \pm 0.24$ |
| | *average* | $81.63 \pm 0.68$ | $80.67 \pm 0.12$ | $81.65 \pm 0.16$ | $81.04 \pm 0.44$ | $81.76 \pm 0.37$ | $84.95 \pm 0.40$ | $\mathbf{93.09 \pm 0.07}$ |
| CIFAR100 | aquatic mammals | $86.21 \pm 1.42$ | $84.67 \pm 0.73$ | $86.52 \pm 0.95$ | $86.19 \pm 0.70$ | $80.32 \pm 1.68$ | $82.09 \pm 1.06$ | $91.36 \pm 0.59$ |
| | fish | $86.59 \pm 1.63$ | $84.82 \pm 0.50$ | $86.01 \pm 0.54$ | $85.33 \pm 1.31$ | $77.60 \pm 1.29$ | $82.61 \pm 0.88$ | $89.22 \pm 0.57$ |
| | flowers | $73.48 \pm 1.73$ | $71.14 \pm 0.35$ | $70.15 \pm 0.53$ | $71.71 \pm 1.57$ | $85.61 \pm 2.77$ | $81.89 \pm 0.64$ | $92.39 \pm 0.56$ |
| | food containers | $85.89 \pm 1.07$ | $85.03 \pm 0.34$ | $86.17 \pm 0.59$ | $84.83 \pm 1.46$ | $76.56 \pm 1.36$ | $79.91 \pm 1.14$ | $92.83 \pm 0.48$ |
| | fruit and vegetables | $81.83 \pm 2.95$ | $76.13 \pm 0.80$ | $82.14 \pm 0.66$ | $77.08 \pm 1.17$ | $84.89 \pm 1.41$ | $81.00 \pm 0.83$ | $91.56 \pm 0.55$ |
| | household electrical devices | $81.54 \pm 0.72$ | $76.74 \pm 0.44$ | $83.08 \pm 0.24$ | $78.14 \pm 1.20$ | $80.02 \pm 1.48$ | $79.98 \pm 1.13$ | $85.70 \pm 0.90$ |
| | household furniture | $83.88 \pm 1.35$ | $81.34 \pm 0.39$ | $86.10 \pm 0.56$ | $82.54 \pm 1.14$ | $79.12 \pm 1.49$ | $78.48 \pm 0.44$ | $93.42 \pm 0.60$ |
| | inserts | $81.45 \pm 1.41$ | $79.14 \pm 0.84$ | $77.43 \pm 0.30$ | $79.05 \pm 1.09$ | $79.85 \pm 1.70$ | $81.06 \pm 0.79$ | $89.13 \pm 0.37$ |
| | large carnivores | $77.78 \pm 2.40$ | $79.79 \pm 0.94$ | $78.84 \pm 0.99$ | $80.69 \pm 1.68$ | $82.53 \pm 0.47$ | $82.99 \pm 0.51$ | $95.06 \pm 0.77$ |
| | large man-made outdoor things | $85.71 \pm 1.19$ | $84.70 \pm 0.85$ | $87.59 \pm 0.84$ | $86.16 \pm 0.70$ | $83.65 \pm 3.58$ | $85.35 \pm 1.67$ | $95.96 \pm 0.33$ |
| | large natural outdoor scenes | $92.62 \pm 1.64$ | $94.23 \pm 0.45$ | $93.83 \pm 0.42$ | $93.66 \pm 1.15$ | $79.56 \pm 2.36$ | $83.76 \pm 0.56$ | $95.19 \pm 0.24$ |
| | large omnivores and herbivores | $79.86 \pm 1.25$ | $79.65 \pm 0.61$ | $81.67 \pm 0.66$ | $81.93 \pm 0.38$ | $83.34 \pm 1.00$ | $82.92 \pm 1.26$ | $94.10 \pm 0.23$ |
| | medium-sized mammals | $80.67 \pm 1.43$ | $81.90 \pm 0.88$ | $81.52 \pm 0.92$ | $84.62 \pm 1.42$ | $83.98 \pm 0.80$ | $83.85 \pm 0.52$ | $93.51 \pm 0.64$ |
| | non-insert invertebrates | $81.85 \pm 0.65$ | $84.57 \pm 0.68$ | $81.29 \pm 0.42$ | $83.40 \pm 0.69$ | $78.09 \pm 1.16$ | $80.05 \pm 0.81$ | $85.12 \pm 0.45$ |
| | people | $79.03 \pm 1.22$ | $73.10 \pm 0.82$ | $77.58 \pm 0.96$ | $75.05 \pm 0.86$ | $83.46 \pm 0.74$ | $81.59 \pm 1.15$ | $96.89 \pm 0.22$ |
| | reptiles | $80.18 \pm 0.42$ | $82.50 \pm 0.48$ | $79.80 \pm 0.48$ | $82.49 \pm 0.57$ | $80.24 \pm 0.70$ | $81.10 \pm 0.72$ | $87.87 \pm 0.76$ |
| | small mammals | $82.46 \pm 1.06$ | $84.94 \pm 0.23$ | $83.51 \pm 0.16$ | $85.10 \pm 0.31$ | $81.45 \pm 0.76$ | $83.23 \pm 0.18$ | $90.23 \pm 0.38$ |
| | trees | $81.66 \pm 1.55$ | $83.32 \pm 0.85$ | $79.89 \pm 1.34$ | $81.95 \pm 0.86$ | $83.39 \pm 1.57$ | $87.89 \pm 0.78$ | $97.34 \pm 0.12$ |
| | vehicles 1 | $71.50 \pm 1.30$ | $72.21 \pm 0.36$ | $71.77 \pm 0.39$ | $71.81 \pm 0.35$ | $84.25 \pm 0.66$ | $82.13 \pm 1.36$ | $96.29 \pm 0.21$ |
| | vehicles 2 | $79.42 \pm 1.47$ | $77.37 \pm 0.72$ | $80.17 \pm 0.79$ | $78.12 \pm 0.98$ | $81.00 \pm 1.61$ | $80.52 \pm 1.12$ | $94.56 \pm 0.40$ |
| | *average* | $81.68 \pm 0.48$ | $80.86 \pm 0.18$ | $81.75 \pm 0.19$ | $81.49 \pm 0.27$ | $81.45 \pm 0.42$ | $82.12 \pm 0.16$ | $\mathbf{92.39 \pm 0.06}$ |
| SVHN | 0 | $79.72 \pm 0.88$ | $83.35 \pm 0.43$ | $80.28 \pm 0.58$ | $80.98 \pm 0.39$ | $80.18 \pm 0.62$ | $92.22 \pm 0.83$ | $83.28 \pm 0.69$ |
| | 1 | $82.16 \pm 0.24$ | $84.88 \pm 0.40$ | $82.79 \pm 0.47$ | $83.16 \pm 0.56$ | $79.27 \pm 0.41$ | $86.72 \pm 0.78$ | $89.13 \pm 0.26$ |
| | 2 | $79.92 \pm 0.43$ | $81.63 \pm 0.45$ | $80.50 \pm 0.52$ | $80.63 \pm 0.43$ | $79.39 \pm 0.49$ | $89.60 \pm 1.25$ | $97.01 \pm 0.24$ |
| | 3 | $79.77 \pm 0.20$ | $80.49 \pm 0.38$ | $79.54 \pm 0.25$ | $79.83 \pm 0.39$ | $79.94 \pm 0.24$ | $86.73 \pm 0.60$ | $95.96 \pm 0.16$ |
| | 4 | $80.81 \pm 0.77$ | $82.52 \pm 0.19$ | $81.74 \pm 0.49$ | $81.67 \pm 0.48$ | $80.16 \pm 0.74$ | $87.60 \pm 1.59$ | $97.21 \pm 0.07$ |
| | 5 | $79.38 \pm 0.39$ | $81.11 \pm 0.36$ | $79.59 \pm 0.28$ | $80.06 \pm 0.57$ | $79.90 \pm 0.23$ | $88.96 \pm 1.46$ | $97.33 \pm 0.10$ |
| | 6 | $79.69 \pm 0.70$ | $80.73 \pm 0.41$ | $78.99 \pm 0.53$ | $79.96 \pm 0.32$ | $79.85 \pm 0.22$ | $90.90 \pm 0.49$ | $96.13 \pm 0.16$ |
| | 7 | $81.79 \pm 0.56$ | $82.59 \pm 0.51$ | $81.28 \pm 0.38$ | $81.85 \pm 0.87$ | $79.79 \pm 0.49$ | $91.38 \pm 1.66$ | $97.54 \pm 0.10$ |
| | 8 | $79.41 \pm 0.58$ | $80.85 \pm 0.36$ | $79.18 \pm 0.40$ | $79.95 \pm 0.46$ | $80.19 \pm 0.45$ | $89.99 \pm 1.35$ | $87.76 \pm 0.74$ |
| | 9 | $79.58 \pm 0.38$ | $82.25 \pm 0.20$ | $79.82 \pm 0.48$ | $80.56 \pm 0.32$ | $80.31 \pm 0.21$ | $91.03 \pm 0.63$ | $96.37 \pm 0.15$ |
| | *average* | $80.22 \pm 0.21$ | $82.04 \pm 0.12$ | $80.37 \pm 0.19$ | $80.87 \pm 0.13$ | $79.90 \pm 0.09$ | $89.51 \pm 0.40$ | $\mathbf{93.77 \pm 0.10}$ |

Table 10: AUPR-in (%) (inliers as positive class) when $\rho = 0.25$

| Dataset | Class name | CAE | CAE-IF | DRAE | RDAE | DAGMM | SSD-IF | $E^3Outlier$ |
|---|---|---|---|---|---|---|---|---|
| MNIST | 0 | $68.43 \pm 2.90$ | $93.31 \pm 1.11$ | $79.59 \pm 6.89$ | $71.32 \pm 1.69$ | $82.83 \pm 13.56$ | $94.19 \pm 0.57$ | $94.53 \pm 0.80$ |
| | 1 | $99.41 \pm 0.29$ | $99.56 \pm 0.09$ | $99.08 \pm 0.22$ | $99.26 \pm 0.32$ | $91.93 \pm 8.24$ | $97.73 \pm 0.46$ | $93.21 \pm 0.28$ |
| | 2 | $69.97 \pm 0.99$ | $81.25 \pm 2.79$ | $77.27 \pm 6.77$ | $72.33 \pm 1.61$ | $72.60 \pm 4.03$ | $92.85 \pm 0.63$ | $96.55 \pm 0.27$ |
| | 3 | $73.61 \pm 1.63$ | $89.74 \pm 1.20$ | $79.89 \pm 6.58$ | $75.84 \pm 1.28$ | $76.36 \pm 3.77$ | $96.17 \pm 0.43$ | $98.51 \pm 0.16$ |
| | 4 | $77.46 \pm 1.68$ | $88.61 \pm 1.56$ | $77.39 \pm 1.09$ | $80.49 \pm 1.38$ | $82.88 \pm 6.21$ | $96.82 \pm 0.22$ | $97.38 \pm 0.12$ |
| | 5 | $74.59 \pm 1.56$ | $87.23 \pm 0.97$ | $80.98 \pm 7.39$ | $77.81 \pm 1.71$ | $75.65 \pm 2.15$ | $93.77 \pm 0.70$ | $95.53 \pm 0.20$ |
| | 6 | $76.49 \pm 1.54$ | $93.81 \pm 0.80$ | $78.78 \pm 6.15$ | $80.51 \pm 1.71$ | $79.84 \pm 8.62$ | $97.71 \pm 0.35$ | $98.53 \pm 0.08$ |
| | 7 | $86.15 \pm 2.66$ | $95.06 \pm 0.56$ | $88.77 \pm 2.40$ | $88.79 \pm 2.35$ | $83.74 \pm 4.47$ | $96.14 \pm 0.49$ | $96.57 \pm 0.10$ |
| | 8 | $65.42 \pm 2.36$ | $83.58 \pm 1.80$ | $68.20 \pm 1.99$ | $67.34 \pm 1.03$ | $75.94 \pm 1.70$ | $92.64 \pm 0.69$ | $93.53 \pm 0.38$ |
| | 9 | $81.28 \pm 1.50$ | $94.30 \pm 0.62$ | $82.02 \pm 5.55$ | $83.48 \pm 1.83$ | $79.41 \pm 4.41$ | $97.43 \pm 0.32$ | $98.02 \pm 0.08$ |
| | *average* | $77.28 \pm 0.58$ | $90.64 \pm 0.25$ | $81.20 \pm 2.15$ | $79.72 \pm 0.39$ | $80.12 \pm 1.05$ | $95.55 \pm 0.12$ | $\mathbf{96.24 \pm 0.13}$ |
| Fashion-MNIST | t-shirt | $74.97 \pm 3.25$ | $93.61 \pm 0.30$ | $82.09 \pm 4.37$ | $84.85 \pm 2.79$ | $81.34 \pm 11.32$ | $93.85 \pm 0.97$ | $96.28 \pm 0.26$ |
| | trouser | $98.11 \pm 0.26$ | $97.42 \pm 0.43$ | $93.37 \pm 5.85$ | $98.32 \pm 0.39$ | $82.72 \pm 6.82$ | $98.81 \pm 0.10$ | $98.69 \pm 0.10$ |
| | pullover | $73.55 \pm 2.51$ | $91.31 \pm 0.25$ | $78.43 \pm 2.18$ | $84.42 \pm 2.81$ | $69.98 \pm 6.86$ | $93.41 \pm 0.36$ | $95.22 \pm 0.50$ |
| | dress | $86.67 \pm 1.22$ | $91.20 \pm 0.42$ | $89.62 \pm 0.53$ | $88.87 \pm 1.90$ | $75.80 \pm 8.88$ | $92.87 \pm 1.09$ | $93.04 \pm 0.30$ |
| | coat | $79.32 \pm 5.13$ | $89.22 \pm 0.41$ | $76.50 \pm 1.24$ | $81.75 \pm 2.45$ | $74.10 \pm 6.64$ | $94.96 \pm 0.39$ | $94.24 \pm 0.15$ |
| | sandal | $80.99 \pm 5.04$ | $78.53 \pm 1.17$ | $85.25 \pm 1.69$ | $83.84 \pm 3.36$ | $88.33 \pm 5.10$ | $90.77 \pm 0.95$ | $93.72 \pm 3.79$ |
| | shirt | $73.97 \pm 2.18$ | $89.59 \pm 0.22$ | $77.18 \pm 0.98$ | $83.91 \pm 1.72$ | $78.66 \pm 2.45$ | $88.85 \pm 0.53$ | $91.85 \pm 0.19$ |
| | sneaker | $91.28 \pm 1.62$ | $94.83 \pm 0.56$ | $93.87 \pm 1.87$ | $94.50 \pm 0.71$ | $84.41 \pm 16.75$ | $98.96 \pm 0.11$ | $99.56 \pm 0.04$ |
| | bag | $67.32 \pm 3.51$ | $77.29 \pm 1.07$ | $70.34 \pm 1.87$ | $69.07 \pm 3.68$ | $73.81 \pm 5.72$ | $83.51 \pm 1.53$ | $93.29 \pm 0.27$ |
| | ankle-boot | $82.61 \pm 3.86$ | $89.30 \pm 2.15$ | $83.67 \pm 1.14$ | $87.32 \pm 2.84$ | $76.99 \pm 16.10$ | $98.68 \pm 0.18$ | $99.62 \pm 0.02$ |
| | *average* | $80.88 \pm 1.12$ | $89.23 \pm 0.19$ | $83.03 \pm 0.57$ | $85.69 \pm 1.21$ | $78.61 \pm 5.70$ | $93.47 \pm 0.22$ | $\mathbf{95.55 \pm 0.30}$ |
| CIFAR10 | airplane | $85.96 \pm 1.89$ | $81.64 \pm 0.66$ | $87.27 \pm 0.64$ | $82.48 \pm 2.27$ | $73.15 \pm 3.79$ | $74.58 \pm 0.66$ | $88.46 \pm 0.46$ |
| | automobile | $68.41 \pm 1.34$ | $63.41 \pm 0.18$ | $66.65 \pm 0.40$ | $63.01 \pm 0.12$ | $81.51 \pm 2.01$ | $83.36 \pm 0.98$ | $96.46 \pm 0.39$ |
| | bird | $80.95 \pm 0.84$ | $84.00 \pm 0.40$ | $83.27 \pm 0.22$ | $84.55 \pm 0.41$ | $72.23 \pm 1.08$ | $76.58 \pm 0.80$ | $84.27 \pm 0.96$ |
| | cat | $79.72 \pm 0.46$ | $74.83 \pm 0.39$ | $79.48 \pm 0.33$ | $76.01 \pm 1.11$ | $76.03 \pm 2.01$ | $76.88 \pm 0.73$ | $79.62 \pm 0.75$ |
| | deer | $80.72 \pm 1.83$ | $85.66 \pm 0.52$ | $80.24 \pm 0.90$ | $84.93 \pm 0.34$ | $75.17 \pm 1.77$ | $81.14 \pm 0.60$ | $89.54 \pm 0.75$ |
| | dog | $76.43 \pm 0.49$ | $73.44 \pm 0.49$ | $78.95 \pm 0.40$ | $76.09 \pm 0.47$ | $78.07 \pm 2.05$ | $80.30 \pm 1.10$ | $89.76 \pm 0.63$ |
| | frog | $74.11 \pm 1.68$ | $79.65 \pm 0.45$ | $69.21 \pm 0.26$ | $77.22 \pm 1.32$ | $76.91 \pm 3.06$ | $81.12 \pm 1.47$ | $87.59 \pm 1.17$ |
| | horse | $71.03 \pm 1.22$ | $69.98 \pm 0.59$ | $71.84 \pm 0.49$ | $70.49 \pm 0.67$ | $77.98 \pm 2.56$ | $83.49 \pm 1.22$ | $95.09 \pm 0.54$ |
| | ship | $83.84 \pm 3.15$ | $84.25 \pm 0.30$ | $87.07 \pm 0.57$ | $85.93 \pm 1.80$ | $74.81 \pm 4.09$ | $82.80 \pm 2.27$ | $95.54 \pm 0.39$ |
| | truck | $67.61 \pm 1.06$ | $63.23 \pm 0.28$ | $66.66 \pm 0.41$ | $63.40 \pm 0.99$ | $80.15 \pm 4.09$ | $81.45 \pm 1.04$ | $93.77 \pm 0.23$ |
| | *average* | $76.88 \pm 0.44$ | $76.01 \pm 0.09$ | $77.06 \pm 0.16$ | $76.41 \pm 0.20$ | $76.60 \pm 1.14$ | $80.17 \pm 0.40$ | $\mathbf{90.01 \pm 0.13}$ |
| CIFAR100 | aquatic mammals | $80.65 \pm 1.18$ | $80.59 \pm 0.58$ | $82.52 \pm 0.16$ | $81.63 \pm 1.35$ | $74.03 \pm 2.11$ | $77.40 \pm 0.69$ | $88.19 \pm 0.48$ |
| | fish | $84.10 \pm 1.13$ | $80.64 \pm 0.75$ | $82.23 \pm 0.86$ | $81.47 \pm 1.35$ | $70.81 \pm 2.32$ | $78.29 \pm 1.76$ | $86.18 \pm 0.88$ |
| | flowers | $66.40 \pm 1.94$ | $65.18 \pm 0.42$ | $63.89 \pm 0.52$ | $65.71 \pm 1.49$ | $76.80 \pm 5.94$ | $78.70 \pm 1.44$ | $88.49 \pm 0.95$ |
| | food containers | $82.73 \pm 1.29$ | $80.33 \pm 0.63$ | $82.31 \pm 0.83$ | $78.51 \pm 1.88$ | $72.55 \pm 2.73$ | $75.17 \pm 1.04$ | $89.88 \pm 0.68$ |
| | fruit and vegetables | $78.61 \pm 2.97$ | $70.72 \pm 0.72$ | $77.33 \pm 0.79$ | $72.15 \pm 1.43$ | $79.63 \pm 2.03$ | $76.06 \pm 1.09$ | $88.94 \pm 0.60$ |
| | household electrical devices | $78.89 \pm 0.79$ | $70.99 \pm 0.53$ | $78.45 \pm 0.55$ | $72.59 \pm 1.81$ | $75.05 \pm 2.06$ | $74.56 \pm 0.77$ | $79.91 \pm 1.24$ |
| | household furniture | $81.66 \pm 0.49$ | $75.85 \pm 0.55$ | $81.62 \pm 0.89$ | $76.59 \pm 2.25$ | $76.05 \pm 3.53$ | $72.34 \pm 0.91$ | $90.84 \pm 0.45$ |
| | inserts | $74.61 \pm 1.85$ | $74.07 \pm 0.72$ | $72.33 \pm 0.66$ | $73.34 \pm 1.28$ | $73.58 \pm 1.79$ | $76.52 \pm 1.02$ | $85.27 \pm 0.61$ |
| | large carnivores | $73.04 \pm 1.63$ | $74.79 \pm 0.92$ | $73.28 \pm 0.66$ | $75.18 \pm 1.04$ | $78.87 \pm 0.99$ | $78.67 \pm 1.29$ | $92.58 \pm 0.67$ |
| | large man-made outdoor things | $80.89 \pm 3.13$ | $80.51 \pm 0.86$ | $83.96 \pm 1.12$ | $81.95 \pm 1.64$ | $80.82 \pm 1.83$ | $79.63 \pm 1.30$ | $94.79 \pm 0.41$ |
| | large natural outdoor scenes | $92.06 \pm 1.08$ | $92.16 \pm 0.22$ | $92.15 \pm 0.32$ | $91.42 \pm 0.91$ | $73.87 \pm 2.97$ | $80.46 \pm 0.97$ | $93.87 \pm 0.30$ |
| | large omnivores and herbivores | $74.09 \pm 1.53$ | $74.64 \pm 0.37$ | $76.06 \pm 0.45$ | $76.89 \pm 0.88$ | $78.61 \pm 2.10$ | $77.21 \pm 0.64$ | $92.37 \pm 0.50$ |
| | medium-sized mammals | $77.08 \pm 1.62$ | $76.67 \pm 0.36$ | $75.88 \pm 1.05$ | $77.90 \pm 1.35$ | $78.31 \pm 0.80$ | $78.59 \pm 0.96$ | $90.52 \pm 0.34$ |
| | non-insert invertebrates | $77.49 \pm 0.56$ | $80.55 \pm 0.26$ | $76.73 \pm 0.27$ | $79.21 \pm 0.79$ | $72.68 \pm 0.97$ | $74.44 \pm 0.86$ | $79.89 \pm 0.91$ |
| | people | $73.07 \pm 0.92$ | $67.02 \pm 0.78$ | $72.17 \pm 0.83$ | $69.28 \pm 1.50$ | $78.92 \pm 1.18$ | $76.00 \pm 1.45$ | $95.13 \pm 0.34$ |
| | reptiles | $75.39 \pm 0.61$ | $78.13 \pm 0.38$ | $74.99 \pm 0.61$ | $78.33 \pm 1.13$ | $74.05 \pm 2.06$ | $77.22 \pm 0.77$ | $84.36 \pm 0.81$ |
| | small mammals | $78.78 \pm 1.16$ | $81.01 \pm 0.28$ | $78.69 \pm 0.67$ | $81.76 \pm 0.75$ | $76.20 \pm 1.28$ | $78.12 \pm 0.58$ | $87.37 \pm 0.35$ |
| | trees | $77.27 \pm 1.90$ | $79.19 \pm 0.68$ | $75.27 \pm 0.83$ | $79.74 \pm 1.20$ | $78.99 \pm 1.55$ | $82.92 \pm 0.60$ | $96.46 \pm 0.21$ |
| | vehicles 1 | $65.03 \pm 0.47$ | $66.23 \pm 0.45$ | $65.86 \pm 0.58$ | $67.15 \pm 1.05$ | $78.55 \pm 1.52$ | $76.27 \pm 1.68$ | $95.09 \pm 0.32$ |
| | vehicles 2 | $73.95 \pm 1.58$ | $72.03 \pm 0.40$ | $75.19 \pm 0.62$ | $72.74 \pm 1.12$ | $75.33 \pm 1.39$ | $74.97 \pm 1.48$ | $92.95 \pm 0.43$ |
| | *average* | $77.29 \pm 0.28$ | $76.06 \pm 0.16$ | $77.05 \pm 0.16$ | $76.68 \pm 0.14$ | $76.19 \pm 0.62$ | $77.18 \pm 0.41$ | $\mathbf{89.65 \pm 0.09}$ |
| SVHN | 0 | $74.76 \pm 0.25$ | $78.71 \pm 0.29$ | $75.23 \pm 0.64$ | $76.17 \pm 0.46$ | $75.05 \pm 0.44$ | $88.74 \pm 0.61$ | $75.19 \pm 0.86$ |
| | 1 | $77.71 \pm 0.93$ | $80.11 \pm 0.41$ | $78.05 \pm 0.65$ | $78.10 \pm 0.71$ | $74.43 \pm 1.42$ | $82.92 \pm 0.94$ | $84.80 \pm 0.39$ |
| | 2 | $75.15 \pm 0.76$ | $76.35 \pm 0.54$ | $75.35 \pm 0.46$ | $75.70 \pm 0.76$ | $74.93 \pm 0.34$ | $84.36 \pm 0.88$ | $95.67 \pm 0.26$ |
| | 3 | $74.64 \pm 0.34$ | $75.43 \pm 0.39$ | $74.47 \pm 0.55$ | $74.93 \pm 0.46$ | $74.88 \pm 0.52$ | $80.88 \pm 2.03$ | $94.14 \pm 0.17$ |
| | 4 | $75.86 \pm 0.60$ | $77.49 \pm 0.18$ | $76.56 \pm 0.47$ | $77.09 \pm 0.39$ | $75.65 \pm 1.16$ | $83.28 \pm 1.13$ | $95.93 \pm 0.35$ |
| | 5 | $74.19 \pm 0.38$ | $76.05 \pm 0.49$ | $74.38 \pm 0.45$ | $74.75 \pm 0.56$ | $75.11 \pm 0.15$ | $85.15 \pm 1.69$ | $96.27 \pm 0.11$ |
| | 6 | $74.32 \pm 0.49$ | $75.78 \pm 0.27$ | $73.83 \pm 0.60$ | $74.73 \pm 0.38$ | $74.82 \pm 0.18$ | $86.41 \pm 1.45$ | $94.45 \pm 0.18$ |
| | 7 | $76.76 \pm 0.73$ | $78.35 \pm 0.54$ | $76.06 \pm 0.65$ | $76.91 \pm 0.30$ | $74.18 \pm 1.23$ | $88.80 \pm 1.13$ | $96.31 \pm 0.10$ |
| | 8 | $73.82 \pm 0.46$ | $76.25 \pm 0.32$ | $74.00 \pm 0.61$ | $75.12 \pm 0.19$ | $75.21 \pm 0.59$ | $85.43 \pm 0.95$ | $81.58 \pm 0.64$ |
| | 9 | $74.68 \pm 1.03$ | $76.54 \pm 0.59$ | $74.73 \pm 0.26$ | $75.40 \pm 0.24$ | $75.99 \pm 1.41$ | $87.06 \pm 1.35$ | $94.73 \pm 0.31$ |
| | *average* | $75.19 \pm 0.21$ | $77.11 \pm 0.14$ | $75.27 \pm 0.25$ | $75.89 \pm 0.13$ | $75.02 \pm 0.32$ | $85.30 \pm 0.38$ | $\mathbf{90.91 \pm 0.13}$ |

Table 11: AUPR-out (%) (outliers as positive class) when $\rho = 0.05$

| Dataset | Class name | CAE | CAE-IF | DRAE | RDAE | DAGMM | SSD-IF | $E^3Outlier$ |
|---|---|---|---|---|---|---|---|---|
| MNIST | 0 | 10.76 ± 1.73 | 43.01 ± 3.25 | 25.43 ± 7.71 | 17.60 ± 1.79 | 36.38 ± 33.30 | 48.88 ± 3.25 | 72.36 ± 1.86 |
| | 1 | 86.89 ± 1.67 | 87.45 ± 1.63 | 47.39 ± 14.83 | 88.15 ± 2.07 | 51.42 ± 19.61 | 70.57 ± 1.54 | 69.95 ± 3.57 |
| | 2 | 9.70 ± 1.74 | 21.46 ± 2.66 | 19.68 ± 3.59 | 11.90 ± 1.68 | 6.94 ± 2.01 | 55.14 ± 1.43 | 41.99 ± 1.69 |
| | 3 | 12.49 ± 1.02 | 26.87 ± 1.62 | 17.60 ± 4.74 | 15.37 ± 1.55 | 6.51 ± 3.24 | 74.22 ± 4.28 | 76.02 ± 1.27 |
| | 4 | 22.94 ± 2.88 | 41.33 ± 6.87 | 29.56 ± 4.26 | 26.35 ± 2.06 | 14.48 ± 5.74 | 72.85 ± 4.42 | 63.10 ± 2.78 |
| | 5 | 10.69 ± 1.24 | 19.04 ± 2.36 | 12.38 ± 4.92 | 13.49 ± 1.81 | 5.69 ± 0.41 | 68.48 ± 3.37 | 48.06 ± 2.20 |
| | 6 | 20.27 ± 6.25 | 38.98 ± 5.54 | 25.57 ± 8.45 | 29.17 ± 4.54 | 8.00 ± 1.58 | 80.24 ± 3.70 | 64.37 ± 2.49 |
| | 7 | 40.18 ± 5.45 | 58.37 ± 2.90 | 32.81 ± 7.21 | 53.74 ± 3.79 | 19.84 ± 5.40 | 72.14 ± 2.21 | 56.28 ± 1.58 |
| | 8 | 7.19 ± 2.68 | 16.88 ± 2.41 | 8.38 ± 1.64 | 6.88 ± 1.18 | 20.51 ± 16.23 | 37.19 ± 5.03 | 46.40 ± 1.07 |
| | 9 | 29.14 ± 4.04 | 38.38 ± 5.60 | 25.41 ± 8.79 | 31.39 ± 3.25 | 10.09 ± 2.90 | 71.57 ± 2.57 | 59.89 ± 1.49 |
| | *average* | 25.03 ± 1.10 | 39.18 ± 0.88 | 24.42 ± 3.41 | 29.40 ± 1.39 | 17.99 ± 4.15 | **65.13 ± 0.70** | 59.84 ± 0.70 |
| Fashion-MNIST | t-shirt | 8.91 ± 1.61 | 23.62 ± 1.72 | 10.17 ± 2.31 | 12.23 ± 0.70 | 18.80 ± 10.24 | 56.19 ± 2.63 | 57.81 ± 2.01 |
| | trouser | 56.14 ± 0.84 | 66.01 ± 1.81 | 37.60 ± 4.58 | 60.02 ± 2.05 | 31.34 ± 11.95 | 73.33 ± 1.02 | 78.53 ± 2.66 |
| | pullover | 9.74 ± 1.98 | 23.28 ± 2.38 | 9.06 ± 0.99 | 11.95 ± 2.30 | 7.82 ± 2.89 | 67.15 ± 2.07 | 72.90 ± 2.37 |
| | dress | 16.79 ± 2.04 | 33.51 ± 4.96 | 21.44 ± 3.71 | 18.34 ± 1.88 | 39.37 ± 14.85 | 66.05 ± 2.50 | 58.35 ± 2.54 |
| | coat | 14.09 ± 1.12 | 31.07 ± 3.54 | 15.62 ± 1.45 | 15.92 ± 2.44 | 12.06 ± 6.65 | 68.94 ± 1.95 | 69.36 ± 1.91 |
| | sandal | 25.47 ± 4.40 | 11.65 ± 2.39 | 19.62 ± 4.53 | 26.48 ± 12.77 | 52.18 ± 12.56 | 63.91 ± 2.77 | 78.07 ± 0.80 |
| | shirt | 5.82 ± 0.88 | 16.05 ± 1.64 | 5.75 ± 0.42 | 8.46 ± 1.06 | 7.76 ± 1.60 | 54.80 ± 3.98 | 53.97 ± 2.29 |
| | sneaker | 61.69 ± 6.47 | 63.16 ± 7.42 | 64.41 ± 7.38 | 65.74 ± 2.02 | 49.66 ± 19.16 | 86.18 ± 2.84 | 94.55 ± 0.96 |
| | bag | 5.38 ± 0.48 | 12.56 ± 0.89 | 4.83 ± 0.30 | 6.46 ± 0.86 | 5.13 ± 1.52 | 36.88 ± 1.46 | 45.49 ± 2.30 |
| | ankle-boot | 34.43 ± 2.90 | 50.47 ± 4.63 | 30.86 ± 3.86 | 37.24 ± 5.05 | 28.61 ± 11.32 | 91.50 ± 1.11 | 94.94 ± 0.54 |
| | *average* | 23.85 ± 1.07 | 33.14 ± 1.12 | 21.94 ± 1.08 | 26.28 ± 1.64 | 25.27 ± 4.75 | 66.49 ± 0.63 | **70.40 ± 0.30** |
| CIFAR10 | airplane | 9.68 ± 0.81 | 7.76 ± 0.36 | 10.55 ± 0.18 | 8.17 ± 1.21 | 6.42 ± 2.94 | 6.54 ± 0.83 | 17.85 ± 1.62 |
| | automobile | 4.01 ± 0.17 | 3.49 ± 0.11 | 3.90 ± 0.21 | 3.50 ± 0.14 | 9.07 ± 2.34 | 18.67 ± 3.87 | 67.11 ± 2.00 |
| | bird | 8.75 ± 0.52 | 9.59 ± 0.67 | 9.34 ± 0.64 | 9.56 ± 0.77 | 5.20 ± 0.34 | 6.45 ± 0.45 | 11.17 ± 0.89 |
| | cat | 7.35 ± 0.71 | 6.01 ± 0.46 | 7.42 ± 0.51 | 7.14 ± 0.71 | 6.54 ± 2.06 | 6.92 ± 0.67 | 11.94 ± 0.41 |
| | deer | 9.14 ± 1.16 | 12.47 ± 1.85 | 11.18 ± 1.88 | 12.29 ± 1.87 | 7.22 ± 2.38 | 8.76 ± 0.59 | 25.35 ± 1.97 |
| | dog | 7.27 ± 0.89 | 5.63 ± 0.22 | 8.23 ± 0.55 | 7.03 ± 0.60 | 6.19 ± 1.04 | 9.14 ± 2.24 | 33.23 ± 4.49 |
| | frog | 5.27 ± 0.50 | 9.01 ± 0.30 | 5.70 ± 0.45 | 7.32 ± 1.20 | 13.36 ± 4.21 | 11.49 ± 2.50 | 26.59 ± 1.12 |
| | horse | 6.08 ± 0.72 | 5.04 ± 0.10 | 6.15 ± 0.37 | 5.78 ± 0.53 | 8.47 ± 1.53 | 19.02 ± 4.19 | 49.53 ± 3.02 |
| | ship | 14.87 ± 1.61 | 10.65 ± 0.97 | 14.14 ± 0.46 | 12.89 ± 0.85 | 8.76 ± 6.07 | 13.86 ± 2.26 | 51.70 ± 1.26 |
| | truck | 3.87 ± 0.20 | 3.50 ± 0.20 | 3.92 ± 0.21 | 3.61 ± 0.22 | 15.39 ± 4.39 | 18.78 ± 2.57 | 51.18 ± 2.17 |
| | *average* | 7.63 ± 0.31 | 7.32 ± 0.22 | 8.05 ± 0.28 | 7.73 ± 0.35 | 8.66 ± 0.82 | 11.96 ± 0.62 | **34.56 ± 1.06** |
| CIFAR100 | aquatic mammals | 11.61 ± 1.92 | 9.44 ± 1.07 | 12.89 ± 1.45 | 12.49 ± 1.76 | 6.40 ± 1.92 | 6.47 ± 1.44 | 17.48 ± 1.69 |
| | fish | 9.79 ± 1.09 | 8.04 ± 1.05 | 9.25 ± 0.75 | 8.96 ± 0.58 | 5.24 ± 0.81 | 7.92 ± 0.96 | 11.19 ± 1.38 |
| | flowers | 3.77 ± 0.21 | 3.78 ± 0.28 | 3.98 ± 0.14 | 4.17 ± 0.56 | 12.41 ± 5.66 | 6.11 ± 0.50 | 18.46 ± 2.82 |
| | food containers | 6.70 ± 0.83 | 7.01 ± 1.19 | 6.88 ± 0.49 | 6.45 ± 0.38 | 5.00 ± 0.86 | 5.08 ± 0.42 | 17.81 ± 0.71 |
| | fruit and vegetables | 5.96 ± 0.82 | 4.49 ± 0.18 | 5.73 ± 0.67 | 5.21 ± 0.85 | 10.95 ± 2.78 | 5.83 ± 0.42 | 16.65 ± 0.54 |
| | household electrical devices | 5.18 ± 0.22 | 4.26 ± 0.20 | 6.03 ± 0.45 | 5.01 ± 0.37 | 5.81 ± 0.34 | 4.83 ± 0.65 | 7.44 ± 0.90 |
| | household furniture | 7.52 ± 1.37 | 5.71 ± 0.68 | 8.35 ± 0.72 | 6.66 ± 1.02 | 6.86 ± 2.24 | 5.13 ± 0.43 | 16.27 ± 1.55 |
| | inserts | 4.44 ± 0.20 | 5.37 ± 0.44 | 4.91 ± 0.39 | 5.14 ± 0.48 | 6.67 ± 1.04 | 5.76 ± 0.40 | 14.55 ± 1.16 |
| | large carnivores | 6.14 ± 1.00 | 7.79 ± 0.75 | 7.77 ± 0.59 | 8.75 ± 1.37 | 11.40 ± 1.87 | 9.53 ± 2.48 | 36.17 ± 4.71 |
| | large man-made outdoor things | 13.40 ± 3.65 | 9.89 ± 1.62 | 15.88 ± 2.89 | 13.44 ± 2.68 | 10.14 ± 1.96 | 9.63 ± 0.82 | 36.57 ± 4.35 |
| | large natural outdoor scenes | 22.22 ± 3.94 | 23.95 ± 4.78 | 23.62 ± 3.46 | 24.15 ± 5.41 | 7.12 ± 1.54 | 8.00 ± 0.96 | 23.87 ± 1.29 |
| | large omnivores and herbivores | 8.32 ± 0.98 | 5.89 ± 0.73 | 8.93 ± 0.53 | 8.31 ± 1.05 | 9.24 ± 2.10 | 6.21 ± 0.43 | 25.93 ± 2.02 |
| | medium-sized mammals | 10.40 ± 1.33 | 7.11 ± 0.34 | 10.94 ± 1.50 | 11.18 ± 1.23 | 11.47 ± 2.64 | 8.17 ± 1.95 | 34.30 ± 3.51 |
| | non-insert invertebrates | 5.40 ± 0.50 | 5.89 ± 0.32 | 5.26 ± 0.27 | 5.61 ± 0.25 | 6.50 ± 0.95 | 5.02 ± 0.29 | 7.22 ± 0.29 |
| | people | 5.09 ± 0.40 | 4.02 ± 0.28 | 5.33 ± 0.71 | 4.86 ± 0.82 | 8.02 ± 1.08 | 6.99 ± 0.60 | 31.45 ± 3.31 |
| | reptiles | 7.45 ± 0.96 | 7.19 ± 1.13 | 7.79 ± 1.23 | 7.92 ± 1.05 | 6.08 ± 0.71 | 5.70 ± 0.49 | 13.22 ± 2.61 |
| | small mammals | 8.79 ± 0.75 | 9.34 ± 0.71 | 10.76 ± 0.86 | 11.41 ± 1.56 | 8.92 ± 1.05 | 6.72 ± 0.72 | 18.53 ± 1.34 |
| | trees | 11.81 ± 1.74 | 10.67 ± 2.04 | 13.37 ± 2.14 | 12.08 ± 1.68 | 11.04 ± 1.20 | 14.71 ± 2.55 | 40.48 ± 2.96 |
| | vehicles 1 | 3.72 ± 0.24 | 3.73 ± 0.15 | 3.94 ± 0.50 | 3.87 ± 0.36 | 8.15 ± 1.67 | 9.28 ± 0.89 | 45.14 ± 2.73 |
| | vehicles 2 | 6.23 ± 0.85 | 5.15 ± 0.27 | 7.81 ± 0.82 | 6.36 ± 0.75 | 5.70 ± 0.90 | 6.23 ± 1.11 | 20.94 ± 1.86 |
| | *average* | 8.20 ± 0.33 | 7.44 ± 0.25 | 8.97 ± 0.27 | 8.60 ± 0.17 | 8.16 ± 0.30 | 7.17 ± 0.29 | **22.68 ± 0.42** |
| SVHN | 0 | 5.22 ± 0.17 | 6.81 ± 0.38 | 5.68 ± 0.19 | 5.27 ± 0.28 | 9.04 ± 4.88 | 17.06 ± 1.93 | 21.77 ± 1.63 |
| | 1 | 6.03 ± 0.47 | 7.56 ± 0.30 | 6.08 ± 0.38 | 6.75 ± 0.30 | 10.22 ± 7.12 | 10.93 ± 1.99 | 16.22 ± 1.21 |
| | 2 | 5.28 ± 0.08 | 5.82 ± 0.24 | 5.09 ± 0.20 | 5.37 ± 0.13 | 15.45 ± 8.89 | 13.82 ± 1.34 | 28.57 ± 1.18 |
| | 3 | 5.04 ± 0.13 | 5.50 ± 0.16 | 4.94 ± 0.07 | 5.06 ± 0.12 | 17.29 ± 4.22 | 10.31 ± 1.15 | 21.63 ± 0.70 |
| | 4 | 5.55 ± 0.43 | 6.67 ± 0.47 | 5.47 ± 0.24 | 5.56 ± 0.17 | 18.69 ± 11.30 | 13.32 ± 1.53 | 32.12 ± 0.91 |
| | 5 | 4.88 ± 0.16 | 5.37 ± 0.16 | 4.76 ± 0.14 | 4.99 ± 0.17 | 22.71 ± 9.10 | 11.81 ± 1.51 | 32.91 ± 0.94 |
| | 6 | 4.84 ± 0.16 | 5.23 ± 0.13 | 4.65 ± 0.12 | 4.91 ± 0.17 | 19.19 ± 13.63 | 15.25 ± 2.48 | 31.25 ± 1.05 |
| | 7 | 6.16 ± 0.40 | 6.71 ± 0.52 | 5.73 ± 0.49 | 6.11 ± 0.31 | 20.68 ± 8.63 | 20.73 ± 1.42 | 40.84 ± 1.06 |
| | 8 | 5.03 ± 0.23 | 5.57 ± 0.15 | 4.82 ± 0.19 | 5.11 ± 0.28 | 11.09 ± 7.50 | 9.78 ± 1.26 | 16.27 ± 0.70 |
| | 9 | 5.13 ± 0.26 | 5.51 ± 0.26 | 4.93 ± 0.09 | 5.32 ± 0.29 | 13.65 ± 9.61 | 15.47 ± 1.05 | 29.86 ± 2.56 |
| | *average* | 5.32 ± 0.09 | 6.08 ± 0.10 | 5.21 ± 0.09 | 5.44 ± 0.04 | 15.80 ± 2.21 | 13.85 ± 0.49 | **27.15 ± 0.46** |

Table 12: AUPR-out (%) (outliers as positive class) when $\rho = 0.1$

| Dataset | Class name | CAE | CAE-IF | DRAE | RDAE | DAGMM | SSD-IF | $E^3Outlier$ |
|---|---|---|---|---|---|---|---|---|
| MNIST | 0 | $19.41 \pm 3.89$ | $53.24 \pm 7.39$ | $28.59 \pm 13.07$ | $21.21 \pm 3.30$ | $49.61 \pm 34.50$ | $55.13 \pm 3.27$ | $73.08 \pm 1.15$ |
| | 1 | $91.68 \pm 0.99$ | $92.27 \pm 1.34$ | $42.86 \pm 13.71$ | $92.09 \pm 0.81$ | $65.54 \pm 24.42$ | $76.55 \pm 1.75$ | $73.05 \pm 0.90$ |
| | 2 | $13.74 \pm 2.12$ | $28.26 \pm 1.66$ | $20.44 \pm 0.95$ | $18.87 \pm 1.54$ | $12.00 \pm 3.67$ | $59.02 \pm 4.19$ | $54.66 \pm 1.08$ |
| | 3 | $19.63 \pm 4.50$ | $36.55 \pm 4.67$ | $27.17 \pm 5.80$ | $22.66 \pm 2.53$ | $11.74 \pm 2.95$ | $72.30 \pm 2.88$ | $81.49 \pm 1.94$ |
| | 4 | $32.07 \pm 6.00$ | $49.19 \pm 4.80$ | $31.97 \pm 5.96$ | $40.08 \pm 2.19$ | $19.45 \pm 6.46$ | $77.80 \pm 3.21$ | $71.34 \pm 1.83$ |
| | 5 | $18.85 \pm 1.30$ | $30.42 \pm 1.57$ | $25.31 \pm 4.19$ | $22.26 \pm 3.29$ | $11.33 \pm 0.65$ | $67.89 \pm 4.01$ | $56.59 \pm 2.00$ |
| | 6 | $31.20 \pm 6.41$ | $54.11 \pm 1.96$ | $37.10 \pm 8.89$ | $33.66 \pm 4.55$ | $25.78 \pm 16.43$ | $80.49 \pm 1.03$ | $72.47 \pm 0.96$ |
| | 7 | $52.41 \pm 3.83$ | $66.39 \pm 3.74$ | $46.73 \pm 4.54$ | $55.90 \pm 3.15$ | $36.51 \pm 20.35$ | $73.55 \pm 1.16$ | $64.63 \pm 0.91$ |
| | 8 | $12.40 \pm 1.16$ | $27.29 \pm 2.61$ | $13.70 \pm 2.71$ | $13.39 \pm 1.02$ | $15.77 \pm 5.52$ | $47.96 \pm 3.61$ | $57.04 \pm 1.45$ |
| | 9 | $37.95 \pm 3.36$ | $52.03 \pm 3.51$ | $31.43 \pm 13.47$ | $38.01 \pm 4.42$ | $18.45 \pm 4.23$ | $76.41 \pm 2.50$ | $70.61 \pm 1.70$ |
| | *average* | $32.93 \pm 0.74$ | $48.98 \pm 1.06$ | $30.53 \pm 1.99$ | $35.81 \pm 0.77$ | $26.62 \pm 5.31$ | $\mathbf{68.71 \pm 1.17}$ | $67.50 \pm 0.31$ |
| Fashion-MNIST | t-shirt | $13.50 \pm 2.24$ | $34.19 \pm 2.51$ | $14.76 \pm 2.93$ | $18.19 \pm 2.04$ | $21.51 \pm 8.91$ | $59.56 \pm 2.03$ | $66.31 \pm 1.95$ |
| | trouser | $65.54 \pm 1.46$ | $74.34 \pm 2.37$ | $33.44 \pm 12.93$ | $71.64 \pm 2.00$ | $34.73 \pm 13.15$ | $78.47 \pm 3.16$ | $83.42 \pm 1.26$ |
| | pullover | $15.04 \pm 1.01$ | $34.81 \pm 2.46$ | $14.63 \pm 1.28$ | $18.97 \pm 3.00$ | $12.63 \pm 4.92$ | $69.48 \pm 2.74$ | $79.16 \pm 1.59$ |
| | dress | $23.77 \pm 1.97$ | $41.97 \pm 2.10$ | $27.16 \pm 4.02$ | $26.82 \pm 3.73$ | $47.57 \pm 15.22$ | $67.29 \pm 2.49$ | $64.01 \pm 1.65$ |
| | coat | $20.54 \pm 1.22$ | $41.62 \pm 2.48$ | $19.14 \pm 2.23$ | $26.76 \pm 1.83$ | $18.24 \pm 10.26$ | $71.59 \pm 0.57$ | $77.48 \pm 0.94$ |
| | sandal | $29.43 \pm 3.37$ | $18.10 \pm 1.63$ | $25.57 \pm 3.82$ | $24.53 \pm 2.06$ | $55.01 \pm 18.88$ | $65.13 \pm 3.10$ | $79.98 \pm 1.16$ |
| | shirt | $10.47 \pm 1.11$ | $24.61 \pm 0.76$ | $10.59 \pm 0.82$ | $14.53 \pm 1.03$ | $15.93 \pm 2.81$ | $56.96 \pm 2.12$ | $60.57 \pm 1.73$ |
| | sneaker | $65.39 \pm 3.41$ | $63.96 \pm 2.97$ | $69.15 \pm 3.79$ | $61.69 \pm 2.45$ | $56.93 \pm 22.49$ | $88.30 \pm 2.21$ | $95.79 \pm 0.24$ |
| | bag | $9.25 \pm 0.75$ | $20.11 \pm 1.05$ | $9.15 \pm 0.39$ | $10.32 \pm 0.67$ | $11.46 \pm 2.05$ | $36.25 \pm 2.75$ | $55.71 \pm 1.08$ |
| | ankle-boot | $39.58 \pm 3.68$ | $49.69 \pm 3.57$ | $31.19 \pm 3.52$ | $43.79 \pm 5.38$ | $28.86 \pm 20.98$ | $92.70 \pm 1.11$ | $96.73 \pm 0.24$ |
| | *average* | $29.25 \pm 0.13$ | $40.34 \pm 0.39$ | $25.48 \pm 1.11$ | $31.72 \pm 1.39$ | $30.29 \pm 6.31$ | $68.57 \pm 0.39$ | $\mathbf{75.91 \pm 0.24}$ |
| CIFAR10 | airplane | $18.24 \pm 2.40$ | $14.68 \pm 0.33$ | $19.65 \pm 0.29$ | $14.59 \pm 1.01$ | $10.88 \pm 3.32$ | $10.43 \pm 0.78$ | $28.53 \pm 2.68$ |
| | automobile | $7.57 \pm 0.51$ | $7.29 \pm 0.31$ | $7.75 \pm 0.19$ | $7.13 \pm 0.29$ | $19.50 \pm 5.92$ | $24.26 \pm 4.72$ | $71.61 \pm 1.21$ |
| | bird | $15.24 \pm 1.94$ | $17.00 \pm 0.55$ | $17.09 \pm 0.67$ | $16.38 \pm 1.28$ | $10.17 \pm 0.98$ | $11.17 \pm 0.98$ | $18.18 \pm 0.62$ |
| | cat | $14.14 \pm 1.28$ | $10.99 \pm 0.52$ | $14.37 \pm 0.49$ | $12.47 \pm 1.22$ | $11.18 \pm 0.80$ | $12.22 \pm 0.95$ | $19.05 \pm 0.91$ |
| | deer | $19.46 \pm 1.06$ | $22.81 \pm 0.88$ | $19.59 \pm 0.92$ | $22.29 \pm 1.91$ | $15.10 \pm 2.59$ | $17.21 \pm 1.31$ | $37.24 \pm 1.26$ |
| | dog | $14.39 \pm 1.88$ | $11.03 \pm 0.53$ | $15.53 \pm 0.83$ | $13.65 \pm 1.25$ | $12.14 \pm 1.79$ | $15.99 \pm 1.48$ | $41.31 \pm 1.67$ |
| | frog | $11.66 \pm 1.52$ | $16.82 \pm 0.75$ | $9.31 \pm 0.34$ | $13.18 \pm 0.87$ | $18.08 \pm 3.60$ | $18.84 \pm 4.38$ | $36.72 \pm 3.04$ |
| | horse | $10.95 \pm 0.59$ | $9.77 \pm 0.45$ | $11.62 \pm 0.59$ | $11.48 \pm 1.14$ | $15.10 \pm 1.68$ | $25.94 \pm 0.84$ | $58.09 \pm 3.09$ |
| | ship | $25.07 \pm 2.84$ | $18.99 \pm 1.04$ | $24.37 \pm 1.23$ | $20.77 \pm 1.67$ | $20.09 \pm 14.18$ | $20.00 \pm 2.47$ | $63.22 \pm 0.81$ |
| | truck | $7.48 \pm 0.38$ | $7.23 \pm 0.23$ | $7.80 \pm 0.20$ | $7.56 \pm 0.25$ | $24.21 \pm 7.35$ | $27.25 \pm 2.56$ | $60.21 \pm 1.78$ |
| | *average* | $14.42 \pm 0.68$ | $13.66 \pm 0.18$ | $14.71 \pm 0.11$ | $13.95 \pm 0.30$ | $15.64 \pm 1.34$ | $18.33 \pm 0.55$ | $\mathbf{43.42 \pm 0.47}$ |
| CIFAR100 | aquatic mammals | $20.76 \pm 2.73$ | $15.18 \pm 0.78$ | $21.18 \pm 1.98$ | $18.96 \pm 2.41$ | $10.73 \pm 0.96$ | $12.07 \pm 0.93$ | $27.30 \pm 2.17$ |
| | fish | $16.55 \pm 2.39$ | $13.28 \pm 0.87$ | $16.99 \pm 1.00$ | $14.76 \pm 1.28$ | $8.69 \pm 0.89$ | $13.09 \pm 0.81$ | $20.09 \pm 1.06$ |
| | flowers | $7.68 \pm 1.45$ | $8.17 \pm 0.52$ | $7.60 \pm 0.35$ | $7.75 \pm 0.81$ | $19.45 \pm 5.82$ | $12.37 \pm 1.34$ | $28.87 \pm 1.09$ |
| | food containers | $13.62 \pm 1.87$ | $12.17 \pm 0.83$ | $13.91 \pm 0.46$ | $12.64 \pm 1.02$ | $10.34 \pm 0.61$ | $9.23 \pm 0.59$ | $27.05 \pm 2.39$ |
| | fruit and vegetables | $10.01 \pm 0.99$ | $8.73 \pm 0.34$ | $10.58 \pm 0.53$ | $8.85 \pm 1.27$ | $19.02 \pm 5.38$ | $11.94 \pm 0.91$ | $24.54 \pm 1.65$ |
| | household electrical devices | $10.49 \pm 1.46$ | $8.28 \pm 0.11$ | $10.35 \pm 1.28$ | $8.97 \pm 0.54$ | $10.05 \pm 1.03$ | $9.52 \pm 0.28$ | $14.03 \pm 1.07$ |
| | household furniture | $14.63 \pm 1.16$ | $10.88 \pm 0.32$ | $14.41 \pm 0.36$ | $11.22 \pm 1.10$ | $12.48 \pm 3.54$ | $9.28 \pm 0.24$ | $26.64 \pm 1.30$ |
| | inserts | $9.67 \pm 0.32$ | $11.35 \pm 0.45$ | $10.02 \pm 0.43$ | $10.36 \pm 1.19$ | $9.86 \pm 0.52$ | $11.41 \pm 0.35$ | $22.39 \pm 1.05$ |
| | large carnivores | $12.77 \pm 1.30$ | $14.53 \pm 1.45$ | $12.56 \pm 0.88$ | $17.12 \pm 1.83$ | $18.30 \pm 1.66$ | $18.40 \pm 3.88$ | $47.93 \pm 3.86$ |
| | large man-made outdoor things | $20.79 \pm 3.02$ | $18.40 \pm 0.91$ | $21.11 \pm 1.51$ | $24.87 \pm 2.28$ | $21.88 \pm 3.73$ | $15.88 \pm 1.92$ | $49.89 \pm 1.91$ |
| | large natural outdoor scenes | $33.77 \pm 6.48$ | $38.32 \pm 4.21$ | $37.91 \pm 2.92$ | $33.81 \pm 6.97$ | $15.16 \pm 3.15$ | $14.51 \pm 1.19$ | $35.79 \pm 1.96$ |
| | large omnivores and herbivores | $15.53 \pm 0.55$ | $11.97 \pm 0.80$ | $16.29 \pm 0.94$ | $16.77 \pm 1.79$ | $17.23 \pm 1.19$ | $13.42 \pm 1.93$ | $38.28 \pm 1.06$ |
| | medium-sized mammals | $17.22 \pm 1.21$ | $12.91 \pm 0.52$ | $15.74 \pm 0.63$ | $18.47 \pm 1.85$ | $20.08 \pm 1.43$ | $15.60 \pm 2.21$ | $43.57 \pm 2.63$ |
| | non-insert invertebrates | $10.21 \pm 0.50$ | $11.86 \pm 0.21$ | $10.22 \pm 0.28$ | $10.11 \pm 0.37$ | $9.54 \pm 0.39$ | $10.43 \pm 0.40$ | $13.60 \pm 0.77$ |
| | people | $10.12 \pm 0.78$ | $7.98 \pm 0.16$ | $10.04 \pm 0.50$ | $10.03 \pm 0.93$ | $13.03 \pm 1.22$ | $11.92 \pm 1.15$ | $44.16 \pm 2.99$ |
| | reptiles | $13.79 \pm 1.10$ | $13.13 \pm 0.65$ | $13.83 \pm 0.50$ | $13.77 \pm 0.99$ | $10.22 \pm 0.33$ | $11.22 \pm 0.81$ | $22.71 \pm 1.84$ |
| | small mammals | $16.92 \pm 2.01$ | $19.30 \pm 1.29$ | $18.13 \pm 1.39$ | $20.81 \pm 1.30$ | $14.90 \pm 0.86$ | $13.48 \pm 0.73$ | $29.54 \pm 2.53$ |
| | trees | $17.14 \pm 3.19$ | $21.37 \pm 0.92$ | $19.58 \pm 1.80$ | $21.38 \pm 2.93$ | $18.12 \pm 3.25$ | $20.31 \pm 3.83$ | $56.26 \pm 2.17$ |
| | vehicles 1 | $7.84 \pm 0.52$ | $8.09 \pm 0.21$ | $7.36 \pm 0.44$ | $7.91 \pm 0.37$ | $13.82 \pm 1.10$ | $13.95 \pm 1.36$ | $56.68 \pm 1.98$ |
| | vehicles 2 | $10.72 \pm 1.30$ | $10.77 \pm 1.00$ | $11.33 \pm 0.94$ | $11.63 \pm 1.30$ | $10.90 \pm 0.84$ | $11.50 \pm 0.39$ | $36.67 \pm 2.35$ |
| | *average* | $14.51 \pm 0.57$ | $13.83 \pm 0.21$ | $14.96 \pm 0.11$ | $15.01 \pm 0.43$ | $14.19 \pm 0.45$ | $12.98 \pm 0.17$ | $\mathbf{33.30 \pm 0.39}$ |
| SVHN | 0 | $10.34 \pm 0.37$ | $13.17 \pm 0.62$ | $10.65 \pm 0.29$ | $10.59 \pm 0.58$ | $15.27 \pm 6.31$ | $30.87 \pm 3.14$ | $25.89 \pm 1.45$ |
| | 1 | $12.79 \pm 0.78$ | $14.57 \pm 0.96$ | $12.38 \pm 0.55$ | $13.11 \pm 0.81$ | $21.94 \pm 7.09$ | $17.59 \pm 0.83$ | $22.15 \pm 1.09$ |
| | 2 | $10.75 \pm 0.16$ | $11.73 \pm 0.20$ | $10.67 \pm 0.27$ | $10.93 \pm 0.25$ | $17.36 \pm 8.07$ | $20.92 \pm 1.69$ | $40.71 \pm 1.28$ |
| | 3 | $10.05 \pm 0.15$ | $10.67 \pm 0.25$ | $9.91 \pm 0.20$ | $10.15 \pm 0.19$ | $15.81 \pm 5.72$ | $17.18 \pm 1.44$ | $32.45 \pm 0.24$ |
| | 4 | $11.01 \pm 0.44$ | $12.43 \pm 0.75$ | $10.85 \pm 0.30$ | $11.72 \pm 0.29$ | $21.97 \pm 5.98$ | $18.62 \pm 2.11$ | $44.26 \pm 1.43$ |
| | 5 | $9.71 \pm 0.24$ | $10.61 \pm 0.40$ | $9.58 \pm 0.31$ | $9.84 \pm 0.32$ | $25.10 \pm 10.95$ | $21.07 \pm 2.55$ | $44.00 \pm 1.80$ |
| | 6 | $9.83 \pm 0.26$ | $10.34 \pm 0.37$ | $9.44 \pm 0.22$ | $9.89 \pm 0.18$ | $16.63 \pm 6.96$ | $22.18 \pm 1.82$ | $41.63 \pm 1.23$ |
| | 7 | $10.87 \pm 0.53$ | $12.72 \pm 0.35$ | $11.35 \pm 0.51$ | $10.98 \pm 0.68$ | $25.77 \pm 14.33$ | $28.50 \pm 4.72$ | $50.71 \pm 1.37$ |
| | 8 | $9.95 \pm 0.27$ | $11.14 \pm 0.32$ | $9.58 \pm 0.22$ | $9.98 \pm 0.31$ | $14.60 \pm 8.64$ | $19.40 \pm 1.99$ | $25.04 \pm 2.20$ |
| | 9 | $10.29 \pm 0.32$ | $11.30 \pm 0.43$ | $10.16 \pm 0.12$ | $10.71 \pm 0.42$ | $18.70 \pm 8.92$ | $24.09 \pm 0.69$ | $39.67 \pm 1.65$ |
| | *average* | $10.56 \pm 0.12$ | $11.87 \pm 0.26$ | $10.46 \pm 0.18$ | $10.79 \pm 0.21$ | $19.31 \pm 2.43$ | $22.04 \pm 0.79$ | $\mathbf{36.65 \pm 0.12}$ |

Table 13: AUPR-out (%) (outliers as positive class) when $\rho = 0.15$

| Dataset | Class name | CAE | CAE-IF | DRAE | RDAE | DAGMM | SSD-IF | $E^3$Outlier |
|---|---|---|---|---|---|---|---|---|
| MNIST | 0 | 19.63 ± 3.45 | 56.80 ± 3.92 | 34.84 ± 17.52 | 20.60 ± 3.94 | 51.66 ± 31.41 | 60.50 ± 7.01 | 70.15 ± 0.88 |
| | 1 | 92.84 ± 1.76 | 94.57 ± 0.99 | 63.41 ± 13.90 | 93.78 ± 0.76 | 69.22 ± 28.68 | 80.60 ± 2.59 | 71.06 ± 2.21 |
| | 2 | 18.99 ± 3.21 | 35.31 ± 1.85 | 23.52 ± 4.03 | 19.38 ± 2.74 | 16.18 ± 3.28 | 60.47 ± 2.86 | 61.70 ± 1.16 |
| | 3 | 21.95 ± 2.07 | 42.16 ± 3.65 | 32.10 ± 8.41 | 29.69 ± 2.20 | 25.17 ± 12.63 | 73.27 ± 1.84 | 84.90 ± 0.55 |
| | 4 | 37.58 ± 5.59 | 50.08 ± 2.86 | 37.21 ± 10.38 | 42.75 ± 2.34 | 30.80 ± 14.52 | 78.07 ± 2.29 | 73.62 ± 1.76 |
| | 5 | 20.65 ± 1.71 | 51.51 ± 1.03 | 24.97 ± 1.52 | 17.81 ± 3.57 | 17.18 ± 1.68 | 65.53 ± 2.68 | 61.18 ± 1.44 |
| | 6 | 31.59 ± 4.34 | 58.36 ± 5.84 | 25.89 ± 7.68 | 34.20 ± 3.28 | 20.58 ± 6.85 | 78.57 ± 1.71 | 76.61 ± 0.74 |
| | 7 | 44.50 ± 11.65 | 66.81 ± 1.79 | 48.08 ± 3.25 | 50.74 ± 5.18 | 38.57 ± 20.82 | 74.09 ± 2.56 | 69.14 ± 0.42 |
| | 8 | 16.78 ± 2.35 | 33.83 ± 2.09 | 22.57 ± 4.92 | 18.78 ± 2.52 | 19.87 ± 5.95 | 50.92 ± 2.83 | 62.90 ± 1.71 |
| | 9 | 43.68 ± 4.63 | 59.83 ± 1.89 | 37.61 ± 8.89 | 43.89 ± 5.99 | 26.06 ± 6.49 | 79.04 ± 1.23 | 75.14 ± 1.42 |
| | *average* | 34.82 ± 1.84 | 54.93 ± 0.76 | 35.02 ± 2.52 | 37.16 ± 2.20 | 31.53 ± 3.75 | 70.11 ± 0.96 | **70.64 ± 0.43** |
| Fashion-MNIST | t-shirt | 21.57 ± 2.11 | 43.29 ± 1.61 | 22.81 ± 1.16 | 25.36 ± 1.93 | 44.81 ± 16.03 | 62.02 ± 3.19 | 72.47 ± 0.70 |
| | trouser | 71.85 ± 2.31 | 79.02 ± 3.27 | 66.75 ± 4.11 | 77.47 ± 2.30 | 50.89 ± 7.97 | 81.51 ± 4.08 | 85.37 ± 1.11 |
| | pullover | 21.73 ± 3.37 | 41.76 ± 0.75 | 20.87 ± 2.75 | 23.69 ± 4.47 | 22.81 ± 5.83 | 72.05 ± 2.58 | 82.45 ± 1.21 |
| | dress | 31.72 ± 2.77 | 47.87 ± 3.54 | 31.56 ± 2.11 | 32.58 ± 2.90 | 49.00 ± 8.61 | 69.36 ± 1.03 | 66.26 ± 1.68 |
| | coat | 25.81 ± 1.80 | 47.50 ± 3.22 | 25.40 ± 1.55 | 29.88 ± 6.34 | 33.56 ± 19.64 | 74.28 ± 0.55 | 79.95 ± 0.42 |
| | sandal | 33.17 ± 2.53 | 21.92 ± 1.94 | 32.66 ± 1.48 | 31.39 ± 7.72 | 71.38 ± 14.66 | 64.65 ± 3.23 | 80.60 ± 1.07 |
| | shirt | 15.18 ± 0.77 | 31.35 ± 1.61 | 14.85 ± 0.53 | 19.22 ± 2.15 | 27.05 ± 12.51 | 59.75 ± 1.75 | 63.88 ± 1.15 |
| | sneaker | 64.38 ± 4.59 | 69.43 ± 2.55 | 68.28 ± 4.36 | 66.56 ± 5.17 | 59.76 ± 11.51 | 90.42 ± 1.99 | 95.87 ± 0.72 |
| | bag | 13.57 ± 2.37 | 22.99 ± 1.47 | 13.42 ± 0.77 | 14.21 ± 2.40 | 26.01 ± 14.59 | 39.34 ± 2.90 | 57.58 ± 0.79 |
| | ankle-boot | 47.24 ± 4.04 | 50.00 ± 3.85 | 36.81 ± 1.84 | 45.48 ± 2.85 | 43.18 ± 25.86 | 91.67 ± 1.84 | 97.21 ± 0.24 |
| | *average* | 34.62 ± 1.14 | 45.51 ± 0.91 | 33.34 ± 1.03 | 36.58 ± 1.36 | 42.84 ± 8.88 | 70.51 ± 0.55 | **78.16 ± 0.22** |
| CIFAR10 | airplane | 28.96 ± 2.69 | 21.62 ± 0.98 | 28.72 ± 0.96 | 22.54 ± 2.06 | 16.90 ± 4.11 | 15.90 ± 0.91 | 35.47 ± 1.46 |
| | automobile | 11.31 ± 0.39 | 10.99 ± 0.19 | 11.58 ± 0.25 | 10.85 ± 0.43 | 24.02 ± 5.03 | 31.48 ± 4.78 | 76.88 ± 1.60 |
| | bird | 22.03 ± 0.73 | 24.27 ± 0.39 | 24.24 ± 0.36 | 24.02 ± 1.08 | 14.16 ± 2.11 | 16.10 ± 0.94 | 24.80 ± 0.52 |
| | cat | 20.05 ± 0.76 | 16.93 ± 0.51 | 20.68 ± 0.23 | 18.73 ± 1.17 | 17.47 ± 1.54 | 17.10 ± 1.17 | 24.04 ± 0.83 |
| | deer | 25.01 ± 1.51 | 31.54 ± 0.86 | 27.07 ± 0.47 | 30.60 ± 1.35 | 19.00 ± 2.68 | 23.24 ± 1.06 | 44.35 ± 1.01 |
| | dog | 21.32 ± 1.08 | 16.54 ± 0.58 | 22.22 ± 1.02 | 19.83 ± 1.46 | 17.14 ± 1.26 | 24.36 ± 1.91 | 44.63 ± 2.11 |
| | frog | 16.51 ± 1.55 | 23.36 ± 0.73 | 14.58 ± 0.59 | 18.64 ± 1.32 | 22.42 ± 3.85 | 25.66 ± 3.27 | 40.98 ± 1.98 |
| | horse | 15.96 ± 0.78 | 14.56 ± 0.44 | 16.73 ± 0.49 | 16.47 ± 0.59 | 21.53 ± 3.31 | 30.66 ± 3.61 | 60.66 ± 1.13 |
| | ship | 32.49 ± 2.49 | 26.70 ± 1.26 | 33.55 ± 0.96 | 28.66 ± 4.67 | 25.81 ± 15.68 | 27.90 ± 4.04 | 68.33 ± 1.41 |
| | truck | 12.06 ± 0.96 | 10.81 ± 0.12 | 11.96 ± 0.07 | 11.00 ± 0.51 | 26.47 ± 6.72 | 30.02 ± 1.79 | 66.14 ± 0.75 |
| | *average* | 20.57 ± 0.23 | 19.73 ± 0.07 | 21.13 ± 0.14 | 20.13 ± 0.38 | 20.49 ± 1.20 | 24.24 ± 0.91 | **48.63 ± 0.18** |
| CIFAR100 | aquatic mammals | 29.51 ± 1.00 | 23.27 ± 0.79 | 29.46 ± 0.87 | 27.64 ± 1.90 | 16.03 ± 0.85 | 19.37 ± 1.70 | 34.82 ± 1.87 |
| | fish | 25.04 ± 1.33 | 20.32 ± 1.28 | 23.92 ± 0.60 | 22.81 ± 1.66 | 13.30 ± 1.78 | 19.21 ± 0.43 | 28.85 ± 1.23 |
| | flowers | 11.00 ± 0.56 | 11.17 ± 0.32 | 11.22 ± 0.64 | 11.41 ± 1.20 | 23.69 ± 7.50 | 17.86 ± 0.96 | 38.07 ± 3.55 |
| | food containers | 19.45 ± 1.01 | 19.37 ± 0.58 | 19.12 ± 0.70 | 17.69 ± 0.83 | 14.29 ± 0.50 | 15.06 ± 0.71 | 32.55 ± 1.07 |
| | fruit and vegetables | 15.65 ± 1.36 | 13.70 ± 0.48 | 15.97 ± 0.49 | 15.02 ± 1.17 | 25.03 ± 5.12 | 17.58 ± 1.67 | 32.06 ± 1.44 |
| | household electrical devices | 15.41 ± 0.84 | 12.66 ± 0.28 | 16.07 ± 0.45 | 13.25 ± 0.70 | 16.29 ± 2.37 | 14.77 ± 0.29 | 19.67 ± 1.26 |
| | household furniture | 20.84 ± 1.59 | 17.26 ± 0.89 | 22.76 ± 1.09 | 18.47 ± 1.74 | 17.47 ± 3.15 | 14.33 ± 0.58 | 34.62 ± 1.89 |
| | inserts | 14.80 ± 1.07 | 15.54 ± 0.47 | 14.46 ± 0.42 | 15.01 ± 0.27 | 16.14 ± 2.95 | 16.37 ± 0.42 | 30.67 ± 0.89 |
| | large carnivores | 16.17 ± 0.98 | 19.68 ± 1.88 | 19.46 ± 0.92 | 19.51 ± 0.53 | 24.64 ± 0.53 | 23.39 ± 3.18 | 55.01 ± 4.95 |
| | large man-made outdoor things | 27.54 ± 1.60 | 26.39 ± 1.10 | 34.00 ± 0.96 | 30.55 ± 1.45 | 28.62 ± 6.09 | 23.06 ± 2.56 | 57.92 ± 2.35 |
| | large natural outdoor scenes | 46.29 ± 2.49 | 48.23 ± 3.14 | 44.54 ± 2.91 | 48.01 ± 3.28 | 18.63 ± 2.31 | 18.68 ± 3.27 | 43.78 ± 2.36 |
| | large omnivores and herbivores | 21.15 ± 0.74 | 17.86 ± 0.14 | 23.48 ± 1.30 | 22.15 ± 0.71 | 23.13 ± 3.16 | 18.05 ± 1.46 | 45.68 ± 1.90 |
| | medium-sized mammals | 20.59 ± 1.76 | 18.62 ± 0.77 | 21.90 ± 1.02 | 24.67 ± 1.56 | 24.45 ± 1.79 | 22.26 ± 3.16 | 49.82 ± 1.72 |
| | non-insert invertebrates | 15.25 ± 0.45 | 16.91 ± 0.54 | 14.86 ± 0.46 | 15.40 ± 0.34 | 14.25 ± 0.69 | 15.04 ± 0.45 | 18.24 ± 0.47 |
| | people | 14.88 ± 0.75 | 12.22 ± 0.22 | 15.15 ± 0.71 | 14.40 ± 0.91 | 20.03 ± 1.78 | 16.65 ± 1.92 | 52.46 ± 2.46 |
| | reptiles | 17.58 ± 0.92 | 18.29 ± 0.84 | 19.39 ± 0.83 | 19.32 ± 0.41 | 15.57 ± 1.48 | 17.73 ± 1.69 | 28.92 ± 1.60 |
| | small mammals | 23.17 ± 0.79 | 25.25 ± 0.92 | 25.82 ± 1.51 | 27.79 ± 1.06 | 20.09 ± 2.29 | 19.43 ± 1.73 | 35.92 ± 0.79 |
| | trees | 25.84 ± 1.34 | 27.58 ± 1.59 | 29.44 ± 1.75 | 28.58 ± 1.11 | 23.48 ± 3.23 | 29.27 ± 3.39 | 63.39 ± 1.68 |
| | vehicles 1 | 11.68 ± 0.48 | 11.60 ± 0.24 | 12.12 ± 0.73 | 11.69 ± 0.61 | 20.87 ± 1.22 | 19.32 ± 2.56 | 65.21 ± 1.73 |
| | vehicles 2 | 15.38 ± 0.95 | 14.77 ± 0.85 | 17.98 ± 1.08 | 15.83 ± 0.85 | 17.13 ± 2.51 | 16.03 ± 0.81 | 46.35 ± 1.83 |
| | *average* | 20.36 ± 0.22 | 19.53 ± 0.35 | 21.60 ± 0.18 | 20.96 ± 0.24 | 19.66 ± 0.44 | 18.67 ± 0.27 | **40.70 ± 0.46** |
| SVHN | 0 | 15.83 ± 0.77 | 18.51 ± 0.23 | 15.73 ± 0.38 | 16.32 ± 0.43 | 20.37 ± 4.33 | 38.43 ± 3.73 | 29.97 ± 1.49 |
| | 1 | 17.59 ± 0.65 | 20.30 ± 0.68 | 18.08 ± 0.99 | 18.65 ± 0.81 | 21.37 ± 6.57 | 24.24 ± 2.65 | 27.28 ± 2.15 |
| | 2 | 15.72 ± 0.28 | 17.02 ± 0.40 | 15.63 ± 0.22 | 16.30 ± 0.41 | 23.08 ± 7.58 | 28.30 ± 3.37 | 47.74 ± 2.10 |
| | 3 | 15.01 ± 0.23 | 15.65 ± 0.20 | 14.69 ± 0.27 | 14.95 ± 0.25 | 24.86 ± 5.48 | 24.08 ± 1.89 | 40.33 ± 0.66 |
| | 4 | 16.40 ± 0.32 | 17.88 ± 0.73 | 16.26 ± 0.30 | 17.12 ± 0.47 | 26.29 ± 9.36 | 25.53 ± 1.16 | 49.82 ± 0.99 |
| | 5 | 14.92 ± 0.23 | 15.85 ± 0.18 | 14.52 ± 0.14 | 15.06 ± 0.11 | 23.13 ± 13.29 | 24.87 ± 1.95 | 50.54 ± 0.89 |
| | 6 | 14.58 ± 0.35 | 15.45 ± 0.23 | 14.12 ± 0.32 | 14.73 ± 0.25 | 29.02 ± 15.07 | 27.17 ± 1.27 | 47.04 ± 1.10 |
| | 7 | 17.42 ± 0.50 | 18.69 ± 0.67 | 17.09 ± 0.65 | 17.59 ± 0.50 | 30.14 ± 7.67 | 36.04 ± 2.12 | 55.95 ± 1.52 |
| | 8 | 14.82 ± 0.25 | 16.16 ± 0.40 | 14.55 ± 0.26 | 15.14 ± 0.33 | 20.52 ± 10.54 | 27.21 ± 2.71 | 29.10 ± 0.95 |
| | 9 | 15.28 ± 0.52 | 16.46 ± 0.49 | 15.17 ± 0.19 | 15.68 ± 0.59 | 23.63 ± 5.31 | 28.39 ± 1.72 | 46.74 ± 1.46 |
| | *average* | 15.76 ± 0.11 | 17.20 ± 0.13 | 15.58 ± 0.18 | 16.15 ± 0.14 | 24.24 ± 2.19 | 28.43 ± 0.34 | **42.45 ± 0.59** |

Table 14: AUPR-out (%) (outliers as positive class) when $\rho = 0.2$

| Dataset | Class name | CAE | CAE-IF | DRAE | RDAE | DAGMM | SSD-IF | $E^3Outlier$ |
|---|---|---|---|---|---|---|---|---|
| MNIST | 0 | $24.25 \pm 2.91$ | $62.27 \pm 7.97$ | $33.91 \pm 6.62$ | $26.76 \pm 5.49$ | $51.46 \pm 29.90$ | $58.60 \pm 5.61$ | $71.09 \pm 1.14$ |
| | 1 | $94.72 \pm 0.76$ | $95.66 \pm 0.89$ | $80.07 \pm 15.25$ | $94.57 \pm 0.26$ | $74.00 \pm 23.72$ | $81.35 \pm 1.71$ | $68.21 \pm 1.78$ |
| | 2 | $23.89 \pm 3.03$ | $39.45 \pm 4.08$ | $25.56 \pm 3.79$ | $25.81 \pm 3.18$ | $21.10 \pm 1.17$ | $60.82 \pm 1.53$ | $65.28 \pm 1.10$ |
| | 3 | $30.11 \pm 3.97$ | $48.24 \pm 2.61$ | $44.39 \pm 11.30$ | $35.18 \pm 2.99$ | $32.53 \pm 11.64$ | $71.18 \pm 2.77$ | $86.03 \pm 0.46$ |
| | 4 | $46.04 \pm 4.39$ | $54.55 \pm 3.66$ | $46.07 \pm 5.90$ | $48.36 \pm 4.38$ | $42.00 \pm 20.73$ | $79.51 \pm 1.49$ | $74.36 \pm 0.93$ |
| | 5 | $29.21 \pm 2.04$ | $38.47 \pm 2.53$ | $26.66 \pm 4.23$ | $30.33 \pm 2.85$ | $23.83 \pm 2.52$ | $68.62 \pm 1.34$ | $63.27 \pm 1.75$ |
| | 6 | $34.40 \pm 2.93$ | $63.89 \pm 5.90$ | $39.16 \pm 7.09$ | $37.31 \pm 3.60$ | $51.17 \pm 24.61$ | $79.81 \pm 1.85$ | $78.60 \pm 0.60$ |
| | 7 | $58.80 \pm 5.55$ | $71.69 \pm 2.60$ | $53.66 \pm 6.73$ | $61.22 \pm 2.33$ | $57.72 \pm 18.90$ | $75.56 \pm 2.11$ | $72.44 \pm 0.52$ |
| | 8 | $19.86 \pm 0.88$ | $35.67 \pm 2.05$ | $32.93 \pm 6.04$ | $23.89 \pm 1.16$ | $27.07 \pm 10.86$ | $54.76 \pm 2.44$ | $66.11 \pm 0.88$ |
| | 9 | $46.01 \pm 2.95$ | $62.02 \pm 2.85$ | $42.44 \pm 6.51$ | $48.90 \pm 3.06$ | $32.57 \pm 10.13$ | $80.10 \pm 1.75$ | $77.75 \pm 1.01$ |
| | *average* | $40.73 \pm 0.99$ | $57.19 \pm 1.69$ | $42.48 \pm 2.33$ | $43.23 \pm 0.75$ | $41.34 \pm 5.16$ | $71.03 \pm 0.84$ | $\mathbf{72.31 \pm 0.33}$ |
| Fashion-MNIST | t-shirt | $23.31 \pm 3.49$ | $46.19 \pm 1.81$ | $26.11 \pm 2.65$ | $32.45 \pm 1.40$ | $38.99 \pm 17.60$ | $66.60 \pm 1.69$ | $73.97 \pm 0.74$ |
| | trouser | $76.39 \pm 1.01$ | $80.92 \pm 1.16$ | $67.48 \pm 7.84$ | $79.83 \pm 1.34$ | $54.67 \pm 18.42$ | $84.15 \pm 1.98$ | $85.84 \pm 0.77$ |
| | pullover | $26.66 \pm 1.09$ | $45.65 \pm 2.45$ | $26.02 \pm 1.92$ | $33.19 \pm 2.30$ | $17.71 \pm 2.38$ | $74.36 \pm 1.77$ | $83.98 \pm 0.66$ |
| | dress | $32.49 \pm 2.64$ | $50.12 \pm 2.53$ | $39.56 \pm 3.17$ | $37.77 \pm 2.77$ | $66.75 \pm 11.43$ | $67.43 \pm 1.39$ | $66.79 \pm 0.51$ |
| | coat | $34.47 \pm 2.42$ | $53.01 \pm 2.41$ | $29.41 \pm 1.87$ | $38.48 \pm 3.14$ | $25.35 \pm 12.35$ | $76.62 \pm 0.90$ | $82.18 \pm 0.86$ |
| | sandal | $33.55 \pm 5.41$ | $24.77 \pm 0.65$ | $38.98 \pm 2.88$ | $30.15 \pm 4.28$ | $63.53 \pm 11.03$ | $62.88 \pm 5.87$ | $78.96 \pm 1.52$ |
| | shirt | $19.89 \pm 1.09$ | $37.51 \pm 0.51$ | $19.92 \pm 1.00$ | $23.99 \pm 1.55$ | $24.03 \pm 7.65$ | $60.59 \pm 1.70$ | $65.75 \pm 0.52$ |
| | sneaker | $60.88 \pm 5.89$ | $68.98 \pm 1.30$ | $59.68 \pm 19.32$ | $69.21 \pm 2.75$ | $64.38 \pm 14.74$ | $91.24 \pm 0.40$ | $96.21 \pm 0.58$ |
| | bag | $17.40 \pm 0.95$ | $26.26 \pm 0.89$ | $17.34 \pm 0.63$ | $18.13 \pm 1.15$ | $28.32 \pm 11.86$ | $39.41 \pm 1.47$ | $57.95 \pm 1.78$ |
| | ankle-boot | $42.82 \pm 4.78$ | $56.35 \pm 3.60$ | $41.21 \pm 6.77$ | $50.74 \pm 4.18$ | $51.50 \pm 23.96$ | $90.28 \pm 0.95$ | $97.35 \pm 0.04$ |
| | *average* | $36.79 \pm 1.34$ | $48.97 \pm 0.97$ | $36.57 \pm 2.52$ | $41.40 \pm 0.88$ | $43.52 \pm 2.89$ | $71.36 \pm 0.76$ | $\mathbf{78.90 \pm 0.32}$ |
| CIFAR10 | airplane | $31.34 \pm 4.43$ | $27.89 \pm 0.60$ | $35.66 \pm 0.66$ | $28.97 \pm 2.92$ | $20.32 \pm 3.75$ | $20.23 \pm 1.27$ | $41.60 \pm 1.15$ |
| | automobile | $15.91 \pm 1.22$ | $14.95 \pm 0.31$ | $15.62 \pm 0.30$ | $14.38 \pm 0.30$ | $32.31 \pm 4.03$ | $33.73 \pm 4.39$ | $79.73 \pm 0.71$ |
| | bird | $28.71 \pm 1.48$ | $30.73 \pm 0.70$ | $31.25 \pm 0.88$ | $31.90 \pm 0.54$ | $18.95 \pm 2.30$ | $22.01 \pm 2.16$ | $29.45 \pm 0.94$ |
| | cat | $26.01 \pm 1.59$ | $22.48 \pm 0.59$ | $26.31 \pm 0.56$ | $24.59 \pm 1.55$ | $22.79 \pm 1.29$ | $22.39 \pm 2.15$ | $27.81 \pm 1.73$ |
| | deer | $31.61 \pm 1.79$ | $38.29 \pm 0.72$ | $33.79 \pm 0.72$ | $37.69 \pm 1.04$ | $23.90 \pm 4.55$ | $28.57 \pm 1.92$ | $48.52 \pm 1.60$ |
| | dog | $25.96 \pm 1.48$ | $21.62 \pm 0.57$ | $28.28 \pm 0.61$ | $25.45 \pm 0.65$ | $23.61 \pm 3.02$ | $29.78 \pm 2.47$ | $47.05 \pm 3.04$ |
| | frog | $20.44 \pm 2.29$ | $29.81 \pm 0.96$ | $19.31 \pm 0.65$ | $23.17 \pm 1.51$ | $29.39 \pm 5.21$ | $28.80 \pm 2.11$ | $45.62 \pm 1.81$ |
| | horse | $20.25 \pm 0.91$ | $19.15 \pm 0.54$ | $21.56 \pm 0.32$ | $20.25 \pm 0.23$ | $25.77 \pm 5.59$ | $32.38 \pm 2.19$ | $66.72 \pm 1.68$ |
| | ship | $38.81 \pm 4.02$ | $33.61 \pm 1.02$ | $40.14 \pm 0.88$ | $36.11 \pm 3.39$ | $26.21 \pm 10.64$ | $31.80 \pm 2.66$ | $72.48 \pm 1.70$ |
| | truck | $15.58 \pm 0.61$ | $14.40 \pm 0.17$ | $15.64 \pm 0.20$ | $14.48 \pm 0.40$ | $39.80 \pm 8.34$ | $31.89 \pm 1.99$ | $68.15 \pm 1.77$ |
| | *average* | $25.46 \pm 0.63$ | $25.29 \pm 0.06$ | $26.76 \pm 0.20$ | $25.70 \pm 0.66$ | $26.30 \pm 0.64$ | $28.16 \pm 0.74$ | $\mathbf{52.71 \pm 0.24}$ |
| CIFAR100 | aquatic mammals | $36.66 \pm 2.20$ | $29.09 \pm 1.15$ | $37.22 \pm 1.66$ | $35.87 \pm 1.39$ | $23.02 \pm 3.30$ | $23.26 \pm 1.57$ | $41.89 \pm 1.35$ |
| | fish | $30.51 \pm 2.17$ | $26.26 \pm 0.47$ | $30.55 \pm 0.93$ | $29.02 \pm 1.86$ | $18.46 \pm 1.59$ | $25.19 \pm 1.83$ | $34.24 \pm 0.79$ |
| | flowers | $14.74 \pm 0.71$ | $14.98 \pm 0.25$ | $15.00 \pm 0.53$ | $16.11 \pm 1.76$ | $31.44 \pm 5.00$ | $22.58 \pm 1.17$ | $45.67 \pm 1.41$ |
| | food containers | $24.99 \pm 1.19$ | $25.80 \pm 0.73$ | $25.02 \pm 0.39$ | $24.17 \pm 0.74$ | $17.65 \pm 1.82$ | $18.28 \pm 0.91$ | $39.26 \pm 1.32$ |
| | fruit and vegetables | $20.23 \pm 1.96$ | $17.91 \pm 0.46$ | $21.13 \pm 0.51$ | $18.98 \pm 1.46$ | $29.48 \pm 4.84$ | $21.98 \pm 0.76$ | $38.17 \pm 1.69$ |
| | household electrical devices | $19.65 \pm 0.64$ | $16.83 \pm 0.11$ | $22.26 \pm 0.42$ | $17.91 \pm 1.05$ | $20.48 \pm 1.18$ | $19.16 \pm 0.88$ | $25.08 \pm 1.04$ |
| | household furniture | $25.99 \pm 2.64$ | $22.77 \pm 0.38$ | $28.40 \pm 0.34$ | $25.26 \pm 1.72$ | $19.72 \pm 1.98$ | $18.95 \pm 0.65$ | $41.66 \pm 2.13$ |
| | inserts | $18.98 \pm 0.90$ | $20.43 \pm 0.51$ | $19.08 \pm 0.57$ | $19.67 \pm 1.15$ | $19.76 \pm 0.67$ | $22.00 \pm 1.20$ | $34.74 \pm 1.23$ |
| | large carnivores | $22.03 \pm 1.94$ | $24.64 \pm 1.35$ | $24.20 \pm 1.01$ | $26.55 \pm 2.33$ | $28.81 \pm 3.13$ | $26.56 \pm 1.98$ | $59.16 \pm 2.97$ |
| | large man-made outdoor things | $34.84 \pm 1.73$ | $32.68 \pm 2.17$ | $39.87 \pm 0.74$ | $35.65 \pm 1.14$ | $31.46 \pm 7.10$ | $27.52 \pm 3.91$ | $63.81 \pm 0.40$ |
| | large natural outdoor scenes | $50.56 \pm 2.94$ | $56.18 \pm 1.93$ | $49.58 \pm 1.49$ | $53.74 \pm 2.93$ | $23.04 \pm 2.22$ | $24.90 \pm 1.57$ | $50.57 \pm 1.73$ |
| | large omnivores and herbivores | $26.28 \pm 1.07$ | $23.81 \pm 0.71$ | $29.68 \pm 0.83$ | $29.28 \pm 1.36$ | $28.73 \pm 2.53$ | $24.98 \pm 2.08$ | $51.20 \pm 1.55$ |
| | medium-sized mammals | $27.10 \pm 0.38$ | $24.12 \pm 1.37$ | $28.99 \pm 1.67$ | $31.94 \pm 2.70$ | $32.96 \pm 2.30$ | $27.86 \pm 1.44$ | $54.05 \pm 2.46$ |
| | non-insert invertebrates | $20.32 \pm 0.87$ | $22.18 \pm 0.60$ | $19.71 \pm 0.36$ | $20.43 \pm 0.61$ | $18.69 \pm 0.86$ | $19.88 \pm 0.90$ | $23.41 \pm 0.46$ |
| | people | $20.36 \pm 1.06$ | $16.54 \pm 0.52$ | $20.09 \pm 0.78$ | $18.95 \pm 0.72$ | $27.29 \pm 1.33$ | $22.76 \pm 1.81$ | $61.09 \pm 1.17$ |
| | reptiles | $23.66 \pm 0.76$ | $24.82 \pm 0.49$ | $24.53 \pm 0.98$ | $24.97 \pm 1.20$ | $19.94 \pm 0.50$ | $21.78 \pm 0.73$ | $35.34 \pm 0.90$ |
| | small mammals | $29.41 \pm 1.26$ | $32.44 \pm 1.25$ | $31.71 \pm 0.77$ | $33.25 \pm 1.55$ | $24.36 \pm 1.29$ | $24.22 \pm 1.84$ | $42.04 \pm 0.88$ |
| | trees | $30.73 \pm 3.21$ | $34.51 \pm 1.98$ | $34.08 \pm 1.45$ | $32.95 \pm 2.07$ | $29.95 \pm 2.59$ | $32.09 \pm 3.10$ | $68.32 \pm 1.04$ |
| | vehicles 1 | $15.06 \pm 0.91$ | $15.51 \pm 0.39$ | $15.79 \pm 0.41$ | $15.30 \pm 0.47$ | $27.42 \pm 2.77$ | $24.05 \pm 2.14$ | $69.71 \pm 0.83$ |
| | vehicles 2 | $20.66 \pm 1.63$ | $19.68 \pm 0.86$ | $22.59 \pm 1.01$ | $20.65 \pm 1.58$ | $21.52 \pm 2.03$ | $20.07 \pm 1.14$ | $50.69 \pm 2.06$ |
| | *average* | $25.64 \pm 0.58$ | $25.06 \pm 0.40$ | $26.97 \pm 0.19$ | $26.53 \pm 0.37$ | $24.71 \pm 0.65$ | $23.40 \pm 0.49$ | $\mathbf{46.51 \pm 0.39}$ |
| SVHN | 0 | $20.71 \pm 0.90$ | $23.42 \pm 0.52$ | $20.54 \pm 0.25$ | $21.10 \pm 0.43$ | $27.42 \pm 6.70$ | $46.08 \pm 4.35$ | $33.12 \pm 0.92$ |
| | 1 | $23.18 \pm 0.72$ | $25.92 \pm 0.84$ | $23.34 \pm 0.70$ | $23.55 \pm 0.41$ | $28.19 \pm 6.77$ | $28.43 \pm 1.74$ | $30.96 \pm 1.14$ |
| | 2 | $20.64 \pm 0.41$ | $22.16 \pm 0.51$ | $20.79 \pm 0.29$ | $21.28 \pm 0.38$ | $35.36 \pm 9.87$ | $32.95 \pm 2.88$ | $51.79 \pm 1.97$ |
| | 3 | $20.00 \pm 0.19$ | $20.75 \pm 0.34$ | $19.59 \pm 0.13$ | $20.18 \pm 0.17$ | $33.42 \pm 7.00$ | $27.36 \pm 0.60$ | $46.89 \pm 0.88$ |
| | 4 | $21.52 \pm 0.45$ | $23.06 \pm 0.40$ | $21.38 \pm 0.47$ | $22.21 \pm 0.44$ | $26.43 \pm 3.86$ | $28.22 \pm 1.60$ | $54.93 \pm 0.93$ |
| | 5 | $19.52 \pm 0.48$ | $21.04 \pm 0.26$ | $19.39 \pm 0.28$ | $20.00 \pm 0.39$ | $28.82 \pm 6.88$ | $30.49 \pm 3.15$ | $55.75 \pm 0.81$ |
| | 6 | $19.57 \pm 0.62$ | $20.28 \pm 0.34$ | $18.81 \pm 0.49$ | $19.73 \pm 0.36$ | $30.68 \pm 8.93$ | $34.32 \pm 1.32$ | $50.58 \pm 1.54$ |
| | 7 | $22.19 \pm 0.56$ | $23.73 \pm 0.57$ | $22.15 \pm 0.37$ | $22.27 \pm 1.12$ | $32.77 \pm 7.82$ | $38.24 \pm 3.78$ | $60.84 \pm 1.01$ |
| | 8 | $19.76 \pm 0.46$ | $21.12 \pm 0.43$ | $19.25 \pm 0.21$ | $20.06 \pm 0.33$ | $26.54 \pm 6.64$ | $35.86 \pm 3.21$ | $33.97 \pm 0.91$ |
| | 9 | $19.98 \pm 0.71$ | $22.42 \pm 0.39$ | $20.14 \pm 0.31$ | $20.76 \pm 0.65$ | $26.65 \pm 7.12$ | $35.45 \pm 1.65$ | $50.85 \pm 1.01$ |
| | *average* | $20.71 \pm 0.14$ | $22.39 \pm 0.11$ | $20.54 \pm 0.18$ | $21.12 \pm 0.19$ | $29.63 \pm 1.91$ | $33.74 \pm 0.68$ | $\mathbf{46.97 \pm 0.47}$ |

Table 15: AUPR-out (%) (outliers as positive class) when $\rho = 0.25$

| Dataset | Class name | CAE | CAE-IF | DRAE | RDAE | DAGMM | SSD-IF | $E^3Outlier$ |
|---|---|---|---|---|---|---|---|---|
| MNIST | 0 | $26.69 \pm 2.62$ | $61.98 \pm 3.88$ | $40.68 \pm 6.14$ | $28.46 \pm 3.48$ | $45.38 \pm 22.35$ | $60.50 \pm 3.40$ | $68.47 \pm 3.22$ |
| | 1 | $94.06 \pm 3.44$ | $94.00 \pm 1.46$ | $90.14 \pm 1.79$ | $93.20 \pm 2.68$ | $70.89 \pm 26.56$ | $82.89 \pm 1.59$ | $65.56 \pm 0.89$ |
| | 2 | $26.39 \pm 1.73$ | $40.16 \pm 2.33$ | $34.85 \pm 3.59$ | $31.38 \pm 2.83$ | $25.39 \pm 4.49$ | $62.53 \pm 1.94$ | $69.14 \pm 0.97$ |
| | 3 | $31.81 \pm 2.64$ | $51.47 \pm 1.95$ | $35.32 \pm 13.26$ | $35.15 \pm 3.33$ | $35.09 \pm 10.86$ | $71.62 \pm 1.94$ | $86.50 \pm 1.27$ |
| | 4 | $43.44 \pm 6.28$ | $59.23 \pm 2.21$ | $42.84 \pm 3.19$ | $50.36 \pm 5.38$ | $47.61 \pm 23.44$ | $79.42 \pm 0.99$ | $77.31 \pm 1.09$ |
| | 5 | $32.41 \pm 1.86$ | $44.70 \pm 2.81$ | $38.17 \pm 12.58$ | $40.02 \pm 3.50$ | $25.95 \pm 2.35$ | $67.59 \pm 2.17$ | $65.84 \pm 1.36$ |
| | 6 | $36.60 \pm 4.95$ | $62.75 \pm 2.00$ | $37.44 \pm 8.17$ | $46.62 \pm 4.35$ | $38.33 \pm 23.37$ | $77.45 \pm 2.75$ | $80.22 \pm 0.96$ |
| | 7 | $62.03 \pm 4.27$ | $74.01 \pm 3.36$ | $55.28 \pm 4.22$ | $65.42 \pm 3.90$ | $52.38 \pm 17.20$ | $76.34 \pm 2.20$ | $74.78 \pm 0.63$ |
| | 8 | $22.26 \pm 2.33$ | $41.25 \pm 2.81$ | $25.36 \pm 2.37$ | $23.14 \pm 0.73$ | $30.70 \pm 8.07$ | $59.19 \pm 2.95$ | $68.80 \pm 0.65$ |
| | 9 | $51.43 \pm 3.32$ | $67.11 \pm 1.99$ | $48.58 \pm 7.40$ | $56.62 \pm 2.20$ | $37.12 \pm 9.85$ | $81.94 \pm 1.58$ | $80.65 \pm 0.60$ |
| | *average* | $42.71 \pm 1.24$ | $59.67 \pm 0.84$ | $44.87 \pm 2.48$ | $47.04 \pm 0.55$ | $40.88 \pm 3.97$ | $71.95 \pm 0.61$ | $\mathbf{73.73 \pm 0.43}$ |
| Fashion-MNIST | t-shirt | $27.79 \pm 3.40$ | $53.62 \pm 1.60$ | $32.42 \pm 4.37$ | $35.58 \pm 2.88$ | $43.22 \pm 15.34$ | $68.48 \pm 1.48$ | $75.03 \pm 1.20$ |
| | trouser | $79.63 \pm 1.17$ | $80.78 \pm 1.23$ | $70.34 \pm 8.39$ | $81.79 \pm 2.14$ | $56.03 \pm 7.78$ | $84.23 \pm 1.43$ | $84.96 \pm 0.99$ |
| | pullover | $27.88 \pm 1.39$ | $51.84 \pm 1.78$ | $31.52 \pm 1.33$ | $38.05 \pm 1.43$ | $25.99 \pm 5.43$ | $76.97 \pm 1.06$ | $84.77 \pm 0.51$ |
| | dress | $40.74 \pm 2.45$ | $56.84 \pm 2.57$ | $43.36 \pm 1.86$ | $42.67 \pm 2.05$ | $44.94 \pm 15.89$ | $69.80 \pm 2.91$ | $68.50 \pm 1.00$ |
| | coat | $37.53 \pm 4.41$ | $57.29 \pm 0.66$ | $34.09 \pm 1.08$ | $43.48 \pm 1.83$ | $29.01 \pm 6.65$ | $79.60 \pm 1.52$ | $82.75 \pm 0.38$ |
| | sandal | $38.60 \pm 5.78$ | $29.14 \pm 1.53$ | $40.25 \pm 1.50$ | $35.98 \pm 4.69$ | $61.05 \pm 18.57$ | $60.85 \pm 5.39$ | $73.96 \pm 6.75$ |
| | shirt | $23.85 \pm 1.08$ | $42.73 \pm 0.84$ | $24.60 \pm 0.83$ | $30.69 \pm 1.80$ | $29.42 \pm 3.78$ | $64.51 \pm 1.72$ | $66.43 \pm 0.39$ |
| | sneaker | $67.30 \pm 3.20$ | $70.95 \pm 1.11$ | $71.05 \pm 5.14$ | $68.38 \pm 2.82$ | $68.99 \pm 21.31$ | $91.21 \pm 1.70$ | $96.67 \pm 0.37$ |
| | bag | $20.41 \pm 2.31$ | $30.39 \pm 0.99$ | $21.55 \pm 0.79$ | $21.53 \pm 2.04$ | $29.38 \pm 5.81$ | $37.12 \pm 4.00$ | $56.80 \pm 1.88$ |
| | ankle-boot | $48.66 \pm 5.26$ | $56.81 \pm 3.54$ | $44.80 \pm 1.86$ | $53.07 \pm 3.63$ | $45.05 \pm 20.44$ | $88.68 \pm 1.62$ | $97.13 \pm 0.14$ |
| | *average* | $41.24 \pm 0.92$ | $53.04 \pm 0.64$ | $41.40 \pm 0.89$ | $45.12 \pm 0.99$ | $43.31 \pm 5.88$ | $72.14 \pm 1.17$ | $\mathbf{78.70 \pm 0.36}$ |
| CIFAR10 | airplane | $38.96 \pm 3.07$ | $33.38 \pm 0.85$ | $41.42 \pm 1.00$ | $34.78 \pm 3.14$ | $23.97 \pm 5.71$ | $24.56 \pm 0.85$ | $44.14 \pm 1.63$ |
| | automobile | $19.54 \pm 0.71$ | $18.82 \pm 0.16$ | $19.69 \pm 0.34$ | $17.90 \pm 0.21$ | $39.18 \pm 5.95$ | $38.53 \pm 3.50$ | $80.42 \pm 2.32$ |
| | bird | $34.55 \pm 0.55$ | $37.27 \pm 0.44$ | $37.00 \pm 0.36$ | $37.87 \pm 0.61$ | $23.04 \pm 1.53$ | $26.59 \pm 1.26$ | $34.57 \pm 1.27$ |
| | cat | $31.41 \pm 0.64$ | $27.32 \pm 0.40$ | $32.59 \pm 0.46$ | $29.90 \pm 1.47$ | $27.39 \pm 3.23$ | $28.18 \pm 0.57$ | $30.73 \pm 0.49$ |
| | deer | $37.47 \pm 2.32$ | $44.59 \pm 0.28$ | $39.53 \pm 0.92$ | $42.77 \pm 0.85$ | $28.76 \pm 4.05$ | $33.95 \pm 1.32$ | $52.42 \pm 1.41$ |
| | dog | $31.17 \pm 0.87$ | $26.41 \pm 0.35$ | $33.82 \pm 0.33$ | $30.68 \pm 1.21$ | $31.95 \pm 2.78$ | $33.73 \pm 3.15$ | $52.31 \pm 2.20$ |
| | frog | $24.56 \pm 1.26$ | $35.89 \pm 0.59$ | $23.40 \pm 0.73$ | $29.77 \pm 1.60$ | $32.87 \pm 4.82$ | $34.05 \pm 2.64$ | $48.20 \pm 1.59$ |
| | horse | $24.79 \pm 0.83$ | $23.87 \pm 0.63$ | $26.91 \pm 0.62$ | $25.06 \pm 0.81$ | $32.53 \pm 4.39$ | $38.86 \pm 3.58$ | $68.17 \pm 2.60$ |
| | ship | $43.19 \pm 4.65$ | $39.69 \pm 0.80$ | $46.05 \pm 0.57$ | $43.47 \pm 2.32$ | $30.15 \pm 9.33$ | $35.85 \pm 4.28$ | $74.13 \pm 1.86$ |
| | truck | $19.79 \pm 0.35$ | $18.22 \pm 0.27$ | $19.71 \pm 0.22$ | $17.96 \pm 0.54$ | $40.91 \pm 7.75$ | $34.47 \pm 2.65$ | $72.34 \pm 0.73$ |
| | *average* | $30.54 \pm 0.71$ | $30.55 \pm 0.11$ | $32.01 \pm 0.19$ | $31.02 \pm 0.25$ | $31.08 \pm 1.81$ | $32.88 \pm 0.73$ | $\mathbf{55.74 \pm 0.51}$ |
| CIFAR100 | aquatic mammals | $42.17 \pm 2.54$ | $35.40 \pm 1.19$ | $42.88 \pm 1.78$ | $39.01 \pm 2.62$ | $25.57 \pm 2.12$ | $29.01 \pm 1.25$ | $47.31 \pm 1.54$ |
| | fish | $37.83 \pm 1.98$ | $32.10 \pm 1.51$ | $36.94 \pm 0.60$ | $35.71 \pm 1.91$ | $21.97 \pm 2.39$ | $30.98 \pm 2.79$ | $39.58 \pm 1.19$ |
| | flowers | $18.09 \pm 0.50$ | $19.17 \pm 0.42$ | $18.64 \pm 0.40$ | $20.22 \pm 1.49$ | $30.54 \pm 6.71$ | $29.80 \pm 3.09$ | $48.77 \pm 1.79$ |
| | food containers | $31.34 \pm 1.00$ | $30.94 \pm 0.93$ | $30.70 \pm 0.58$ | $27.36 \pm 2.04$ | $23.19 \pm 1.65$ | $23.87 \pm 1.13$ | $42.69 \pm 1.76$ |
| | fruit and vegetables | $25.70 \pm 2.10$ | $22.75 \pm 0.46$ | $25.88 \pm 0.31$ | $24.81 \pm 1.18$ | $35.12 \pm 4.33$ | $26.15 \pm 1.34$ | $44.05 \pm 1.02$ |
| | household electrical devices | $26.00 \pm 1.10$ | $20.91 \pm 0.35$ | $27.27 \pm 0.52$ | $22.36 \pm 1.12$ | $25.54 \pm 3.74$ | $23.93 \pm 0.83$ | $29.21 \pm 0.75$ |
| | household furniture | $33.97 \pm 0.47$ | $26.97 \pm 0.71$ | $34.26 \pm 0.62$ | $28.66 \pm 4.48$ | $26.80 \pm 5.16$ | $22.29 \pm 0.35$ | $47.64 \pm 1.37$ |
| | inserts | $22.91 \pm 0.96$ | $25.33 \pm 0.84$ | $23.58 \pm 0.51$ | $24.33 \pm 1.07$ | $23.34 \pm 2.39$ | $28.23 \pm 0.83$ | $40.23 \pm 0.66$ |
| | large carnivores | $26.33 \pm 1.05$ | $29.78 \pm 1.32$ | $28.75 \pm 0.57$ | $30.54 \pm 1.06$ | $36.38 \pm 1.68$ | $32.37 \pm 2.48$ | $60.85 \pm 2.53$ |
| | large man-made outdoor things | $40.66 \pm 4.33$ | $37.86 \pm 1.32$ | $45.53 \pm 1.06$ | $43.49 \pm 3.14$ | $39.77 \pm 5.00$ | $30.71 \pm 1.07$ | $70.37 \pm 1.41$ |
| | large natural outdoor scenes | $58.07 \pm 0.69$ | $60.32 \pm 0.90$ | $56.01 \pm 0.51$ | $59.39 \pm 2.45$ | $27.85 \pm 3.17$ | $32.72 \pm 2.16$ | $56.96 \pm 1.66$ |
| | large omnivores and herbivores | $31.52 \pm 0.76$ | $28.80 \pm 0.46$ | $34.34 \pm 0.82$ | $33.78 \pm 0.95$ | $33.20 \pm 3.81$ | $29.08 \pm 1.35$ | $57.69 \pm 1.40$ |
| | medium-sized mammals | $33.18 \pm 0.43$ | $29.39 \pm 0.78$ | $32.86 \pm 1.30$ | $34.31 \pm 0.84$ | $37.37 \pm 2.60$ | $30.85 \pm 1.56$ | $56.59 \pm 1.56$ |
| | non-insert invertebrates | $25.24 \pm 0.42$ | $27.58 \pm 0.47$ | $24.60 \pm 0.50$ | $25.73 \pm 0.98$ | $23.03 \pm 1.20$ | $24.53 \pm 1.11$ | $28.13 \pm 0.66$ |
| | people | $24.36 \pm 0.66$ | $20.83 \pm 0.47$ | $25.00 \pm 0.87$ | $23.41 \pm 1.28$ | $31.73 \pm 2.29$ | $26.46 \pm 1.14$ | $63.12 \pm 1.63$ |
| | reptiles | $28.78 \pm 1.46$ | $30.05 \pm 0.15$ | $30.14 \pm 0.97$ | $30.74 \pm 1.31$ | $25.00 \pm 2.41$ | $27.32 \pm 1.12$ | $41.48 \pm 0.40$ |
| | small mammals | $35.88 \pm 1.50$ | $39.24 \pm 0.80$ | $37.20 \pm 1.34$ | $41.20 \pm 0.60$ | $29.51 \pm 1.53$ | $28.89 \pm 1.85$ | $46.62 \pm 1.06$ |
| | trees | $37.26 \pm 2.71$ | $39.51 \pm 1.26$ | $38.63 \pm 1.24$ | $41.60 \pm 1.76$ | $35.39 \pm 0.81$ | $36.33 \pm 1.39$ | $73.18 \pm 1.49$ |
| | vehicles 1 | $18.98 \pm 0.41$ | $19.72 \pm 0.57$ | $19.68 \pm 0.68$ | $20.68 \pm 0.96$ | $31.55 \pm 2.30$ | $27.76 \pm 1.88$ | $74.05 \pm 1.63$ |
| | vehicles 2 | $24.53 \pm 1.52$ | $24.37 \pm 0.63$ | $27.32 \pm 0.97$ | $25.49 \pm 2.01$ | $25.14 \pm 1.66$ | $24.68 \pm 1.48$ | $58.09 \pm 1.83$ |
| | *average* | $31.14 \pm 0.45$ | $30.05 \pm 0.38$ | $32.01 \pm 0.30$ | $31.64 \pm 0.34$ | $29.40 \pm 0.92$ | $28.30 \pm 0.65$ | $\mathbf{51.33 \pm 0.45}$ |
| SVHN | 0 | $25.65 \pm 0.15$ | $28.93 \pm 0.55$ | $25.43 \pm 0.45$ | $26.37 \pm 0.23$ | $31.04 \pm 4.68$ | $48.86 \pm 1.17$ | $34.80 \pm 1.07$ |
| | 1 | $29.07 \pm 0.94$ | $31.72 \pm 0.71$ | $28.75 \pm 0.34$ | $29.29 \pm 1.06$ | $38.81 \pm 14.53$ | $36.22 \pm 2.46$ | $35.50 \pm 0.44$ |
| | 2 | $26.06 \pm 0.49$ | $27.01 \pm 0.55$ | $25.90 \pm 0.49$ | $26.54 \pm 0.40$ | $38.99 \pm 8.09$ | $35.80 \pm 1.11$ | $56.45 \pm 1.68$ |
| | 3 | $24.73 \pm 0.29$ | $25.29 \pm 0.33$ | $24.52 \pm 0.56$ | $24.98 \pm 0.35$ | $37.41 \pm 7.81$ | $30.83 \pm 2.58$ | $51.16 \pm 0.91$ |
| | 4 | $26.48 \pm 0.51$ | $28.32 \pm 0.53$ | $26.63 \pm 0.32$ | $27.23 \pm 0.25$ | $41.05 \pm 8.46$ | $33.94 \pm 2.11$ | $59.22 \pm 1.78$ |
| | 5 | $24.12 \pm 0.23$ | $25.90 \pm 0.44$ | $24.33 \pm 0.41$ | $24.71 \pm 0.50$ | $29.96 \pm 5.51$ | $34.70 \pm 1.81$ | $60.51 \pm 0.63$ |
| | 6 | $24.12 \pm 0.37$ | $25.18 \pm 0.49$ | $23.63 \pm 0.56$ | $24.25 \pm 0.27$ | $32.34 \pm 5.84$ | $37.64 \pm 2.15$ | $54.78 \pm 0.90$ |
| | 7 | $27.85 \pm 0.56$ | $29.40 \pm 0.63$ | $26.88 \pm 1.01$ | $28.12 \pm 0.57$ | $39.21 \pm 14.43$ | $44.16 \pm 2.30$ | $64.05 \pm 0.94$ |
| | 8 | $24.49 \pm 0.40$ | $26.45 \pm 0.46$ | $24.47 \pm 0.35$ | $25.11 \pm 0.42$ | $31.96 \pm 5.79$ | $39.07 \pm 2.67$ | $37.75 \pm 1.14$ |
| | 9 | $24.79 \pm 0.57$ | $26.72 \pm 0.67$ | $24.79 \pm 0.23$ | $25.23 \pm 0.56$ | $39.40 \pm 11.69$ | $38.84 \pm 2.13$ | $55.71 \pm 1.42$ |
| | *average* | $25.74 \pm 0.13$ | $27.49 \pm 0.16$ | $25.53 \pm 0.25$ | $26.18 \pm 0.18$ | $36.02 \pm 5.04$ | $38.01 \pm 0.69$ | $\mathbf{50.99 \pm 0.40}$ |