[Reviews · NeurIPS 2019]

Reviewer 1



## After rebuttal The authors well addressed my questions. I think that even inliers do not have a unified learning target in AE-based methods is the key reason why AE-based methods fails. It will be nice to empirically verify this. For example, the authors can do a similar experiment as that in Figure 2. Anyway, I believe this paper has its contribution to the community. I will raise my score to 7. ---------------------------------------------------------------------- Strengths: - I like the idea that we can create some high-level supervision from unlabeled data rather than just using the low-level pixel-level supervision like in the autoencoder. And surprisingly, the algorithms greatly improve the outlier detection performance compared to multiple AE-based methods. Questions: - The key point of this method is to create pseudo labels for the unlabeled data, which will augment the training data by default. When training the AE-based methods. Did the authors also used the same kind of data augmentation? What is the latent dimension of the AE? In my opinion, to fairly compare the proposed algorithm and the AE-based methods, there should also be the same data augmentation for the AE-based methods and the latent dimension for the AE should be smaller than K, which is the number of different operations. This is because the capacity of the network is very important when using AE to detect the outlier. Suppose the network capacity is infinite and the latent dimension is equal to the input dimension, then the model will perfectly remember all the data and has no ability to detect the outliers. So using data augmentation and choosing a small latent dimension might help improve the outlier detection performance since it will make the network has little extra capacity to fit the outliers. Of course the proposed method used the extra high-level information, but I am curious about whether it is the pseudo label that is working or it is just the data augmentation / limited output dimension that is helping more. - What is the point of all the derivations from line 155 to line 180? With the randomly initialized network, isn't it obvious that the inlier will dominate the gradients since it has more samples? These derivations are trying to make things complicated but no extra intuition/information is given. I know the authors are trying to make things more rigorous. But in the derivation, strong and impractical assumptions are needed and the conclusion is trivial. What I care more about is that as the learning procedure goes on, whether the gradients from the inlier or the outlier will dominate. My intuition would be that in the early stage the inlier will dominate, as the authors suggested. However, as the training procedure goes on, each inlier sample will have very small gradients. If the network capacity is really huge, it will also try to fit the outlier. Otherwise the gradient of each inlier sample will be much smaller than that of each outlier sample, though the gradient of ALL the inlier samples may still be larger than that of all the outlier samples. Either a rigorous derivation or an empirical validation will be much interesting than the trivial derivations from line 155 to 180. - The analysis between line 181 and 197 is kind of the same thing as the analysis between line 155 to 180. Suppose you have two vectors g_{in} and g_{out}. The magnitude of g_{in} is much larger than g_{out}. Let g_{sum} = g_{in} + g_{out}. Of course the angle between g_{in} and g_{sum} is much smaller and thus g_{sum} will have a larger projection in the direction of g_{in} than g_{out}. I am glad that the authors validate this empirically in Figure 2, which verifies the intuition. - It seems the choice of the operation set is also important. As the authors mentioned, a digit "8" will still be an "8" when applying a flip operation. The authors argued this will cause a misclassification. Though I agree with this, I think such cases will also harm the training. Ideally speaking, for a digit "8", it will give 0.5 probability to FLIP and 0.5 probability to NOT FLIP. This will make any "8" looks like an outlier based on this operation alone. Of course the model can still remedy this problem by using other operations. What I just argued is just an example where some kinds of operations may make things worse. A good way to check this is to train another model without the flip operator on digit "8" and see if the performance increases. It would also be nice to provide a general guidance about how to select the operations. - How did the authors deal with the regions in the transformed image that has no correspondence in the original image? For example, if we try to shift the image 1 step in the left direction, then how did the authors deal with the right most column in the transformed image? If we just use zeros to fill that column, the network will easily distinguish this for both inlier and outlier samples without learning any high-level representations.

Reviewer 2



Originality The use of self supervision for this particular application is novel as far as I know. The idea of incorporating the self supervised transformations during inference (outlier scoring) adds additional novelty. Quality Clarity The motivation for inlier priority seems overly complicated. This already seems intuitive (there is a lot of calculation to conclude that the expected norm of gradients will be proportional to ratio of inlier/outlier) and it feels as if it can been motivated in an even simpler way, or perhaps leave the mathematical rigor to supplementary material. The paper is clear and introduces both outlier detection and self supervision to the unfamiliar reader. However, I do feel that the paper could cite more self supervision papers (right now it only has 2: the immediate methods used) and at least once call it the popular name “self supervision” once. I suggest this to bring this paper to the attention of that community since knowing that there is an application for their work could inspire them use this benchmark. This would increase the mutual benefits between the two communities (self sup and outliers). Significance I think this is an interesting new application of self supervision. The experimental results look compelling on established benchmarks (although I was not previously familiar with them). From my perspective (familiar with self supervision but not the outlier detection), this looks like a significant contribution, both for connecting the areas but also for the method itself.

Reviewer 3



In this paper, the authors provide a novel unsupervised outlier detection framework for images. The authors first propose a surrogate supervision-based framework to learn the representation of each datum. The authors then provide a theoretical explanation on why the proposed framework would prioritize the inliers in the data set. The authors then provide several outlier score definitions based on the proposed framework. They also conduct a series of experiments to verify the effectiveness of the proposed method and show substantial improvement over baselines. Outlier detection is a classical research problem in unsupervised learning with many applications. The authors provide a novel methodology to attack outlier detection problem for images. Unlike previous methods which largely aim to learn a model/distribution where normal data can be recovered/generated from, the authors introduce surrogate supervision to learn the representation. I think the idea is brilliant. Although the transformations in this paper are image-specific, it is definitely inspiring for outlier detection methods in other fields like text or social networks. The theoretical analysis in Section 3.2 is insightful. It shows how the proposed network would be dominated by inliers. More importantly, I think it shows the relations between how rare the outliers are and how much they will disrupt the training. I believe this part would be much more convincing if the authors could also provide an analysis for AE/CAE based method and do a comparison. This will provide more insight on why the proposed method in this paper outperforms the baselines. The authors conduct thorough experiments on multiple data sets. The experimental results seem promising. A minor concern I have is the setting of \rho. In many real scenarios of outlier detection, \rho could be a much smaller value such as 1% or 0.5%. Do the authors have experimental results on these configurations? %==After rebuttal==% Thanks for the authors' response. Based on the reviews, I think the authors need to clarify the value of the analysis in Section 3.2, i.e. "the quantitative correlation between inliers/outliers’ gradient magnitude and the inlier/outlier ratio". I still think it would be interesting to perform similar analysis on other algorithms, even an extremely naive one. But overall I remain my positive opinion.

[Author Response · NeurIPS 2019]

We sincerely appreciate all reviewers' efforts and valuable comments. Please find our point-by-point rebuttal below:

**Reviewer 1:** Thanks for detailed comments and constructive advice. **Q1:** Data augmentation and latent dimension for
AE. **Reply:** First, we must point out that using data from pseudo classes, as SSD of $E^3Outlier$ did, cannot make AE
perform better. Since AE cannot exploit the discriminative label information of pseudo classes, in original paper we did
not use their data to train AE. As suggested, we train AE with the same augmented data, but the performance typically
becomes worse (e.g. 55.5%/63.9%/54.2%/50.0%/53.8% AUROC on MNIST/F-MNIST/CIFAR10/SVHN/CIFAR100
when $\rho = 10\%$). Second, to fairly compare the quality of learned representation, we must ensure its dimension to be
equal for AE and SSD. Note that SSD's penultimate layer, rather than its final $K$-node classification layer, is used to
yield SSD's learned representation (explained in line 136-138). Thus, AE's hidden layer shares SSD's penultimate layer
dimension, which is fixed to 256 by Wide-ResNet architecture. It is already smaller than input dimension (3072 or 1024)
here. We also test even smaller AE latent dimensions (16, 32, 64, 128): The results show that even for optimal latent
dimension (64) that performs best on most benchmarks, it brings minimal gain to AE performance on difficult datasets
CIFAR10/CIFAR100 (e.g. 56.3%/56.1% AUROC when $\rho = 10\%$), and on simpler datasets MNIST/F-MNIST/SVHN
AE's performance (71.9%/75.6%/53.4%, $\rho = 10\%$) is still far behind $E^3Outlier$ (94.1%/93.3%/86.0%) despite limited
improvement. In fact, choosing AE's latent dimension priorly is difficult in itself. Our test shows that neither way above
helps AE-based methods perform comparably to $E^3Outlier$, and we will add detailed comparisons to paper as suggested.
**Q2:** The point of derivation in line 155-180. **Reply:** Although inlier priority seems intuitive, the derivation not only
justifies this intuition theoretically, but also provides quantitative measure on "how much" priority inliers will gain in
SSD training, i.e. it reveals the quantitative correlation between inliers/outliers' gradient magnitude and the inlier/outlier
ratio. We believe that a rigorous conclusion that matches intuition does NOT make it "trivial". Meanwhile, the method
to analyze the simplified case can serve as a foundation to inspire further theoretical analysis of complex cases. As to
the gradient evolution in training, we did observe that the inliers' gradient magnitude decreases as the training continues,
and the performance will drop moderately if too many training epochs are used (see Fig. 4(i)), which implies a better
fitting of outliers at this stage. However, the network is still observed to classify inliers better and achieve satisfactory
UOD performance. **Q3:** The analysis in line 181-197. **Reply:** We discuss the network updating direction here to
provide a more holistic empirical justification of inlier priority, as magnitude and direction are two key factors for the
back-propagated gradient vector. **Q4:** The choice of operations. **Reply:** We have conducted an ablation study in Sec.
4.2 based on the combination of different operation sets (see Fig. 4(f)) instead of each individual operation, as it is very
time-consuming. In fact, the evaluation and selection of operations is exactly what we are interested in for our next-step
research, and our solution will be training a network to examine the geometric property (e.g. symmetry, straightness)
in an image to guide the selection of operations. **Q5:** The image artifact of shifting operation. **Reply:** In fact, our
experiments show that using operations that create image artifacts (e.g. shifting) alone as surrogate supervision indeed
leads to poor performance just as the reviewer pointed out. However, when they are combined with those operations that
do not create artifacts (e.g. rotation by $90°$), they can improve the performance of surrogate supervision. We simply fill
the artifact with 0, as other padding methods (e.g. nearest neighbor) produce very similar performance. **Improvements:**
**(1)** Please see reply to Q1. **(2)** Please see reply to Q2 and Q3, and it should be clarified that the theoretical analysis was
NOT intended to illustrate the advantages of $E^3Outlier$ over AE-based methods (in fact the discussion for this purpose is
given in Sec. 3.1). More importantly, we must point out that inlier priority does NOT apply to AE-based methods: First,
AE uses the raw image pixels as learning targets, but the intra-class difference of inlier images can be very large, which
means AE does not have an unified learning target. Second, AE is ineffective in learning high-level representations
(discussed in Sec. 3.1), which makes it difficult to capture common high-level semantics of inlier images. Both factors
above disable inliers from being a joint force to dominate the training of AE and produce inlier priority like SSD, which
is also demonstrated by AE's poor UOD performance in empirical evaluations. **(3)** Please see reply to Q4.

**Reviewer 2:** Thanks for the comments and beneficial suggestions. **Q1:** More introduction to self-supervision. **Reply:**
Although both are used in the literature, we prefer surrogate supervision to self-supervision here because it can better
reflect the fundamental difference between our method and the commonly-used autoencoder in UOD (autoencoder is
also viewed as "self-supervised" in some literature). In the camera-ready version where an additional page is granted,
we will provide a more detailed review on this topic as suggested. **Improvements: (1)** Please see reply to Q1.

**Reviewer 3:** Thanks for the comments and useful feedback. **Q1:** Theoretical analysis of AE/CAE. **Reply:** We have
made an attempt to analyze AE/CAE theoretically just like $E^3Outlier$. However, since AE/CAE is trained to reconstruct
the images from the latent representation, its learning target is different from one image to another, which prevents
us from yielding the expectation of its gradient magnitude in theory. In fact, the reason why our method outperforms
AE-based methods is mainly discussed in Sec. 3.1. **Q2:** Smaller outlier ratio. **Reply:** In fact, experiments show that our
method will perform even better when the outlier ratio $\rho$ is set to smaller values, since the inlier priority will be even
larger in such case. For example, when $\rho = 0.5\%$, $E^3Outlier$ achieves 96.0%/93.6%/87.4%/91.0%/80.7% AUROC for
MNIST/F-MNIST/CIFAR10/SVHN/CIFAR100, and results for $\rho = 1\%$ show a similar trend. These results indicate
that our method can perform satisfactorily in a wide range of outlier ratio. **Improvements: (1)** Please see reply to Q1.

[Meta-Review · NeurIPS 2019]

All reviewers converged to acceptance scores for this submission, and the AC agrees with this consensus. However, in preparing a final version for publication, I would recommend a couple of modifications. First, the theoretical analysis from Section 3.2 relating to gradient priority seemed distracting and unnecessary. It is seemingly quite obvious that the gradient magnitude of the inliers or outliers will increase proportionally to the number of samples, and it is not at all clear what additional benefit we get from all the derivations and heuristic approximations presented here (on this point I completely agree with Reviewer 1). Additionally, I don't see how any of this differentiates the proposed method from standard approaches based on AEs. Certainly they would also experience a similar gradient priority if the inlier proportion was greater than the outliers. In any event, by condensing Section 3.2, then there would be more space for moving important modeling details from the supplementary or adding additional experiments. Secondly, the proposed model relies on a 10-layer wide residual network for comparative testing, while alternative AE baselines are built on simpler 4-layer encoder/decoder pairs with no residual connections. Is this really a fair benchmark? In this regard, how can we tell what benefit comes from the proposed surrogate supervision and what is merely the result of different capacity or network structure? Further testing with richer AE structures would be more convincing.